# Leucine inhibits degradation of outer mitochondrial membrane proteins to adapt mitochondrial respiration

Qiaochu Li [1,2], Konstantin Weiss [3], Fuateima Niwa[1,2], Jan Riemer[2,3] & Thorsten Hoppe [1,2,4] ✉

The mitochondrial proteome is remodelled to meet metabolic demands, but how metabolic cues regulate mitochondrial protein turnover remains unclear. Here we identify a conserved, nutrient-responsive mechanism in which the amino acid leucine suppresses ubiquitin-dependent degradation of outer mitochondrial membrane (OMM) proteins, stabilizing key components of the protein import machinery and expanding the mitochondrial proteome to enhance metabolic respiration. Leucine inhibits the amino acid sensor GCN2, which selectively reduces the E3 ubiquitin ligase cofactor SEL1L at mitochondria. Depletion of SEL1L phenocopies the effect of leucine, elevating OMM protein abundance and mitochondrial respiration. Disease-associated defects in leucine catabolism and OMM protein turnover impair fertility in *Caenorhabditis elegans* and render human lung cancer cells resistant to inhibition of mitochondrial protein import. These findings define a leucine–GCN2–SEL1L axis that links nutrient sensing to mitochondrial proteostasis, with implications for metabolic disorders and cancer.

Mitochondrial proteostasis ensures that mitochondria can adjust their function to meet changing demands. Multiple quality control pathways, including mitochondria-associated degradation[1–6], mitochondrial protein translocation-associated degradation[7], mitochondria-derived vesicles[8–10], mitochondria-derived compartments[11,12] and mitophagy[13,14] support this dynamic regulation. The mitochondrial proteome is highly adaptable; for example, the total mitochondrial protein mass in yeast more than doubles under respiratory versus fermentative growth conditions[15]. However, the specific nutrient signals and mechanisms that control this metabolic reshaping of the mitochondrial proteome remain unknown.

Communication between the mitochondria and the cytosol is critical for cellular adaptation to metabolic changes. The outer mitochondrial membrane (OMM), located at the interface between the cytosol and mitochondria, plays a key role in this crosstalk. OMM proteins coordinate protein import[16,17], metabolite transport[18–20], mitochondrial fusion and fission[21] and interactions with other organelles[22]. Altered OMM proteostasis could thus coordinate changes in the cellular environment with mitochondrial activity. Nevertheless, whether and how nutrient signals impact OMM proteostasis for cellular adaptation remains largely unexplored.

Degradation of many OMM proteins is regulated by the ubiquitin-proteasome system (UPS)[3–6,23–31], we therefore explored the role of ubiquitin-dependent regulation of OMM proteostasis in metabolic adaptation. To identify metabolic control of OMM protein stability, we conducted a genetic screen in *Caenorhabditis elegans* using a newly developed green fluorescent protein (GFP)-based reporter system that monitors UPS-mediated degradation of OMM-localized substrates. This approach unexpectedly revealed amino acid metabolism, particularly the branched-chain amino acid (BCAA) leucine (Leu),

[1]Institute for Genetics, University of Cologne, Cologne, Germany. [2]Cologne Excellence Cluster on Cellular Stress Responses in Aging-Associated Diseases, University of Cologne, Cologne, Germany. [3]Institute for Biochemistry, University of Cologne, Cologne, Germany. [4]Center for Molecular Medicine Cologne, University of Cologne, Cologne, Germany. ✉e-mail: thorsten.hoppe@uni-koeln.de

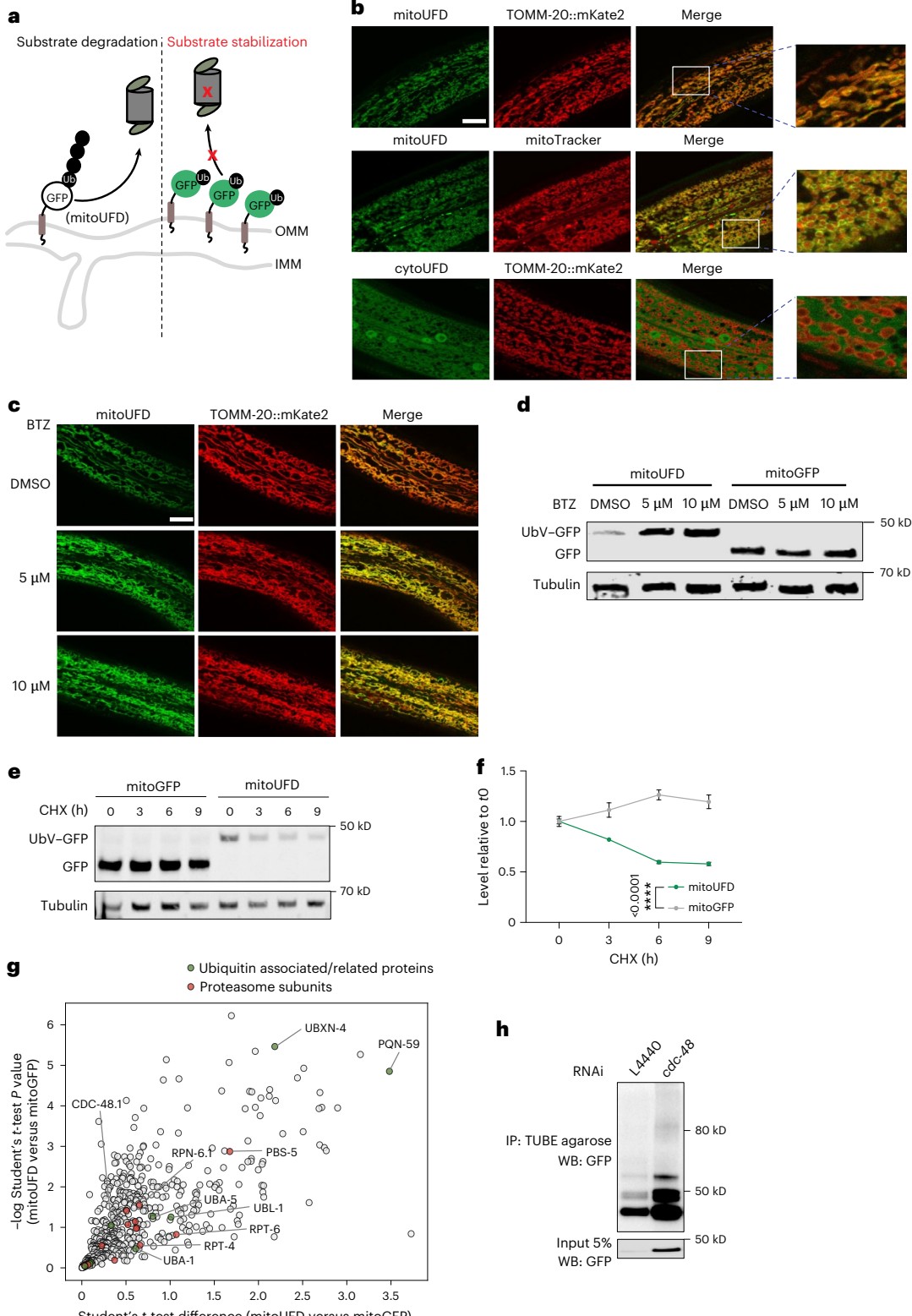

**Fig. 1 | mitoUFD is a model substrate for monitoring protein turnover at the OMM. a**, The mitoUFD model substrate is located at the OMM and a substrate for the UPS. IMM, inner mitochondrial membrane; Ub, ubiquitin. **b**, Confocal images showing that mitoUFD colocalizes with a previously established OMM marker TOMM-20::mKate2 (top row), surrounding the mitoTracker-stained mitochondrial matrix (middle row), and cytoUFD does not colocalize with TOMM-20::mKate2 (bottom row). Scale bar, 10 μm. **c**, Confocal images showing that BTZ treatment at 5 μM or 10 μM concentration stabilizes mitoUFD at the OMM. Scale bar, 10 μm. **d**, Western blot analysis showing that BTZ treatment stabilizes mitoUFD but not mitoGFP. **e**, Western blot showing chase assay of

mitoUFD (degraded) versus mitoGFP (stable). **f**, Quantification of GFP levels in worms by fluorescence imaging over a 9 h CHX-chase assay. Mean ± s.e.m., $n = 5$ independent biological replicates. Two-way ANOVA; the $P$ value indicates the effect of substrates (mitoUFD versus mitoGFP). **g**, A volcano plot showing enriched binding partners of mitoUFD compared with mitoGFP, which includes ubiquitin associated/related proteins and proteasome subunits. **h**, Mitochondrial enrichment followed by pulldown of polyubiquitylated proteins by TUBE agarose in *L4440* control and *cdc-48* RNAi worms; western blot (WB) against GFP to show polyubiquitylated mitoUFD substrate. Input: isolated and lysed mitochondria before TUBE IP. The same number of worms was used for both conditions.

as an essential modulator of OMM protein degradation that tunes respiratory activity, dependent on the conserved amino acid sensor GCN-2. The HRD1 E3 ubiquitin ligase cofactor SEL1L is downregulated by Leu specifically at mitochondria, which leads to the stabilization of OMM substrates and enhancement of mitochondrial respiration. A disease-associated mutation in *BCAT2*, a key enzyme in Leu catabolism, stabilizes OMM proteins and impairs fertility under stress in *C. elegans*. Moreover, human lung cancer cells with elevated intracellular BCAA levels exhibit reduced OMM ubiquitylation and increased resistance to mitochondrial import inhibition. Together, our results identify a conserved mechanism by which Leu regulates OMM proteostasis and mitochondrial function, linking amino acid availability to mitochondrial remodelling and organismal health.

## Results

### A model substrate for monitoring OMM protein degradation

To monitor UPS-mediated degradation of OMM proteins, we developed a GFP-based ubiquitin fusion degradation (UFD) reporter assay in the multicellular organism *C. elegans* (Fig. 1a). UFD substrates, in which a protein is fused to a non-cleavable ubiquitin, are short-lived due to their targeting to the UPS[32,33]. Briefly, we fused non-cleavable ubiquitin–GFP (UbV–GFP) to the transmembrane domain of the OMM protein mitochondrial fission protein 1 (FIS-1) (Fig. 1a). The resulting OMM-anchored UFD substrate, termed mitoUFD is ubiquitously expressed in all *C. elegans* tissues, colocalizes with the OMM protein translocase of outer mitochondrial membrane 20 (TOMM-20)[34] and surrounds the mitochondrial matrix (Fig. 1b). CytoUFD, a UbV–GFP fusion protein lacking the FIS-1-derived transmembrane domain, by contrast, did not colocalize with TOMM-20 (Fig. 1b). Next, we assessed mitoUFD turnover, using mitoGFP as a negative control, which is the same construct but lacking the ubiquitin as the degradation signal. Treatment of worms with the proteasome inhibitor bortezomib (BTZ) increased the abundance of mitoUFD, but not mitoGFP, at the mitochondrial surface (Fig. 1c,d and Extended Data Fig. 1a,b). In addition, overexpression of the proteasome subunit *rpn-6.1* significantly reduced mitoUFD levels (Extended Data Fig. 1c,d). Furthermore, the cycloheximide (CHX) chase assay demonstrated fast turnover of mitoUFD while the levels of the control mitoGFP remained stable (Fig. 1e,f). Taken together, we conclude that mitoUFD is targeted for degradation by the UPS.

In yeast, the AAA⁺ ATPase Cdc48 (p97 in vertebrates)[35,36] and its cofactors extract client OMM proteins for proteasomal degradation[1,3] (Extended Data Fig. 1e). By pulldown of the mitoUFD substrate, we identified CDC-48 as its binding partner, as well as some ubiquitin

associated/related proteins and subunits of the proteasome (Fig. 1g). RNA interference knockdown of *cdc-48/CDC48* or its cofactors *ufd-3/DOA1*, *npl-4/NPL4* and *ufd-1/UFD1* increased mitoUFD but not mitoGFP levels at the mitochondrial surface (Extended Data Fig. 1f–h). In addition, *cdc-48* RNAi knockdown animals accumulated ubiquitylated forms of mitoUFD (Fig. 1h). These results suggest that CDC-48-dependent extraction of mitoUFD from the OMM is required for its proteasomal degradation.

### Amino acid metabolism affects OMM protein degradation

To investigate how mitochondrial homeostasis affects OMM protein degradation, we used RNAi to knockdown 38 nuclear-encoded mitochondrial proteins that have mammalian orthologues and belong to different mito-pathways in the MitoCarta 3.0 datasets[37], including mitochondrial central dogma; protein import, sorting and homeostasis; small-molecule transport; signalling; mitochondrial dynamics; surveillance and oxidative phosphorylation (Extended Data Fig. 2a,b and Supplementary Table 1). In most cases, depletion of these mitochondrial proteins (24/38) increased mitoUFD levels but did not affect, or even decreased, cytoUFD levels (Extended Data Fig. 2c). By contrast, depletion of genes encoding subunits of the respiratory chain complexes had a weaker effect on stabilizing mitoUFD compared to cytoUFD (Extended Data Fig. 2c).

We noticed that knockdown of SLC-25A42 led to a consistently strong elevation of mitoUFD (Extended Data Fig. 2c). SLC-25A42 is a mitochondrial transporter for coenzyme A (CoA)[38], which forms acetyl-CoA in the mitochondria. Acetyl-CoA is involved in many biochemical reactions, most notably the tricarboxylic acid (TCA) cycle. To determine whether other genes involved in the TCA cycle affect mitoUFD, we downregulated eight key enzymes for TCA activity (Extended Data Fig. 2d). Indeed, knockdown of three (*aco-2*, *idh-1* and *sdhb-1*) significantly increased mitoUFD levels (Extended Data Fig. 2e), suggesting a general role for the TCA cycle in regulating UPS-dependent mitoUFD degradation.

Disruption of the TCA cycle has a profound effect on amino acid metabolism[39,40]. TCA cycle intermediates serve as building blocks for the synthesis of various amino acids[41–43], and amino acid catabolism produces intermediates such as acetyl-CoA that feed into the TCA cycle. Given this tight coupling, we hypothesized that disrupting amino acid metabolism might affect mitoUFD levels. To test this, we performed a genetic screen of 135 genes that encode enzymes involved in amino acid metabolism[44] (Supplementary Table 2). Knockdown of 67/135 enzymes involved in amino acid metabolism increased mitoUFD (Fig. 2a and Extended Data Fig. 3a), suggesting that amino

---

**Fig. 2 | Leu controls OMM protein degradation via GCN-2. a**, RNAi knockdown of amino acid metabolic genes has a broad effect on mitoUFD degradation. Fifty per cent of the amino acid metabolic genes stabilize mitoUFD by more than 10% (GFP intensity of 50–250 animals for each gene was measured). **b**, Changes in mitoUFD and cytoUFD levels upon RNAi knockdown of genes involved in amino acid sensing. Mean ± s.e.m., *n* = 4 independent biological replicates with a total of 24–32 animals. ImageJ quantification. Two-way ANOVA with Holm–Sidak correction for multiple comparisons with *L4440* control within mitoUFD or cytoUFD group. **c**, Double knockdown of *hoe-1* and *gcn-2* does not show an additive effect on mitoUFD degradation. Mean ± s.e.m., *n* = 4 independent biological replicates with a total of 24–32 animals. ImageJ quantification. Unpaired two-tailed *t*-test between double knockdown and single-knockdown control. **d**, BCAA catabolic pathway and the key enzymes involved. The five candidates indicated with arrowhead are among the top hits from the targeted genetic screen in **a**. **e**, Changes in mitoUFD and cytoUFD levels upon RNAi knockdown of the five genes involved in BCAA catabolism in the top hits. Quantification by worm sorter. Mean ± s.e.m., *n* = 3 independent biological replicates with 50–250 animals in each replicate. One-way ANOVA with Fisher's least significant difference (LSD) test for multiple comparisons was performed using *L4440* as the control for both mitoUFD and cytoUFD. While the knockdown of *mccc-1* and *ech-1.2* did not result in statistically significant changes in cytoUFD (*P* > 0.05), the average percentage changes were greater than 10%. **f**, After Leu,

isoleucine or valine supplementation for 3 h, mitoUFD and cytoUFD levels were measured. Mean ± s.e.m., *n* = 4 independent biological replicates (with a total of 24–32 animals). ImageJ quantification. Two-way ANOVA with Holm–Sidak correction for multiple comparisons with *L4440* control within the mitoUFD or cytoUFD group. **g**, Leu supplementation stabilizes mitoUFD at 20 mM and 50 mM. mitoGFP was used for control. Mean ± s.e.m., *n* = 6 independent biological replicates for mitoUFD (with a total of 36–48 animals) and *n* = 8 independent biological replicates for cytoUFD (with a total of 48–64 animals). ImageJ quantification. Two-way ANOVA with Holm–Sidak correction for multiple comparisons with 0 mM Leu control. **h**, RNAi knockdown of *gcn-2* but not *let-363* followed by 3 h Leu supplementation abolished Leu-induced increases in mitoUFD levels. Mean ± s.e.m., *n* = 4 independent biological replicates with a total of 24–32 animals. ImageJ quantification. One-way ANOVA with Holm–Sidak correction for multiple comparisons with *L4440* control. **i**, Western blot showing polyubiquitylated mitoUFD substrate expressed in HEK293 cells. Cells treated with BTZ at 0.2 or 1 μM showed stabilization of mitoUFD compared with control treatment. **j**, Confocal imaging of HEK293 cells expressing mitoUFD after treatment with 0, 1 or 3 mM Leu for 3 h. Scale bar, 20 μm. **k**, CHX chase assay showing the increased stability of human mitoUFD upon 1 mM Leu treatment. Mean ± s.e.m., *n* = 4 independent biological replicates. Two-way ANOVA; the *P* value indicates Leu treatment effect. NS, not significant.

acid metabolism has a broad impact on OMM protein degradation. By contrast, knockdown of genes involved in amino acid transport (*aat-1*, *aat-2*, *aat-3* and *F13G3.7*), glucose sensing (*egl-30* and *gpa-4*)[45,46] and metabolite transport (*C16C10.1*, *slc-25A21* and *slc-25A29*)[12,38] had little or no effect on mitoUFD (Extended Data Fig. 3b). Of these 67 genes, knockdown of 16 led to a >20% increase in mitoUFD levels (Fig. 2a and Extended Data Fig. 3a). These top 16 genes belong to different metabolic pathways involving 14 amino acids (Extended Data Fig. 3c).

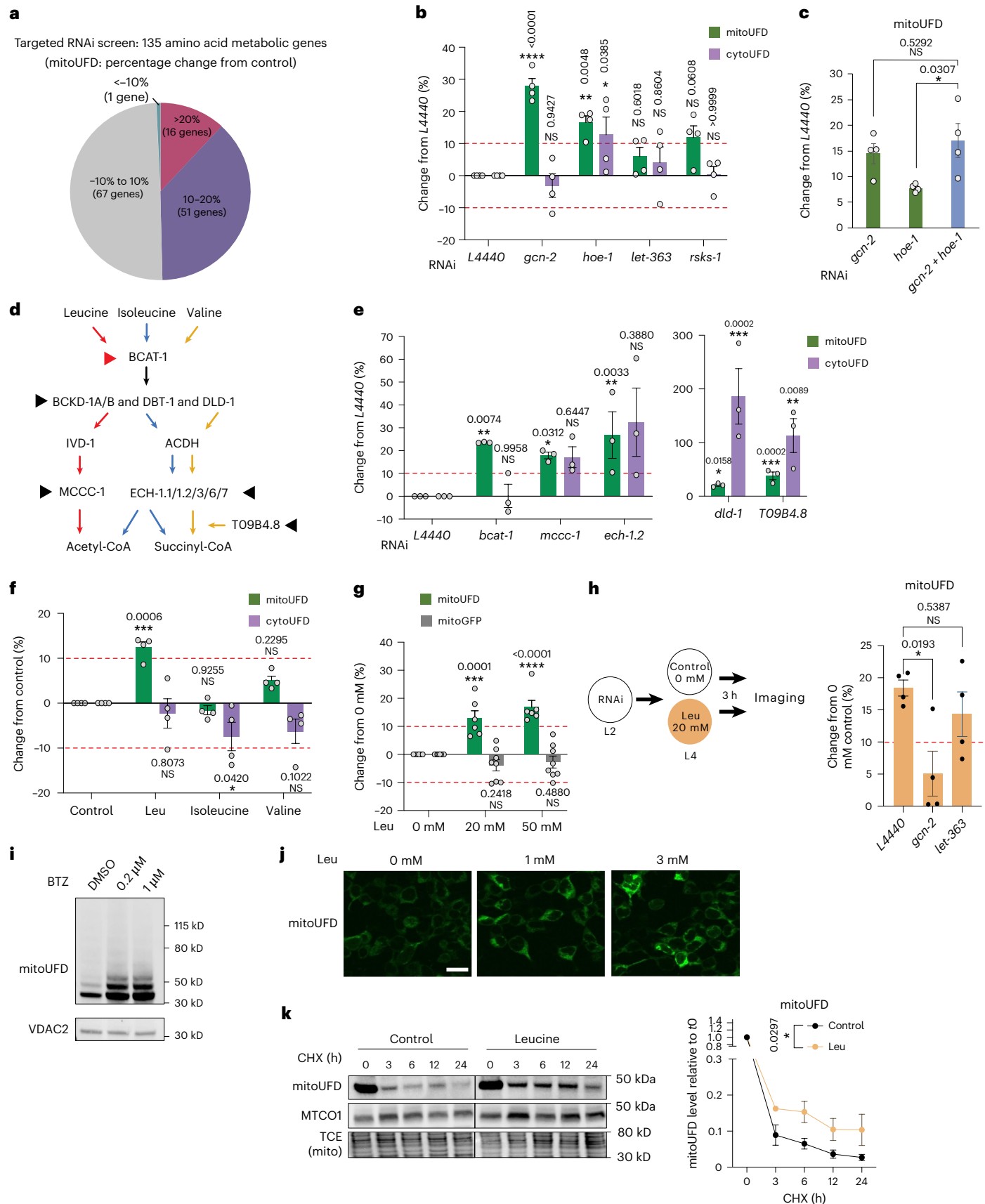

Given this widespread influence, we hypothesized that amino acid sensor(s) might link amino acid availability to mitoUFD degradation. We focused on GCN-2, a cytosolic kinase that is activated by uncharged tRNA in response to amino acid imbalance[47–50]. Knockdown of *gcn-2* resulted in a strong, selective increase in mitoUFD (Fig. 2b and Extended Data Fig. 4a). Knockdown of *hoe-1*, which encodes the tRNA 3′ processing enzyme whose loss is known to impair tRNA maturation, also increased mitoUFD levels (Fig. 2b), consistent with reduced activation of GCN-2 under tRNA deficiency[50]. Double knockdown of *gcn-2* and *hoe-1* increased mitoUFD levels similarly to knockdown of *gcn-2* alone (Fig. 2c), suggesting that they act in the same pathway. Importantly, control mitoGFP levels were not affected by knockdown of *gcn-2* (Extended Data Fig. 3d), indicating that GCN-2 is specifically required for UPS-mediated turnover. Since amino acids are well known to activate mTORC1 signalling[51], we also examined the *C. elegans* orthologue of mTOR *let-363* and the mTORC1 downstream effector *rsks-1/S6K*. Knockdown of *let-363* did not affect mitoUFD and knockdown of *rsks-1* only slightly increased mitoUFD levels (Fig. 2b), suggesting that the regulation of OMM protein degradation is independent of mTORC1 signalling. These data reveal that GCN-2 links amino acid availability to changes in protein degradation at the OMM.

### Leu selectively inhibits OMM protein degradation

We next asked which amino acid(s) regulate mitoUFD levels. Among the top hits in our screen were five enzymes involved in the catabolism of the BCAAs Leu, isoleucine and valine (*bcat-1*, *dld-1*, *mccc-1*, *ech-1.2* and *T09B4.8*; Fig. 2d and Extended Data Fig. 3c). We validated the five genes and compared their impact on the degradation of mitoUFD and cytoUFD. Knockdown of *bcat-1*, which encodes the first enzyme in the BCAA catabolic pathway, increased mitoUFD but not cytoUFD, whereas knockdown of the other four downstream enzymes increased both mitoUFD and cytoUFD levels (Fig. 2e and Extended Data Fig. 4a). These data recapitulate the effects of GCN-2 and suggest a specific role for BCAT-1 in the regulation of mitoUFD.

As the loss of *bcat-1* increases cellular levels of free BCAAs[52,53], we hypothesized that elevated BCAA concentrations would mimic depletion of *bcat-1*. To test this, we treated the worms with high concentrations of Leu, isoleucine or valine. We found that 3 h of treatment with Leu, but not isoleucine or valine, significantly increase mitoUFD levels without affecting cytoUFD (Fig. 2f and Extended Data Fig. 4b), phenocopying *gcn-2* and *bcat-1* knockdowns. mitoGFP levels were not affected (Fig. 2g), indicating that Leu does not broadly alter OMM protein levels but specifically regulates UPS-dependent OMM protein degradation. Notably, Leu supplementation failed to elevate mitoUFD levels when GCN-2 was depleted, whereas Leu was still able to elevate mitoUFD levels when *let-363* was knocked down (Fig. 2h), suggesting that high levels of Leu reduce proteolytic activity at the OMM via GCN-2-mediated amino acid sensing independent of mTOR.

To determine whether the Leu-mediated regulation of mitoUFD is evolutionarily conserved, we expressed mitoUFD in human HEK293 cells. The mitoUFD substrate was ubiquitylated and targeted for proteasomal degradation in HEK293 cells (Fig. 2i and Extended Data Fig. 4c,d) and it displayed increased stability in the presence of excess Leu (Fig. 2j,k). These data suggest that Leu-dependent regulation of OMM protein degradation is conserved in worms and human cells.

### Leu decreases ubiquitylation of OMM proteins via SEL-1–SEL-11

We next asked how Leu treatment increases the stability of OMM proteins. We did not observe any significant differences in the activity of the 26S proteasome between HEK293 cells with and without Leu treatment (Extended Data Fig. 5a,b). By contrast, global ubiquitylation of OMM proteins was significantly reduced by Leu supplementation, whereas whole-cell ubiquitylation was not affected (Fig. 3a and Extended Data Fig. 5c). Furthermore, cells treated with a GCN-2 inhibitor[54] showed a substantial reduction in OMM protein ubiquitylation, comparable to that achieved with Leu supplementation, and these effects were not additive (Fig. 3b). These results suggest that excess Leu specifically reduces OMM but not cytosolic protein ubiquitylation via GCN-2 inhibition.

To determine how Leu reduces the ubiquitylation of OMM proteins, we performed proteomic analysis to identify ubiquitin ligases whose levels changed in response to Leu supplementation. We treated worms with 20 mM Leu or with dH$_2$O. After 3 h of treatment, whole-worm lysates and mitochondria-enriched fractions were analysed by mass spectrometry. Interestingly, we found that SEL-1, the worm orthologue of SEL1L, which is known to form an E3 ubiquitin ligase complex together with HRD1 (SEL-11 in *C. elegans*) at the endoplasmic reticulum membrane[55,56], was significantly reduced upon Leu treatment in the mitochondrial fraction, and this change was not observed in the whole worm (Fig. 3c and Extended Data Fig. 6a). HRDL-1 and RNF-126 were also downregulated in the mitochondrial fraction, but their changes were not statistically significant ($-\log_{10}P < 1.3$) (Fig. 3c). Therefore, we focused on SEL-1. We confirmed that the decrease in SEL-1 abundance was specific to the mitochondrial fraction by western blot (Fig. 3d). Moreover, the levels of the endoplasmic reticulum-associated degradation (ERAD) substrate CPL-1*, whose degradation is mediated by SEL-1 and SEL-11 (ref. 57), were not stabilized by Leu supplementation (Extended Data Fig. 6b,c).

SEL-1 is membrane bound in worms, and cofractionates with the OMM-anchored mitoUFD (Extended Data Fig. 6d). SEL-1 and SEL-11 exhibit enriched binding to mitoUFD compared with mitoGFP by more than two fold (Extended Data Fig. 6e), suggesting that the SEL-1–SEL-11 E3 ligase complex directly interacts with and ubiquitylates OMM proteins. Immunogold analysis further supported the localization of SEL-1 to both the endoplasmic reticulum and OMM (Extended Data Fig. 6g,h). We performed RNAi knockdown of *sel-1* and *sel-11* and indeed observed

**Fig. 3 | High levels of Leu downregulate mitochondria-associated SEL1L/SEL-1 and reduce global ubiquitylation of OMM proteins. a**, Ubiquitylation of whole cell proteins and OMM proteins of cells cultured in medium supplemented with 0 mM or 1 mM Leu and/or 0 µM or 1 µM BTZ. For each sample, 10 µg proteins were loaded. Ubiquitylation levels were normalized to the total OMM protein amount. Mean ± s.e.m., *n* = 4 independent biological replicates. Unpaired two-tailed *t*-test. **b**, Ubiquitylation of whole-cell proteins and OMM proteins of cells cultured in medium supplemented with 0 mM or 1 mM Leu and/or 0 µM or 10 µM GCN-2 inhibitor for 3 h. For each sample, 10 µg proteins were loaded. Ubiquitylation levels were normalized to the total OMM protein amount. Mean ± s.e.m., *n* = 3 independent biological replicates. Unpaired two-tailed *t*-test. **c**, A volcano plot showing the proteomic analysis of Leu (20 mM, 3 h treatment) on enriched mitochondria (mito). *P* values were determined by an unpaired two-tailed *t*-test. E3 ligases/cofactors are labelled in red. SEL-1 was the only significantly downregulated ($-\log_{10}P > 1.3$) E3 ligase/cofactor in enriched mitochondria upon Leu treatment. **d**, Western blot quantification of SEL-1 in enriched mitochondria and whole worms. Animals were treated with and without 20 mM Leu supplementation for 3 h. Unpaired two-tailed *t*-test. Mean ± s.e.m., *n* = 5 independent biological replicates. **e**, CHX chase assay showing the increased stability of mitoUFD substrate in worms upon RNAi knockdown of *sel-1* and *sel-11*. Mean ± s.e.m., *n* = 3 independent biological replicates. **f**, Confocal imaging showing mitoUFD stabilization at the OMM upon RNAi knockdown of *sel-1* or *sel-11*. Stabilized mitoUFD is colocalized with TOMM-20. **g**, ImageJ quantification of mitoUFD levels upon RNAi knockdown of *sel-1* or *sel-11*. Mean ± s.e.m., *n* = 4 independent biological replicates with 24–32 animals in each replicate. One-way ANOVA with Holm–Sidak correction for multiple comparisons with *L4440* control. **h**, Western blot quantification of SEL1L and HRD1 protein level with and without 3 h Leu supplementation in the whole-cell, mitochondrial (P10) and microsome (P100) fraction. Mean ± s.e.m., *n* = 7 independent biological replicates. Unpaired two-tailed *t*-test.

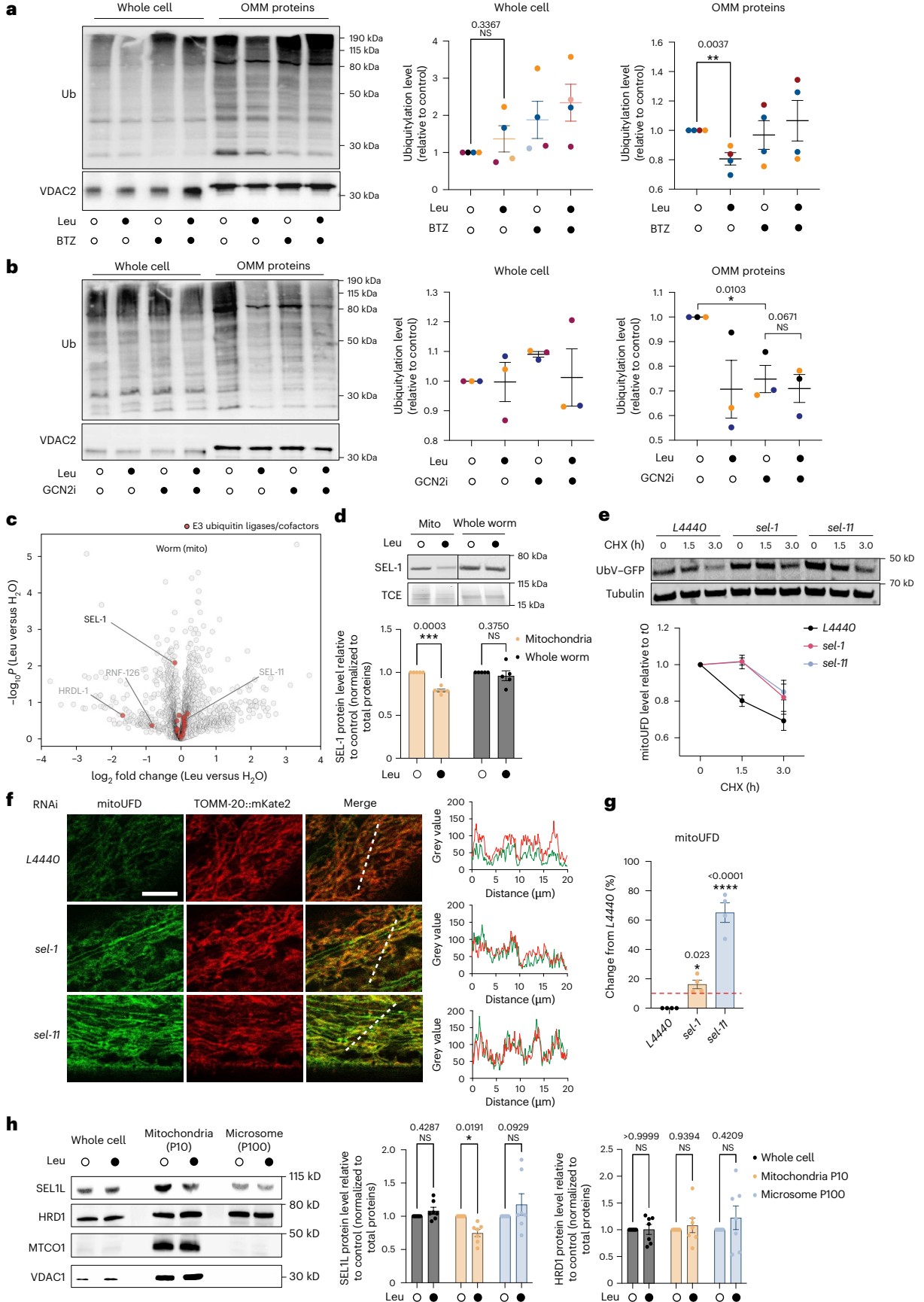

an increased stability of the mitoUFD substrate at the OMM (Fig. 3e–g and Extended Data Fig. 6f), suggesting that the reduced abundance of SEL-1 due to high levels of Leu limits ubiquitin-dependent degradation of OMM proteins. In addition, in HEK293 cells, Leu supplementation reduced the level of mitochondrial-associated SEL1L, whereas the level of SEL1L in the endoplasmic reticulum-derived microsomal fraction remained unchanged (Fig. 3h), suggesting a conserved regulation of SEL-1/SEL1L by Leu at the OMM in worms and human cells. Together, our results suggest that high Leu concentrations reduce SEL-1–SEL-11-dependent degradation of OMM proteins.

### Leu supplementation triggers OMM proteome expansion

To determine whether Leu regulates endogenous OMM proteins, we re-analysed our proteomics data by filtering mitochondrial proteins using MitoCarta 3.0 and OMM annotations (Fig. 4a). We observed an overall increase in the mitochondrial and OMM proteome, but not in the total worm proteome upon Leu supplementation (Extended Data Fig. 7a–d). To evaluate whether specific protein classes were regulated by Leu supplementation, we performed Gene Ontology (GO) enrichment analysis and found that mitochondrial transporters, protein processing and organization, TCA cycle and respiratory chain activity were highly enriched for upregulated proteins (Extended Data Fig. 7e).

To exclude a possible effect of Leu on protein translation, we repeated the proteomic analysis in the presence of CHX to block cytosolic translation (Fig. 4a). CHX treatment did not eliminate the overall increase in OMM proteins upon Leu treatment (Fig. 4b–d and Extended Data Fig. 7f), supporting our hypothesis that the increase in OMM proteins is the result of reduced post-translational protein degradation. We found that nine OMM proteins in worms were consistently upregulated in response to Leu treatment in two independent proteomic analyses and were also upregulated by Leu in the presence of CHX (Fig. 4d and Extended Data Fig. 7b,f). The nine OMM proteins are DNJ-9 (DNAJC11, regulation of protein import and sorting), FKB-6 (FKBP8, regulation of apoptosis), GOP-3 (SAMM50, assembly of TOMM-40 into the TOM complex), MTCH-1 (MTCH1/2, protein insertase that mediates insertion of transmembrane proteins into the OMM), PGAM5 (PGAM5, regulation of mitophagy and mitochondrial dynamics), TOMM-40 (TOMM-40, subunit of the TOM complex, import of protein precursors into mitochondria), VDAC-1 (VDAC1/2/3, the channel at the OMM allows the transport of calcium and small molecules), KMO-1 (KMO, mitochondrial dynamics) and FZO-1 (MFN1/2, mitochondrial fusion). Interestingly, four of them (TOMM-40, GOP-3, MTCH-1 and DNJ-9) are involved in mitochondrial protein import and assembly, suggesting that Leu treatment increases mitochondrial protein import capacity.

Indeed, there was a consistent upregulation of the total mitochondrial proteome upon Leu treatment in two independent proteomic analyses (Extended Data Fig. 7c,g), which was maintained in the presence of CHX (Fig. 4b,c,e). CHX treatment alone depleted mitochondrial proteins (Fig. 4b and Extended Data Fig. 7h). As CHX specifically reduces translation in the cytosol while leaving mitochondrial translation unaffected[58], the reduced mitochondrial protein content was probably due to reduced import as a compensatory response to arrested cytosolic protein synthesis, in agreement with a previous report[59].

To determine whether this response is conserved, we performed the same experiment with HEK293 cells. We observed a striking increase in the abundance of the OMM proteome in the presence of Leu regardless of CHX treatment (Fig. 4f–j and Extended Data Fig. 7i–k), including seven orthologues of the nine *C. elegans* OMM proteins regulated by Leu. Again, several upregulated proteins, such as SAMM50, DNAJC11, TOMM-40 and MTCH1/2, are involved in protein import and assembly (Fig. 4i). We confirmed the increased protein stability of TOMM-40 upon Leu supplementation by western blot (Extended Data Fig. 8a,b). The conserved increase in these proteins along with the mitochondrial proteome upon Leu supplementation suggests a fundamental role for Leu in rewiring the mitochondrial proteome by modulating the import machinery. GO term analysis revealed a consistent enrichment of upregulated mitochondrial proteins involved in mitochondrial transport, amino acid metabolism, TCA cycle and respiratory chain activity in both worms and HEK293 cells (Fig. 4k,l and Supplementary Table 3), suggesting an overall increase in mitochondrial metabolic activity in response to excess Leu.

Together, our results suggest that Leu expands the OMM proteome by reducing the degradation of endogenous OMM proteins, including the subunit of the TOM complex TOMM-40, which allows the import of protein precursors into mitochondria, thereby remodelling the mitochondrial proteome involved in metabolism and respiration.

### Leu-driven OMM remodelling boosts mitochondrial respiration

We hypothesized that the reduced OMM protein degradation and increased abundance of the mitochondrial proteome in response to Leu treatment would improve mitochondrial activity. Indeed, 3 h of Leu treatment resulted in increased mitochondrial respiration in both worms and human cells (Fig. 5b and Extended Data Fig. 9a,b). Notably, although treating HEK293 cells with other essential amino acids resulted in a slight increase in mitochondrial respiration, Leu had the highest impact (Extended Data Fig. 9a,b).

To test whether the Leu-induced increase in mitochondrial respiration requires elevated OMM protein abundance, we examined worms

---

**Fig. 4 | Leu supplementation reduces the degradation of endogenous OMM proteins and increases mitochondrial protein content. a**, The experimental paradigm of proteomic profiling of whole worms and mitochondria of worms supplemented with dH$_2$O, 20 mM Leu, CHX and Leu + CHX for 3 h. Four independent biological replicates were analysed for each condition. **b**, The percentage of proteins increased or decreased in the whole-worm proteome, MitoCarta 3.0 proteins and OMM proteins for each comparison indicated. **c**, The log$_2$ fold change of the whole-worm proteome, MitoCarta 3.0 proteins and OMM proteins comparing Leu + CHX with CHX treatment. Two-tailed Mann–Whitney *U* test between whole-worm proteins and other protein groups. **d**, Volcano plot showing changes of OMM proteins upon Leu + CHX treatment compared with CHX treatment. *P* values were determined by an unpaired two-tailed *t*-test. OMM proteins show the tendency to be more abundant in Leu-treated worms in the presence of CHX. **e**, Volcano plot showing changes of whole worm and MitoCarta 3.0 proteins upon Leu + CHX treatment compared with CHX treatment. *P* values were determined by an unpaired two-tailed *t*-test. Mitochondrial proteins show the tendency to be more abundant in Leu-treated worms in the presence of CHX. **f**, The experimental paradigm of proteomic profiling of HEK293 whole cells and mitochondria of cells supplemented with dH$_2$O, Leu, CHX and Leu + CHX for 3 h. Four independent biological replicates were analysed for each condition.

**g**, The percentage of proteins increased or decreased in the whole cell proteome, MitoCarta 3.0 proteins and OMM proteins in each comparison indicated. **h**, The log$_2$ fold change of the whole cell proteome, MitoCarta 3.0 proteins and OMM proteins comparing Leu + CHX with CHX treatment. Two-tailed Mann–Whitney *U* test between whole cell proteins and other protein groups. **i**, A volcano plot showing changes of OMM proteins upon Leu + CHX treatment compared with CHX treatment. *P* values were determined by an unpaired two-tailed *t*-test. OMM proteins show the tendency to be more abundant in Leu-treated cells in the presence of CHX. **j**, A volcano plot showing the changes in whole cell and MitoCarta 3.0 proteins upon Leu + CHX treatment compared with CHX treatment. *P* values were determined by an unpaired two-tailed *t*-test. Mitochondrial proteins show the tendency to be more abundant in Leu-treated cells in the presence of CHX. **k**, Enriched GO terms (*P* < 0.001 and FDR < 0.2 cut-off) for upregulated proteins from the worm mitochondrial proteome in the Leu + CHX condition compared with the CHX condition. **l**, Enriched GO terms (*P* < 0.001 and FDR <0.2 cut-off) for upregulated proteins from the HEK293 cell mitochondrial proteome in the Leu + CHX condition compared with the CHX condition. A full list of significantly enriched GO terms is presented in Supplementary Table 3 for worm and HEK293 cells.

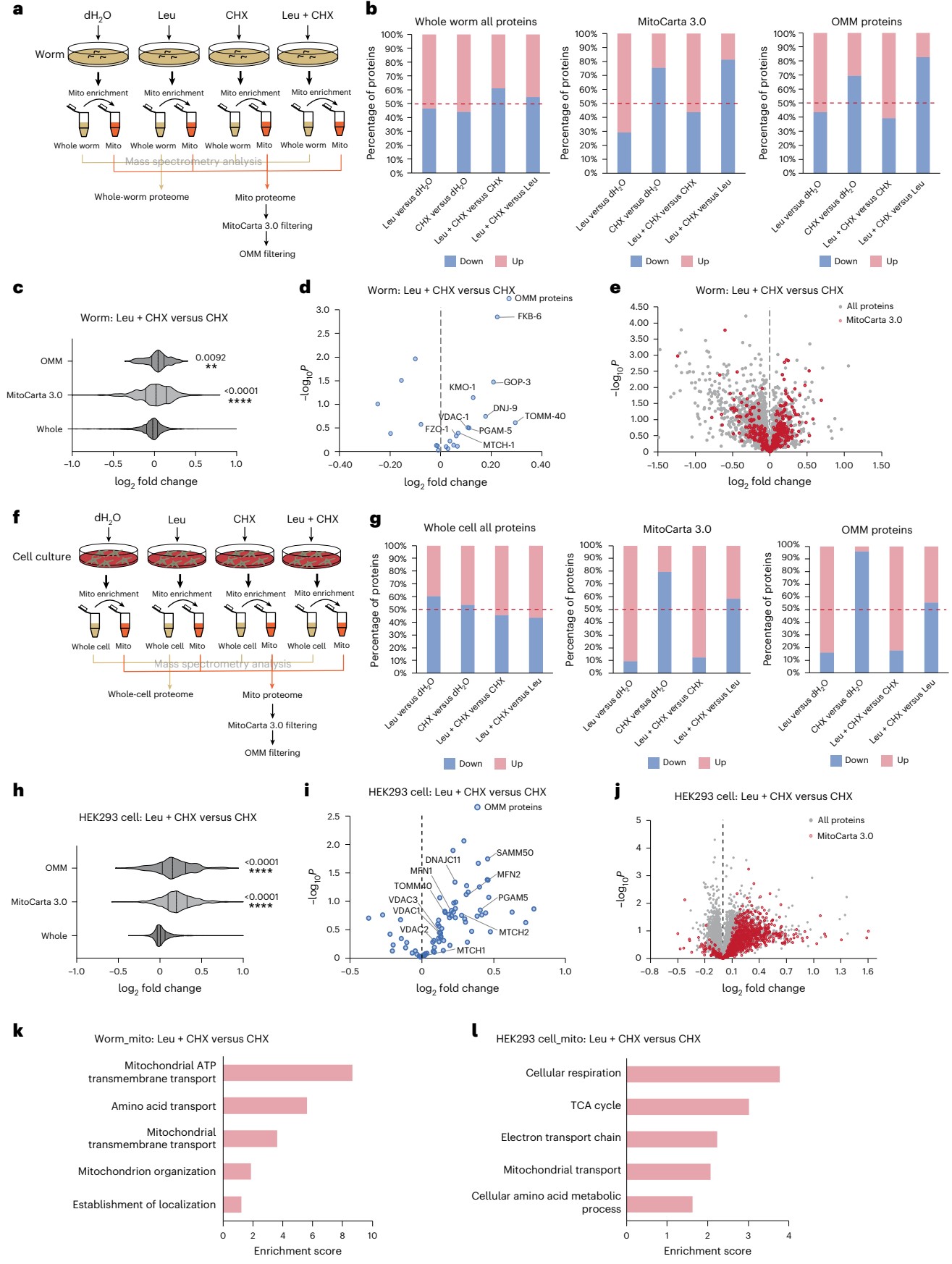

overexpressing *cdc-48.1* (ref. 60) (Extended Data Fig. 9c). Overexpression of *cdc-48.1* reduced the levels of mitoUFD (Fig. 5a), consistent with its role in extraction and degradation of OMM proteins. Notably, the maximal respiratory rate was lower in worms overexpressing *cdc-48.1* compared with controls, and this rate was unaffected by Leu supplementation in contrast to controls (Fig. 5b). Proteomic analysis indicated that *cdc-48*.1 overexpression prevented the increase in the OMM and mitochondrial proteome in response to Leu (Fig. 5c–f). Similarly, we found that overexpression of GCN-2 reduced mitoUFD levels (Fig. 5g and Extended Data Fig. 9d) and prevented the Leu-induced increase in maximal respiration (Fig. 5h and Extended Data Fig. 9e). To investigate whether mitochondrial import is required for the Leu-dependent increase in mitochondrial respiration, we examined the role of TOMM-40. Acute *tomm-40* knockdown abolished the Leu-dependent increase in their maximal respiratory rate (Fig. 5i and Extended Data Fig. 9f). Moreover, Leu treatment failed to enhance mitochondrial respiration in HEK293 cells when mitochondrial import was inhibited by MitoBlock-12 (MB-12) treatment[61,62] (Extended Data Fig. 9g,h). This supports the idea that the Leu-induced increase in mitochondrial respiration is mediated by the mitochondrial protein import machinery.

Importantly, RNAi knockdown of *sel-1* or *sel-11* resulted in enhanced mitochondrial respiration (Fig. 5j and Extended Data Fig. 9i), phenocopying Leu supplementation. This observation suggests that the Leu-dependent reduction of SEL-1 and the resulting increased stability of OMM proteins enhanced mitochondrial respiration. Taken together, these results suggest that Leu acts via stabilizing OMM proteins, facilitating mitochondrial proteome expansion through the import machinery, to increase mitochondrial respiratory capacity.

## BCAA–OMM proteostasis axis influences fertility and cell growth

Patients with an E264K mutation in *BCAT2* (the orthologue of *C. elegans bcat-1*) have reduced BCAT2 activity and elevated plasma levels of BCAAs[63]. To determine how this disease-associated mutation affects organismal physiology, we used CRISPR–Cas9 to generate a *bcat-1(E279K)* mutant in *C. elegans*, which corresponds to human *BCAT2^E264K* (Extended Data Fig. 10a)[63]. Homozygous mutant worms were lethal, so we examined heterozygous *bcat-1(E279K/+)* worms. We observed higher levels of mitoUFD throughout *bcat-1(E279K/+)* compared with wild-type (WT) animals (Fig. 5k and Extended Data Fig. 10b), suggesting reduced UPS activity at the OMM. Despite this phenotype, *bcat-1(E279K/+)* animals produced a normal number of eggs, with no differences in brood size or hatch rate compared with WT animals (Fig. 5l and Extended Data Fig. 10c). Given our data that *gcn-2* regulates UPS activity at the OMM, we asked whether disrupting this regulatory node affects fitness in the context of impaired BCAA metabolism. To test this, we knocked down *gcn-2* in *bcat-1(E279K/+)* animals. Knockdown of *gcn-2* alone did not impact brood size or egg hatch rate in WT animals, but in the *bcat-1(E279K/+)* background, it caused a significant reduction in brood size (Fig. 5l). This genetic interaction suggests that *bcat-1(E279K/+)* animals are sensitized to further disruption of mitochondrial proteostasis. Overexpression of *gcn-2*, which enhanced OMM protein degradation (Fig. 5g), reduced egg laying in WT animals, without affecting hatch rate (Fig. 5m and Extended Data Fig. 10d). These results suggest that an optimal level of BCAA metabolism and OMM protein degradation is linked to animal fertility.

Defects in BCAA metabolism are associated with tumour growth and progression[64,65]. We measured intracellular BCAA levels in three non-small cell lung cancer cell lines, H2030, H1437 and H1666, and found that H2030 and H1666 have higher intracellular BCAA levels compared with H1437 (Fig. 5n). The increased BCAA level in H2030 is probably due to a loss-of-function mutation in the *BCAT2* gene (Extended Data Fig. 10e). We isolated OMM proteins from the three lung cancer cell lines and found that OMM protein ubiquitylation was significantly lower in H2030 and H1666 compared with H1437 (Fig. 5o). Interestingly, blocking mitochondrial import with MB-12 substantially inhibited the proliferation of H1437 and HEK293 cells, but only mildly affected the proliferation of H2030 and H1666 cells with increased MB-12 concentration (Fig. 5p–r and Extended Data Fig. 10f–j). Collectively, our results suggest that the BCAA-dependent regulation of OMM proteostasis is crucial for maintaining animal fertility and cell viability.

## Discussion

Our work reveals a conserved, nutrient-responsive mechanism that regulates mitochondrial proteome homeostasis through targeted modulation of protein ubiquitylation at the OMM. Using a novel reporter system (mitoUFD), we demonstrate that Leu, one of the essential BCAAs, stabilizes OMM proteins by decreasing their ubiquitin-dependent degradation. This process occurs through a conserved pathway involving GCN2 in response to Leu supplementation. As a result, key components of the mitochondrial import machinery, such as TOMM40, accumulate,

---

**Fig. 5 | Leu–OMM proteostasis axis boosts mitochondrial respiration and influences fertility and cell proliferation. a**, *cdc-48.1* overexpression (OE) reduces mitoUFD levels. Mean ± s.e.m., *n* = 4 independent biological replicates. Unpaired two-tailed *t*-test. **b**, Seahorse OCR measurement for WT and *cdc-48.1* OE animals with/without 50 mM Leu supplementation. Two-way ANOVA with Fisher's LSD test for multiple comparisons. Mean ± s.e.m., *n* = 4 independent biological replicates. **c**, The percentage of proteins increased or decreased in whole-worm proteome, MitoCarta 3.0 proteins and OMM proteins in Leu-treated *cdc-48.1* OE animals. **d**, A volcano plot showing changes of OMM proteins upon Leu treatment. *P* values were determined by an unpaired two-tailed *t*-test. **e**, A volcano plot showing changes of whole-worm and MitoCarta 3.0 proteins upon Leu treatment. *P* values were determined by an unpaired two-tailed *t*-test. **f**, The log$_2$ fold change of whole-worm proteome, MitoCarta 3.0 proteins and OMM proteins upon Leu treatment. Two-tailed Mann–Whitney *U* test between whole-worm proteins and other protein groups. **g**, *gcn-2* OE reduces mitoUFD levels. Mean ± s.e.m., *n* = 4 independent biological replicates. Unpaired two-tailed *t*-test. **h**, Seahorse OCR measurement for control SCR OE and *gcn-2* OE animals with or without 50 mM Leu supplementation. Two-way ANOVA with Fisher's LSD test for multiple comparisons. Mean ± sem, *n* = 4 independent biological replicates. Statistical analysis in Extended Data Fig. 9e. **i**, Seahorse OCR measurement for control *L4440* RNAi and *tomm-40* RNAi animals with or without 50 mM Leu supplementation. Two-way ANOVA with Fisher's LSD test for multiple comparisons. Mean ± s.e.m., *n* = 4 independent biological replicates. Statistical analysis in Extended Data Fig. 9f. **j**, Seahorse OCR measurement for control *L4440* RNAi and *sel-1*, *sel-11* RNAi animals. One-way ANOVA with Fisher's LSD test for multiple comparisons. Mean ± s.e.m., *n* = 4 independent biological replicates. Statistical analysis in Extended Data Fig. 9i. **k**, *bcat-1(E279K)/+* worms showed higher mitoUFD levels compared with WT animals. Mean ± s.e.m., *n* = 3 independent biological replicates with a total of 18–24 animals. ImageJ quantification. **l**, Brood size of WT and *bcat-1(E279K)/+* animals with *gcn-2* RNAi knockdown from L4 stage compared with *L4440* control knockdown. One-way ANOVA with Fisher's LSD test for multiple comparisons. **m**, Brood size of animals with *gcn-2* OE from L4 stage. Unpaired two-tailed *t*-test. **n**, BCAA levels in H2030, H1437 and H1666. Unpaired two-tailed *t* test. Mean ± s.e.m., *n* = 3 independent biological replicates with 3 technical replicates each biological replicate. **o**, Ubiquitylation of OMM proteins in H2030, H1437 and H1666. Mean ± s.e.m., *n* = 3 independent biological replicates. Ubiquitylation levels were normalized to total OMM protein amount. Unpaired two-tailed *t*-test. **p–r**, The cell viability of H2030 (**p**), H1437 (**q**) and H1666 (**r**) over 4 days with 0 μM, 4 μM and 10 μM MB-12 treatment. Two-way repeated ANOVA; the *P* value indicates the MB-12 treatment effect. Mean ± s.e.m., *n* = 3 independent biological replicates with 3 technical replicates each day each condition. **s**, A model showing the regulation of OMM protein degradation by the Leu–GCN-2–SEL1L axis. High levels of Leu inhibit GCN-2 and reduce SEL1L–HRD1-dependent degradation of OMM proteins, stabilizing key components of the protein import machinery such as TOMM40. This leads to mitochondrial proteome expansion for respiratory chain, TCA cycle, metabolism and metabolite transport, thereby enhancing mitochondrial respiratory capacity and cell viability.

thereby enhancing protein import, expanding the mitochondrial proteome particularly for proteins involved in metabolism and respiration, and boosting respiratory activity. Specifically, the SEL-1–SEL-11/SEL1L–HRD1 E3 ubiquitin ligase complex is required for OMM protein degradation, and the SEL-1/SEL1L level is regulated by Leu acting at the OMM for adjusting mitochondrial proteostasis and respiration. Together, these findings connect nutrient sensing and mitochondrial protein turnover with organismal health.

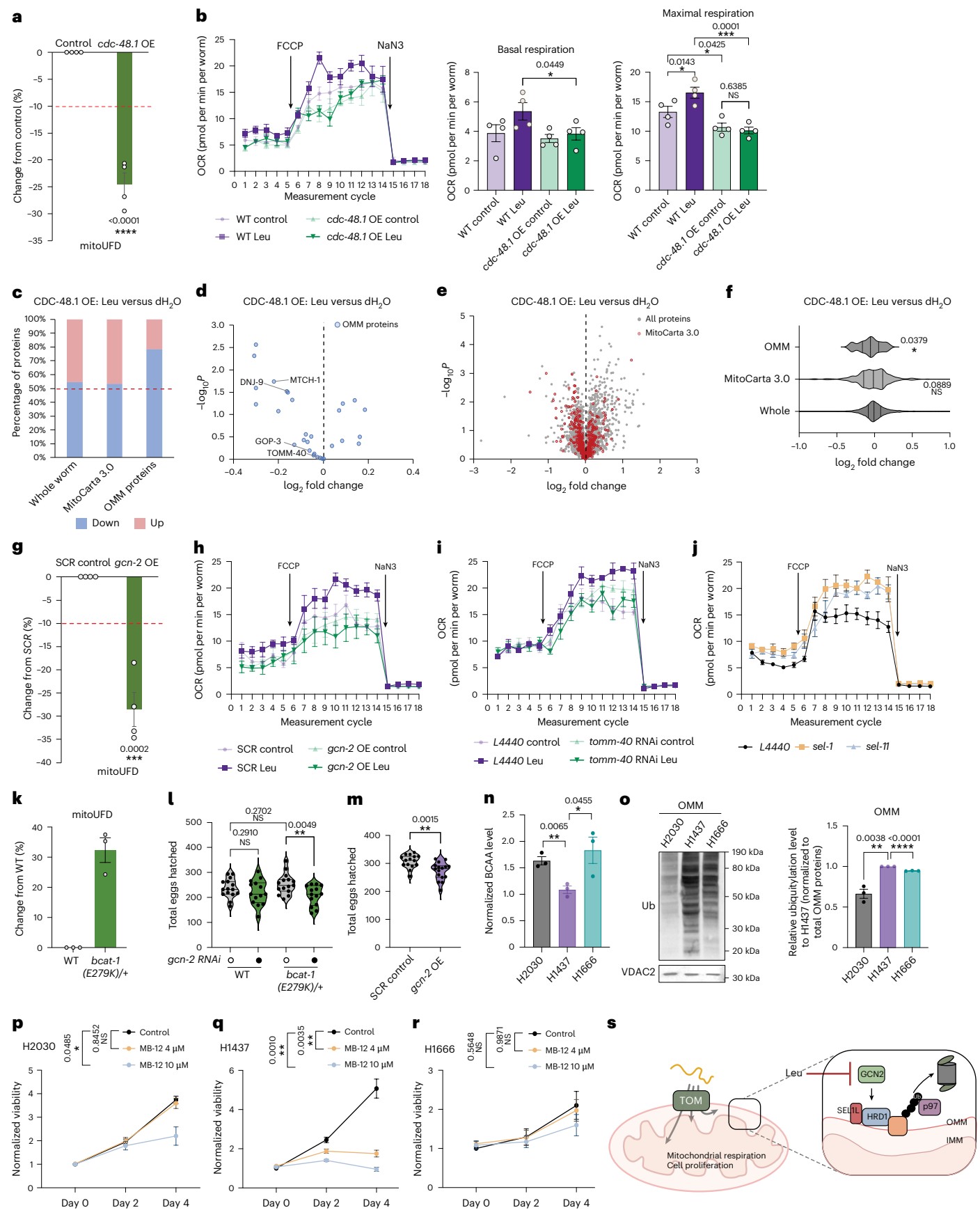

Previous studies have suggested a crosstalk between amino acid metabolism and mitochondrial homeostasis. For example, when impairment occurs to lysosomal function, which is important to import and sequester amino acids to avoid amino acid toxicity, cells can generate mitochondrial-derived compartments to remove the mitochondrial protein import receptor Tom70 and SLC25A carriers in response to amino acid overload stress[12]. Amino acid sensing also controls mitochondrial fusion to regulate mitophagy and apoptosis[66]. Long-term Leu exposure, for example, 48 h treatment, increases mitochondrial biogenesis and respiration in C2C12 muscle cell culture by activating SIRT1–AMPK signalling[67]. Instead of the conventional role of Leu in anabolism, our data uncover a distinct mechanism that acute Leu supplementation leads to a selective reduction in the ubiquitylation and degradation of OMM proteins, thereby remodelling the mitochondrial proteome independently of translation. This rewiring is indispensable for increasing the abundance of the import machinery and elevating mitochondrial respiration, potentially preparing cells for the bioenergetic demands of high nutrient availability (Fig. 5s).

The identification of the SEL-1–SEL-11/SEL1L–HRD1 complex acting specifically at the OMM and its modulation by Leu availability adds an important new dimension to our understanding of mitochondrial proteostasis. Recent studies have shown that the SEL1L–HRD1 complex binds to OMM proteins such as SAMM50, PGAM5 and VDAC1 and might regulate the stability of MTCH2 (ref. 68), implying a role for endoplasmic reticulum-associated E3 ligases and cofactors in OMM protein degradation. Furthermore, SEL1L–HRD1 has been shown to regulate mitochondrial dynamics by modulating endoplasmic reticulum–mitochondria contacts in brown adipocytes, which is essential for metabolic adaptation to cold challenge[69]. Other ERAD components, such as Cdc48/p97 and Ubx2, have also been implicated in mitochondrial protein quality control[1–3,70–72], and there is evidence that Ubx2 localizes to both the endoplasmic reticulum and OMM[7]. Although we find that SEL-1 is membrane bound and associated with mitochondria, it lacks an obvious mitochondrial targeting sequence. Therefore, further investigation is required to determine how this E3 ligase complex is targeted to the OMM, whether endoplasmic reticulum–mitochondria contact sites play a role in this process and how Leu specifically regulates mitochondria-associated SEL-1.

Leu serves as an important building block for protein synthesis, a nutrient signal and an energy source. Altered BCAA/Leu metabolism and mitochondrial changes have been implicated in various diseases. For example, lifelong inhibition of BCAA metabolism leads to reproductive ageing and is associated with mitochondrial dysfunction in C. elegans[53]. Leu is also a key factor driving tumour growth of aggressive rhabdomyosarcoma via modulation of mitochondrial metabolism and oxidative phosphorylation[65]. Our work suggests that BCAA metabolism sustains degradation of OMM proteins and tunes mitochondrial activity, which is associated with fertility when encountering stress. By analysing lung cancer cells with different intracellular BCAA levels, we show that BCAAs are directly related to OMM proteostasis and cellular resistance to mitochondrial import inhibition, which may be an important aspect to consider for cancer treatment.

In conclusion, our study provides evidence that cytosolic nutrient status directly modulates OMM proteostasis as a rapid response to adjust mitochondrial activity. This reshaping of the OMM proteome enhances mitochondrial import and respiratory capacity, linking nutrient sensing to mitochondrial adaptation. Our findings provide new mechanistic insights into the metabolic regulation of the OMM proteome, which may link diet-induced changes in mitochondrial proteostasis and metabolism to diseases such as cancer and infertility.

## Online content

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

## Methods

### *C. elegans* maintenance and transgenic lines

*C. elegans* were grown at 20 °C and kept on nematode growth medium (NGM) plates seeded with *E. coli* OP50 bacteria as food source as previously described[73]. The strain PP4027: *unc-119(ed4)III; hhIs286[Peft-3::UbV::GFP::fis-1-TM::unc-54 3′UTR; unc-119 + ]II* was generated by ballistic gene transfer of a plasmid containing a *UbV::GFP::fis-1-TM* fusion gene under the ubiquitous promoter *eft-3*. The strains PP4032: *unc-119(ed4)III; hhEx203[Peft-3::GFP::fis-1-TM::unc-54 3′UTR; unc-119 + ]* and PP4035 *unc-119(ed4)III; hhIs287[Peft-3::GFP::fis-1-TM::unc-54 3′UTR; unc-119 + ]* were generated by ballistic gene transfer of a plasmid containing a *GFP::fis-1-TM* fusion gene under the ubiquitous promoter *eft-3*. PP4028: *unc-119(ed4)III; hhIs286[Peft-3::UbV::GFP::fis-1-TM::unc-54 3′UTR; unc-119 + ]II; foxSi75 [eft-3p::tomm-20::mKate2::HA::tbb-2 3′UTR] I* was generated by crossing PP4027: *unc-119(ed4)III; hhIs286[Peft-3::UbV::GFP::fis-1-TM::unc-54 3′UTR; unc-119 + ]II* with SJZ328: *foxSi75 [eft-3p::tomm-20::mKate2::HA::tbb-2 3′UTR]I*. PP4029: *unc-119(ed4)III; foxSi75[eft-3p::tomm-20::mKate2::HA::tbb-2 3′UTR]I; hhIs64[unc-119(+); sur-5::UbV-GFP]III* was generated by crossing SJZ328: *foxSi75 [eft-3p::tomm-20::mKate2::HA::tbb-2 3′UTR]I* with PP563: *unc-119(ed4); hhIs64[unc-119(+); sur-5::UbV-GFP]III*. PP4031: *cdc-48.1(tm544)II; unc-119(ed4)III; hhIs286[Peft-3::UbV::GFP::fis-1-TM::unc-54 3′UTR; unc-119 + ]II* was generated by crossing *cdc-48.1(tm544)II* with PP4027: *unc-119(ed4)III; hhIs286[Peft-3::UbV::GFP::fis-1-TM::unc-54 3′UTR; unc-119 + ]II*. PP4038: *hhIs286[Peft-3::UbV::GFP::fis-1-TM::unc-54 3′UTR; unc-119 + ]II; risIs33 [K03A1.5p::3xFLAG::SV40-NLS::dCas9::SV40-NLS::VP64::HA + unc-119(+)]* was generated by crossing MIR249: *risIs33 [K03A1.5p::3xFLAG::SV40-NLS::dCas9::SV40-NLS::VP64::HA + unc-119(+)]* with PP4027; PP4039: *hhIs287[Peft-3::GFP::fis-1-TM::unc-54 3′UTR; unc-119 + ]; risIs33 [K03A1.5p::3xFLAG::SV40-NLS::dCas9::SV40-NLS::VP64::HA + unc-119(+)]* was generated by crossing MIR249 with PP4035. PP4044: *unc-119(ed3); hhIs286[Peft-3::UbV::GFP::fis-1-TM::unc-54 3′UTR; unc-119 + ]II; qaIs7201[pcdc-48.1::FLAG::cdc-48.1, unc-119(+)]* was generated by crossing XA7201: *unc-119(ed3); qaIs7201[pcdc-48.1::FLAG::cdc-48.1,unc-119(+)]* with PP4027. The CRISPR–Cas9 strategy to generate the *bcat-1(E279K)* mutant strain PP4209: *+/szT1[lon-2(e678) umnIs39]I;bcat-1(E279K)/szT1[umnIs40]X* and the subsequent balanced strain PP4211: *hhIs286[Peft-3::UbV::GFP::fis-1-TM::unc-54 3′UTR; unc-119 + ]II; +/szT1[lon-2(e678) umnIs39]I; bcat-1(E279K)/szT1[umnIs40]X* were developed together with SunyBiotech and mutant isolation was performed by the company. All strains that were used in this study are listed in Supplementary Table 4.

### Human cells maintenance

HEK293 cells (Sigma-Aldrich, 85120602) were maintained in DMEM (Gibco, 10566-016) and 10% FBS (VWR, 89510-186). Lung cancer cell lines were cultured in RMPI with 10% FBS. H1437 (NCI-H1437 (H1437) ATCC CRL-5872) and H1666 (NCI-H1666 (H1666) ATCC CRL-5885) were purchased directly from American Type Culture Collection (ATCC). H2030 (NCI-H2030 (H2030) ATCC CRL-5914) was a kind gift from R. Jachimowicz's laboratory. All cells were routinely tested for mycoplasma contamination using a Mycoplasma PCR Detection kit (abmGood, G238). All cell lines tested negative for mycoplasma. For the generation of stable cell lines constitutively expressing mitoUFD, the human UbV–GFP–FIS1TM construct was cloned into the EGFP-N1 vector and transfected into HEK293 Flp-In T-REx cells (Invitrogen, R78007) using the transfection reagent FuGene HD (Promega, E2311) according to the manufacturer's guideline. Positive clones were selected using DMEM supplemented with 10% FCS, 500 mg ml⁻¹ penicillin–streptomycin and 500 μg ml⁻¹ geneticin (InvivoGen, ant-gn-1). Expression of constructs was verified by western blot analysis of cell lysates.

### mitoTracker staining for mitochondria

For mitochondrial staining, MitoTracker Deep Red FM (Thermo Fisher, M22426) was used. Specifically, 3.5 cm NGM plates were seeded with 100 μl OP50 2 days before the assay. A 1 mM stock solution was prepared on the day of assay and 50 μl of the 1 mM stock solution was evenly spread on top of the OP50 food lawn. The MitoTracker-treated plates were protected from light and the staining plate was left to dry for 30 min. Fifty worms on day 1 of adulthood were then transferred to the staining plate and incubated for 2 h in dark. The MitoTracker-stained worms were then transferred into M9 drops next to the food lawn of a fresh OP50-seeded plate. The worms were allowed to crawl onto the fresh OP50 food lawn and incubated for 2 h in the dark ready for imaging.

### Chase assay

The chase assay in worms was performed as previously described[57,74]. CHX plates were prepared by spreading 50 μl of 50 mg ml⁻¹ CHX (Sigma) in ethanol stock solution on a 3.5 cm Petri dish containing 5 ml NGM using a spatula. Ethanol was added to the plates as controls and all plates were incubated at room temperature for 30 min. Plates were then seeded with OP50 bacteria overnight at 37 °C. Then, 100 L4 stage animals (synchronized by timed egg laying) were transferred to NGM plates containing freshly prepared CHX or control plates for 0 h, 3 h, 6 h or 9 h. Worms were then collected and analysed by western blotting. For cell culture, 100 μg ml⁻¹ CHX was used.

### Western blotting

To collect worm samples, 50 animals were picked into 100 μl M9 buffer, left to settle down and then 75 μl of M9 was removed. Then 25 μl 2× SDS buffer was added, followed by boiling at 95 °C for 10 min, sonication and boiling again at 95 °C for 5 min. For each sample, 30 μl was loaded on Bis–Tris 4–12% polyacrilamide gels for electrophoresis. Proteins were transferred to Amersham Protran 0.1 NC nitrocellulose membranes (Cytiva) with a semi-dry blotting system (Bio-Rad, Trans-Blot Turbo) using NuPAGE transfer buffer (Thermo Fisher Scientific). Membranes were blocked with 5% milk (in PBST + 0.1% Tween 20) for 1 h at room temperature and incubated with the primary antibodies overnight at 4 °C with RotiBlock (Carl Roth). Secondary antibodies were incubated at room temperature for 2 h. Fluorescence detection was conducted with a Li-Cor Odyssey scanner. The following antibodies were used: GFP (Clontech, 632380, 1:5,000), tubulin (Abcam, ab52866, 1:5,000), VDAC2 (Proteintech, 11663-1-AP, 1:1,000), TOMM40 (Proteintech, 66658-1-lg, 1:2,000), MTCO1 (Abcam, ab14705, 1:2,000), SEL1L (Sigma-Aldrich, S3699, 1:1,000), SYVN1 (Cell Signalling, 14773, 1:2,000), CDC-48 (Custom antibody, Hoppe lab, 1:5,000), FLAG (Sigma-Aldrich, F7425, 1:2,500), ubiquitin (Sigma-Aldrich, 05-944, 1:3,500), SEL-1 (Custom antibody, Jarosch Lab, 1:8,000), donkey anti-rabbit (Li-Cor, 926-68073, 1:10,000), donkey anti-mouse (Li-Cor, 926-32212, 1:10,000), goat anti-mouse (Jackson ImmunoResearch Laboratories, AB2338503, 1:10,000) and goat anti-rabbit (Jackson ImmunoResearch Laboratories, AB2339150, 1:10,000).

### RNAi by feeding

RNAi by feeding was performed as previously described[75] with modifications. RNAi plates were supplemented with 100 μg ml⁻¹ ampicillin and 1 mM IPTG and stored at 4 °C for no more than 1 month. Bacterial strains (RNase-deficient *E. coli* HT115) expressing double-stranded RNA were obtained from either Ahringer library[75] or Vidal library[76] and confirmed by sequencing. Bacteria were grown on LB plates with ampicillin and tetracycline at 37 °C overnight. Bacterial liquid culture with 100 μg ml⁻¹ ampicillin was grown at 37 °C with shaking at 150 rpm for 6–8 h. The liquid culture was seeded on the plates and left to dry at room temperature for 2 days. Animals were synchronized by timed egg laying: young adults were picked onto OP50 bacteria-seeded plates to lay eggs for 4 h. After 2 days, L4 stage animals were picked onto corresponding RNAi plates and day 1 adults were imaged. This acute RNAi treatment was performed to avoid potential long-term adaptation to metabolic and mitochondrial homeostasis changes and to avoid potential developmental effects for many of the mitochondrial genes.

For BioSorter quantification, L4 stage animals were washed off from OP50 plates, washed 3× by M9, and then transferred to corresponding RNAi plates. Double RNAi knockdown was performed by mixing two liquid bacterial cultures with 1:1 ratio, and the mixture of L4440 with the gene of interest was used as control.

## BioSorter quantification of fluorescence

For BioSorter (Union Biometrica) quantification, synchronized day 1 adults by timed egg laying were used. The fluorescence values (GFP green) of 50–250 gated worms were collected from each independent biological replicate. Gating of day 1 adults was based on time of flight >1,000 and extinction >600. Three independent biological replicates were used for statistical analysis. A representative illustration of the gating strategy is shown in Extended Data Fig. 2b.

## Fluorescence microscopy

For confocal imaging in worms, L4 stage or day 1 adult hermaphrodites were immobilized with 25 mM levamisole on a 3% agarose pad. Worms were imaged using an LSM980 Airyscan 2 equipped with Plan-Apochromat 63×/1.4 oil DIC and ZEN Connect Modul (Carl Zeiss Microscopy GmbH). For whole-worm imaging, an Axiozoom V16 (Carl Zeiss Microscopy GmbH) equipped with Axiocam 506mono and Zeiss 2.3 software (Carl Zeiss Microscopy GmbH) was used. ImageJ was used for image processing and fluorescence intensity quantification. For confocal imaging, cells were plated on glass-bottom dishes 2 days before treatment and imaging.

## mitoUFD pulldown for proteomic analysis or TUBE agarose IP

Twenty thousand worms were prepared by bleaching each strain for each condition. Animals were washed three times with M9 and once with dH$_2$O. Then, 200 µl fractionation buffer (50 mM Tris pH 7.4, 150 mM NaCl, 1× EDTA-free PI cocktail (Roche) and 1 mM Pefabloc) was added to each sample, followed by 20 strokes with a Dounce homogenizer. The worm lysate was spun down at 200$g$ for 5 min and 800$g$ for 10 min at 4 °C, and the supernatant was transferred to a new tube. The supernatant was centrifuged at 12,000$g$ for 30 min at 4 °C to pellet the mitochondrial fraction. The mitochondria pellet was resuspended and solubilized in 500 µl fractionation buffer containing 1% (w/v) digitonin for 45 min at 4 °C. Insoluble material was removed by centrifugation at 16,000$g$ for 10 min at 4 °C. For mitoUFD pulldown, 50 µl GFP-Trap Magnetic Agarose (ChromoTek) was added to the supernatant. The samples were incubated at 4 °C while rotating at 10 rpm for 1 h. The sample was transferred on a magnet stand and washed with fractionation buffer containing 0.1% (w/v) digitonin, 0.5% NP40 and 0.1% Triton-100. A mild wash was performed with fractionation buffer containing 0.1% (w/v) digitonin. The pellet was resuspended in 25 µl fractionation buffer and 25 µl 10% SDS (in PBS) was added. The samples were incubated at 95 °C for 5 min and transferred to a new tube without beads. Samples were sonicated and further processed with a modified SP3 protocol[77] and analysed by the Cologne Excellence Cluster on Cellular Stress Responses in Aging-Associated Diseases (CECAD) Proteomics Facility. For immunoprecipitation (IP) with TUBE1 agarose (LifeSensors), fractionation buffer was supplemented with 10 mM NEM and 25 µl supernatant was used as a pre-IP control. The rest of the sample was incubated with TUBE agarose at 4 °C while rotating at 10 rpm overnight. Beads were collected by low-speed centrifugation (1,000$g$) at 4 °C for 5 min. Beads were washed with 1 ml fractionation buffer containing 0.1% (w/v) digitonin, collected by low-speed centrifugation and the supernatant aspirated, leaving a small volume cushion to avoid disturbing the beads (2×). The pellet was resuspended in 25 µl fractionation buffer for western blot analysis.

## Proteasome activity assay

Proteasome activity measurement was performed as previously described[78] with modification. Cells were seeded on a 6-well plates 2 days before the measurement. On the day of measurement, cells were treated with 1 mM Leu or dH$_2$O control for 3 h. Cells were collected with ice-cold PBS and washed twice. For whole-cell samples, 200 µl proteasome lysis buffer (50 mM Tris–HCl pH 7.5, 250 mM sucrose, 5 mM MgCl$_2$, 0.5 mM EDTA, freshly added 2 mM ATP and 1 mM DTT) was added, and the cells were lysed by passing ten times through 27 G syringe. After centrifuging at 10,000$g$ for 10 min at 4 °C, the supernatant was collected. For mitochondria samples, cells were resuspended in 200 µl MTiso buffer (3 mM HEPES pH 7.4, 210 mM mannitol, 70 mM sucrose, 0.2 mM EGTA and freshly added 1× EDTA-free protease inhibitor) and transferred into an ice-cold Dounce homogenizer. Fifty strokes were applied and the homogenate was piled up to 200 µl of 340 mM sucrose. After centrifugation at 500$g$ for 5 min at 4 °C, the supernatant was collected and followed by centrifugation at 10,000$g$ for 10 min at 4 °C. The pellet was resuspended in 200 µl proteasome lysis buffer. The following AMC substrates were used: Trypsin-like proteasome activity: Ac-Arg-Leu-Arg-AMC (Enzo, BWL-AW9785-0005); Chymotrypsin-like proteasome activities: Z-Gly-Gly-Leu-AMC (Enzo, BML-ZW8505-0005); and Caspase-like proteasome activities: Z-Leu-Leu-Glu-AMC (Enzo, BWL-ZW9345-0005)

## OMM protein isolation

Isolation of OMM proteins was performed as described previously with modification[79,80]. Cells were seeded 1 day before collection. For each sample, four 151 cm$^2$ dishes were seeded with with $5.7 \times 10^6$ cells per dish. On the day of collection, cells were treated with the corresponding reagents for 3 h. GCN-2 inhibitor (HY-100877) was purchased from MedChemExpress. Cells were collected and washed twice with PBS. Cells were resuspended in 5 ml MTiso buffer as described before (3 mM HEPES pH 7.4, 210 mM mannitol, 70 mM sucrose, 0.2 mM EGTA and 1× EDTA-free protease inhibitor). Cells were transferred into an ice-cold Dounce homogenizer followed by 50 strokes. Then 500 µl was removed as the whole-cell control and the homogenate was piled up in a 50 ml Falcon to an equal volume of 340 mM sucrose, centrifuged at 500$g$ at 4 °C for 5 min and the supernatant collected. The samples were then centrifuged at 4 °C for 10 min at 10,000$g$ to pellet mitochondria. The mitochondria pellet was resuspended with 500 µl MTiso buffer containing 1 mg ml$^{-1}$ digitonin and 10 mM NEM. The samples were mixed intensely in a thermo mixed at 4 °C, 400 rpm for 15 min and 500 µl MTiso buffer added to stop digitonin extraction followed by centrifugation at 4 °C, 10,000$g$ for 10 min to collect the supernatant, which contains solubilized OMM and intermembrane space proteins. Protein concentration was measured by the BCA assay (Pierce) and the same amount of proteins were loaded for each sample for western blot analysis.

## Fractionation assay

A fractionation assay was performed as previously described[79]. The cells were collected and washed twice with ice-cold PBS. Cells were resuspended with 500 µl MTiso buffer followed by 50 strokes with a Dounce homogenizer. The homogenate was piled up to an equal volume of 340 mM sucrose, centrifuged at 500$g$ at 4 °C for 5 min and the supernatant collected. The samples were centrifuged at 4 °C for 10 min at 10,000$g$ to pellet mitochondria. The supernatant was centrifuged at 15,000$g$ for 20 min and collected, then centrifuged at 100,000$g$ at 4 °C for 1 h to pellet the microsomes. Concentrations of mitochondria and microsomes were measured by the BCA assay.

## BTZ treatment in worms

BTZ stock was diluted in DMSO at 10 mM. Then 3.5 cm plates seeded with OP50 were supplemented with dH$_2$O-diluted BTZ or DMSO control to a final concentration of 5 µM and 10 µM. The plate was left to dry for 2 h. L4 stage animals were transferred to BTZ or control plates for 6 h and immediately used for fluorescence imaging or western blotting.

## Leu supplementation

Leu stock solution was prepared at 100 mM diluted in dH$_2$O. Synchronized L4 stage animals were washed off from OP50-seeded plates to a 1.5 ml Eppendorf tube. Heat-killed OP50 was prepared as follows: OP50 culture was prepared by shaking at 37 °C for 6 h. The culture was then concentrated 20× and diluted in M9 followed by incubation at 75 °C for 90 min. The heat-killed OP50 was cooled down to room temperature before use. In a 100 µl liquid culture, 40 µl HK-OP50 and 3–5 µl 20× concentrated live OP50 was added. Then 100 mM Leu stock solution was added to reach the final concentration of 20 mM and 50 mM. dH$_2$O was used as a control. Worms were incubated with gentle shaking for 3 h before the assay. For the RNAi + Leu supplementation assay, animals were cultured on corresponding RNAi plates from the L2 to L4 stage, and Leu supplementation was performed on L4 stage animals. The corresponding RNAi bacteria culture was heat killed and used for the assay.

## Overexpression by feeding

Overexpression of genes of interest by bacterial feeding was performed according to a previous publication[81] with slight modifications. L4440_gcn-2_sgRNA were constructed using L4440_BioBrick-sgRNA as backbone. L4440_BioBrick-sgRNA was digested with BbsI (NEB) and gcn-2_gRNA_A fragments (annealed AGGGAAACAAGCGCCAAAAAGTGG and AAACCCACTTTTTGGCGCTTGTTT) were inserted using T4 ligation (NEB) (vector: insert of 1:10). L4440_gcn-2_gRNA_A was digested using BsaI HF-V2 (NEB) and the gcn-2_gRNA_B fragments (annealed AGGGAGAGGTTCCAACTAATCAAG and AAACCTTGATTAGTTGGAACCTCT) were inserted using T4 ligation (NEB). Both L4440_gcn-2_sgRNA and L4440_SCR_sgRNA (control) were transformed to HT115 bacteria for worm feeding. The feeding procedure was performed as described in 'RNAi by feeding'. Overexpression of gcn-2 was validated by qPCR.

## RNA extraction, cDNA synthesis and qPCR

Total RNA isolation was performed using TRIzol (Invitrogen). Age-synchronized worms were collected and washed twice with M9 buffer. Then 1 ml of TRIzol was then added to the worm pellet. Worms were frozen at −80 °C for overnight and then thawed at 37 °C. Zirconia beads were added for the Precellys 24-Dual cell homogenizer (Peq-Lab) disruption twice for 20 s at 6,000 rpm. Samples were incubated for 5 min at room temperature. Next, 300 µl 1-bromo-3-chloropropane was added followed by shaking for 15 s. Samples were then incubated at room temperature for 2–3 min and centrifuged for 15 min at 12,000g at 4 °C to separate the aqueous and organic phase. The aqueous phase was used to isolate total RNA with the RNeasy Mini kit (Qiagen) following the manufacturer's instructions. The quality and concentration of the isolated RNA was measured using a NanoDrop 8000 spectrophotometer (Thermo Fisher Scientific). A total of 1,000–2,000 ng of total RNA was used for cDNA synthesis with the High-Capacity cDNA Reverse Transcription kit (Applied Biosystems) following the manufacturer's instructions. qPCR was performed with Luna Universal qPCR Master Mix (New England Biolabs) and the Bio-Rad CFX96 Real-Time PCR Detection System. Three technical replicates were analysed per sample. gpd-1 was used as reference for normalization. The following primers were used: gcn-2 F: CAAATAGTACTTGACGAACGGGTA; gcn-2 R: CACCCAGACATGCCAATGAG; gpd-1 F: ATCACGTTGTTTCTAACGCATC; gpd-1 R: ATGAGTCCTTCGATGATACCG

## Proteomic analysis of whole-worm samples and enriched mitochondria

For proteomic analysis in C. elegans, 10k worms (N2) were used for each sample. Worms were synchronized by egg preparation with bleaching. Leu (final concentration of 20 mM) and/or CHX (final concentration of 500 µg ml$^{-1}$) were supplemented to L4 stage animals in HK-OP50 for 3 h with dH$_2$O as a control treatment. Worms were then washed 3× with M9 and 1× with ddH$_2$O. For the worm pellet, 200 ml fractionation buffer (50 mM Tris pH 7.4, and 150 mM NaCl; before use add 1× EDTA-free PI

cocktail and 1 mM Pefabloc) was added followed by 20 stokes with a Dounce homogenizer. Then 25 µl was removed as whole-worm samples and the remainder of the worm lysate was spun down at 200g for 5 min and 800g for 10 min at 4 °C. The supernatant was transferred to fresh Eppendorf tubes and centrifuged at 20,000g for 60 min at 4 °C. The pellet was resuspended in 25 µl fractionation buffer and 25 µl 2× SP3 buffer (10% SDS in 1× PBS). For whole-worm samples, 25 µl of 2× SP3 buffer was added to each sample. Both whole-worm and pellet samples were then incubated at 95 °C for 5 min followed by sonication. For all samples, DTT was added to a final concentration of 5 mM, vortexed and incubated at 55 °C for 30 min. Chloroacetamide was added to a final concentration of 40 mM, vortexed and incubated in the dark for 30 min at room temperature and then centrifuged for 10 min at 20,000g. The supernatant was transferred to a new tube when a pellet was visible. Samples were further processed with a modified SP3 protocol[77] and analysed by the CECAD Proteomics Facility on an Orbitrap Exploris 480 (granted by the German Research Foundation under INST 216/1163-1 FUGG) mass spectrometer coupled to a Vanquish neo in trap-and-elute set up (Thermo Scientific). The system was equipped with a FAIMSduo differential ion mobility device running at a compensation voltage of −50 V (Thermo Scientific) and an electrode temperatures of 99.5 °C (inner) and 85 °C (outer electrode). Samples were loaded onto a precolumn (Acclaim 5 µm PepMap 300 µ cartridge) with a flow of 60 µl min$^{-1}$ before being reverse-flushed onto an in-house packed analytical column (30 cm length, 75 µm inner diameter, filled with 2.7 µm Poroshell EC120 C18, Agilent). Peptides were chromatographically separated with an initial flow rate of 400 nl min$^{-1}$ and the following gradient: initial 2% B (0.1% formic acid in 80% acetonitrile), up to 6% in 4 min. Then, flow was reduced to 300 nl min$^{-1}$ and B increased to 20% B in 50 min, up to 35% B within 27 min and up to 95% solvent B within 1.0 min while again increasing the flow to 400 nl min$^{-1}$, followed by column wash with 95% solvent B and re-equilibration to initial conditions. MS1 scans were acquired from 399 $m/z$ to 1,001 $m/z$ at 15k resolution. Maximum injection time was set to 22 ms and the AGC target to 100%. MS2 scans ranged from 400 $m/z$ to 1,000 $m/z$ and were acquired at 15k resolution with a maximum injection time of 22 ms and an AGC target of 100%. DIA scans covering the precursor range from 400 to 1,000 $m/z$ were acquired in 60 × 10 $m/z$ windows with an overlap of 1 $m/z$. All scans were stored as centroid.

## Proteomic analysis of whole-cell samples and enriched mitochondria

For whole-cell quantitative label-free proteomics, HEK293 Flp-In T-Rex WT cells were seeded on 6-well plates. At 2 days after seeding, the cells were treated with 1 mM L-Leu and 6.67 µl ml$^{-1}$ ddH$_2$O, respectively. For CHX-chase samples, 100 µg ml$^{-1}$ CHX were added during the L-Leu/ddH$_2$O treatment. After incubating the cells with the respective treatments for 3 h, cells were washed with 1 ml PBS per well and collected in 0.7 ml PBS. Subsequently, cells were centrifuged for 7 min at 300g and 4 °C. The supernatant was removed and cell pellets were suspended in 50 µl lysis buffer per well (4% SDS in PBS supplemented with protease inhibitors). Cell lysates were sonicated and incubated at 96 °C for 5 min. Isolation of crude mitochondria from HEK293 cells was performed as previously described[82]. In short, cells were seeded on 15 cm dishes and cultivated for 3 days. Then, cells were treated with 1 mM L-Leu and 6.67 µl ml$^{-1}$ ddH$_2$O, respectively, and incubated for 3 h. For collection, the cells were washed with 10 ml ice-cold PBS per dish and collected in 10 ml ice-cold PBS. Subsequently, cells were centrifuged for 5 min at 500g and 4 °C. The supernatant was removed and cell pellets were suspended in 5 ml M buffer (220 mM mannitol, 70 mM sucrose, 5 mM HEPES−KOH and 1 mM EGTA−KOH, pH 7.4) containing cOmplete Protease Inhibitor Cocktail. Cells were homogenized using a precooled potter homogenizer (1,000 rpm, 15 strokes). The cell homogenate was centrifuged for 5 min at 600g and 4 °C. The supernatant was distributed to 2 ml reaction tubes and centrifuged again for 5 min at 600g

and 4 °C. Afterwards, the supernatant containing the crude mitochondria was centrifuged for 10 min at 8,000*g* and 4 °C. The pellet was washed with 2 ml ice-cold M buffer (without protease inhibitor cocktail) and centrifuged 10 min at 6,000*g* and 4 °C. Next, the supernatant was removed, and the pellet was suspended in 400 µl ice-cold M buffer (without protease inhibitor cocktail). Then, the protein content was determined using the BCA Reagent ROTI Quant assay according to the manufacturer´s instructions. Next, 100 µg mitochondria per replicate were added to fresh low-binding reaction tubes and centrifuged for 5 min at 10,000*g* and 4 °C. The supernatant was removed and the pellet containing crude mitochondria was suspended in 50 µl lysis buffer (4% SDS in PBS supplemented with protease inhibitors). Crude mitochondria lysates were sonicated and incubated at 96 °C for 5 min. From this point, whole-cell and mitochondrial samples were handled in parallel and treated using the same experimental procedure. First, lysates were subjected to an acetone precipitation: 200 µl ice-cold acetone were added and samples were stored at -80 °C overnight. The following day, samples were thawed on ice and centrifuged for 15 min at 16,000*g* and 4 °C. The supernatant was removed and the pellet was washed with 500 µl ice-cold acetone and subsequently air dried. Afterwards, pellets were solved in 50 µl 8 M urea in TEAB buffer supplemented with protease inhibitors and sonicated. The protein concentration of the samples was determined using Pierce 660 nm protein assay reagent following the manufacturer's instructions. Then, 50 or 30 µg (for whole-cell and mitochondrial samples, respectively) of each sample was transferred to a fresh low-binding reaction tube and the volume was adjusted to 40 µl by the addition of 8 M urea in TEAB buffer supplemented with protease inhibitors. Next, DTT was added to a final concentration of 5 mM and samples were incubated for 1 h at 25 °C. Then chloroacetamide was added to a final concentration of 40 mM and samples were incubated for 30 min in the dark at room temperature. Afterwards LysC protease was added in an enzyme to substrate ratio of 1:200 and samples were incubated for 4 h at 25 °C. Next, 160 µl of 50 mM TEAB buffer were added to reduce the urea concentration to below 2 M. After the addition of Trypsin in an enzyme to substrate ratio of 1:75, samples were incubated overnight at 25 °C. The next day, digestion was stopped by addition of formic acid to a final concentration of 1%. Finally, samples were centrifuged for 5 min at 20,000*g* and room temperature and loaded onto SDB-RP StageTips, which were previously equilibrated with 20 µl methanol, followed by 20 µl buffer B (0.1% formic acid in ddH$_2$O) and two times 20 µl buffer A (0.1% formic acid in 80% acetonitrile). Samples on StageTips were washed once with 40 µl buffer A and twice with 40 µl buffer B. Afterwards, the StageTips were dried and stored at 4 °C until measurement. Samples were analysed by the CECAD Proteomics Facility on an Orbitrap Exploris 480 (granted by the German Research Foundation under INST 216/1163-1 FUGG) mass spectrometer coupled to a Vanquish neo in trap-and-elute set up (Thermo Scientific). Project-specific gas-phase fractionation libraries[83] were generated by injecting individual pools stemming from the respective samples six times each covering the range of 400 to 1,000 *m/z* in 100 *m/z* steps running with an MS1 resolution of 60k and an MS2 resolution of 30k with staggered 4 *m/z* windows, resulting in effective 2 *m/z* windows after deconvolution using ProteoWizard software[84].

## Data processing for proteomic analysis

Samples were analysed in DIA-NN 1.8.1 (ref. [85]). The WormBase database (PRJNA13758.WBPS15, downloaded 24 January 2023) was used for *C. elegans* data analysis. The Uniprot canonical Human database (UP5640, downloaded 09 January 2024) was used for HEK293 cell data analysis. For all analysis, DIA-NN was run with the additional command line prompts '−report-lib-info'. Further output settings were filtered at 0.01 false discovery rate (FDR), N-terminal methionine excision enabled, maximum number of missed cleavages set to 1, min peptide length set to 7, max peptide length set to 30, min precursor *m/z* set

to 400, max precursor *m/z* set to 1,000, cysteine carbamidomethylation enabled as a fixed modification and heuristic protein inference activated. For the first analysis of Leu effect only in *C. elegans*, library building was performed with settings matching the acquisition parameters and the match-between-runs function enabled. Here, samples were directly used to refine the simulated FASTA-based library for a second search of the sample data. For the remaining datasets, the six gas-phase fractionation runs were first used to generate dedicated project libraries using identical settings as described above. Afterwards, samples were analysed against the corresponding library, again with identical settings as described above. Afterwards, DIA-NN output was further filtered on library *q* value and global *q* value ≤0.01 and at least two unique peptides per protein using R (4.1.3). Finally, label-free quantification values were calculated using the DIA-NN R package. Afterwards, analysis of results was performed in Perseus 1.6.15 (ref. [86]). For analyses from *C. elegans* samples, the resulting WormBase identifiers were translated into the UniProt identifier and these were used for term annotations and further analyses. GO term analysis was performed by GOrilla[87] with a ranked list of proteins according to *t*-test statistics. The mass spectrometry proteomics data have been deposited to the ProteomeXchange Consortium via the PRIDE[88] partner repository with the dataset identifiers PXD051398, PXD051401, PXD051403, PXD062734 and PXD062736.

## Seahorse O$_2$ consumption rate measurement

The oxygen consumption rate (OCR) was measured by a Seahorse XFe96 Analyser (Agilent) as previously described for worms and human cells[89,90] with modifications. Worm RNAi or overexpression feeding were performed at the L2 stage (synchronized by timed egg laying) for 24 h, followed by Leu supplementation for 3 h before the assay. We used a 2 min mix, 30 s wait, 2 min measure cycle protocol for all measurements. Fifteen worms were picked into each well and bacteria-only wells were used as control. Five basal measurements were taken before the injection of the mitochondrial uncoupler carbonyl cyanide-*p*-trifluoromethoxyphenylhydrazone (FCCP) (40 µM final concentration). After nine measurements, NaN$_3$ (50 mM final concentration) was injected to inhibit mitochondrial complex IV and V activity for a further four measurements. Six technical replicates and four biological replicates were measured for each condition. Data were normalized to worm number. For HEK293 cells, 3000 cells per well were seeded 1 day before the assay and cells were treated with corresponding amino acids for 3 h before the assay. We used 1.5 µM oligomycin, 8 µM FCCP and 0.5 µM rotenone + antimycin. For measurement, we used a 3 min mix, 3 min measure cycle protocol, with three measurement cycles for each step. Data were normalized to cell number.

## Immunogold labelling of frozen rehydrated *C. elegans*

Five adult worms were transferred into a flat carrier with a 100 µm recess (Leica, 1093), which was filled with 20% polyvinylpyrrolidone in ddH$_2$O. The carrier was high-pressure frozen using an EMPact 2 (Leica) and stored in liquid nitrogen until freeze substitution. The samples were then incubated in a solution of 0.5% uranyl acetate, 2% ddH$_2$O and 0.5% glutaraldehyde in acetone at −90 °C for 8 h in the AFS2 (Leica). The temperature was then raised to −30 °C within 33 h, after which the fixative was exchanged for 0.5% glutaraldehyde and 2% ddH$_2$O in acetone, and the samples were kept for 2 h. The temperature was then raised to −20 °C within 1.5 h. Rehydration was started using a solution of 50% 0.1 M PHEM buffer and 0.5% glutaraldehyde in acetone, raising the temperature to 0 °C within 3 h. The samples were rehydrated using 75% and 90% 0.1 M PHEM in acetone with 0.5% glutaraldehyde for 15 min each at 0 °C. Then, 100% PHEM with 0.5% glutaraldehyde was added for 16 h at 0 °C. The sample was then washed with 0.1 M PHEM for 15 min at room temperature and incubated for 30 min with 0.1 M PHEM, 0.25% glycine, 0.025% azur II (Sigma-Aldrich, 861065-25 G) and 0.025% methylene blue (Sigma-Aldrich, M9140-25G). Individual

worms were placed on a layer of solidified 10% gelatin and covered with another layer of 10% gelatin. Small cubes containing the worms were cut and incubated overnight or longer in 2.3 M sucrose (Sigma-Aldrich) at 4 °C. The gelatin cubes were mounted onto aluminium pins (Leica, 16701950) for cryo-ultramicrotomy, then frozen at −115 °C inside the FC7 cryo-chamber (Leica), which was mounted to the UC6 ultramicrotome (Leica). Ultrathin cryo-sections (70 nm) were cut using a diamond knife (Diatome). The sections were picked up using wire loops in an ice-cold 1:2 mixture of 2% methylcellulose (Sigma-Aldrich) and 2.3 M sucrose and transferred onto 100 mesh formvar-coated copper grids. The gelatin was then removed by incubating the grids in PBS at 40 °C for 1 h and washing them three times with PBS. Free aldehyde groups were then inactivated with a 15 min wash in 0.05 M glycine in PBS. The grids were then incubated in a drop of blocking solution for protein A-gold (Aurion), after which they were washed three times with 0.1% BSA-C (Aurion) in PBS. The primary antibody was diluted 1:50 in 0.1% BSA-c in PBS and incubated for 90 min, followed by six rinses with 0.1% BSA-c in PBS. No primary antibody controls were performed in parallel. The antigen was detected by incubating for 90 min with protein A-gold (10 nm), diluted 1:20 in 0.1% BSA-c in PBS. After 5 min of fixation with 2% glutaraldehyde, the sections were washed in PBS and $H_2O$, then contrasted for 5 min with 0.4% uranyl acetate in 2% methylcellulose on ice. The sections were picked up with a wire loop, with the excess fluid drained by gently dragging the loop over Whatman filter paper. After air drying, the grids were removed from the loops using forceps. Electron micrographs were taken using a JEM-2100 Plus transmission electron microscope (JEOL) equipped with a OneView camera 4 K 32-bit (Gatan) and Digital Micrograph software (Gatan).

### Brood size assay
Brood size assay was performed as previously described[91] with modification. Individual L4 animals were transferred to each 3.5 cm NGM plates seeded with corresponding RNAi or overexpression bacteria. Animals were transferred to new plates every day until day 3 of adulthood. Hatched animals, unhatched eggs and unfertilized oocytes were counted for each plate after 2 days of animal transfer.

### Cell viability and BCAA-Glo assay
Cell viability assay was performed as previously described[92]. Three thousand cells were seeded per well in a 96-well plate in 200 ml medium on day 0. For H1666, 6,000 cells were seeded on day 0. For each condition each day, three wells were seeded as technical replicates. The DMSM (for HEK293) or RPMI (for lung cancer cells) medium contain the corresponding drugs indicated in each figure legends. Next, 40 µl Cell-Titer-Glo reagent (Promega) was added to each well and mixed briefly for 2 min. The plate was incubated at room temperature for 10 min and luminescence was measured with a plate reader at day 0 (5 h after plating), day 2 and day 4. For the BCAA measurement, the BCAA-Glo kit (Promega) was used. Cells were washed with cold PBS and resuspended at the concentration of $5 × 10^5$ cells ml$^{-1}$. Then 500 µl cells were lysed with 250 µl 0.6 N HCl, mixed well and incubated at room temperature for 5 min. Next, 250 µl neutralization buffer was added and 50 µl cell lysate was transferred to each well, followed by adding 50 µl freshly prepared BCAA detection reagent then mixed by shaking the plate for 1 min and incubated at room temperature for 1 h before measuring.

### Statistics and reproducibility
For ImageJ quantification of fluorescence intensity, at least three independent biological replicates were performed with six to eight animals in each replicate. For BioSorter quantification of fluorescence intensity, three independent biological replicates were performed with 50–250 animals in each replicate. For statistical analysis, the percentage change was calculated as (GFP intensity of experimental animals − GFP intensity of control animals)/GFP intensity of control animals. The dashed line at 10% was used to evaluate the changes of

substrate levels, in addition to the *P* value. Prism 10 was used for statistical analysis. Data distribution was assumed to be normal but this was not formally tested. For multiple comparison, one-way analysis of variance (ANOVA) or two-way ANOVA were performed, followed by post hoc test indicated under individual figures. For two-sample comparisons, an unpaired two-tailed *t*-test or Mann–Whitney *U* test was performed as indicated under the corresponding figures. Pathway analysis of amino acid genes was performed by WormFlux Pathway Enrichment Analysis (http://wormflux.umassmed.edu/WormPaths/pea.php)

$$\text{Stabilization score} = \frac{\text{AA 1} \sum_{i=\text{gene 1}}^{\text{gene } n} (\text{Percentage stabilized})\, i × 100 + \cdots\cdots +}{\text{AA} k \sum_{j=\text{gene 1}}^{\text{gene } m} (\text{Percentage stabilized})\, j × 100}$$

All representative images from fluorescence microscopy and western blot were performed three times independently. No statistical methods were used to pre-determine sample size. No data were excluded from the analyses. The experiments were not randomized. Data collection and analysis were not performed blind to the conditions of the experiments.

### Reporting summary
Further information on research design is available in the Nature Portfolio Reporting Summary linked to this article.

### Data availability
The mass spectrometry proteomics data have been deposited to the ProteomeXchange Consortium via the PRIDE partner repository with the dataset identifiers PXD051398, PXD051401, PXD051403, PXD062734 and PXD062736. Plasmids and *C. elegans* strains generated in this study will be distributed to researchers upon request. Source data are provided with this paper.

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

## Acknowledgements

We thank the Caenorhabditis Genetics Center (funded by NIH Office of Research Infrastructure Programs (no. P40 OD010440) and the National BioResource Project) for worm strains. We thank the CECAD imaging facility for confocal imaging (C. Jüngst) and immunogold-EM analysis (F. Gaedke and B. Martiny) (funding for instrumentation JEOL JEM-2100 Plus DFG-INST 216793-1 FUGG) and CECAD proteomics facility (supported by the large instrument grant INST 216/1163-1 FUGG by the German Research Foundation (DFG Großgeräteantrag) for the proteomics analysis. We thank G. Stellbrink and A. Seiler for technical support, S. Efstathiou for advice on BioSorter fluorescence quantification, L. Müller for support on cellular fractionation experiments, A. Segref for suggestion on CHX chase assay, R. Jachimowicz for providing and suggestions on lung cancer cell lines, M. Krüger for suggestions on proteomic data analysis and A. Correns and S. Balmert for the preliminary test of the OMM TM domain. We thank A. Andersen, N. Aravindan, S. Efstathiou, A. Franz, C. Frezza, E. Jarosch, F. Ottens, N. Podvalnaya and M. Rapé for critical comments and helpful suggestions on the paper. We also thank all members of the Hoppe lab for helpful suggestions throughout this project. This work was funded by the Deutsche Forschungsgemeinschaft (DFG, German Research Foundation) under Germany's Excellence Strategy (EXC 2030–390661388, SFB 1218, project no. 269925409; SFB 1678, project no. 520471345; SPP 2453, project no. 541742459; RU 5762, project no. 531902955) and by the European Research Council (ERC, CellularPQCD, no. 101141579) to T.H. Views and opinions expressed are however those of the author(s) only and do not necessarily reflect those of the European Union or the European Research Council. Neither the European Union nor the granting authority can be held responsible for them. Q.L. is supported by a postdoctoral fellowship from the Alexander von Humboldt Foundation.

## Author contributions

Q.L. designed, performed and analysed the experiments. K.W. performed proteomic experiment and analysis in HEK293 cells and generated the human mitoUFD cell line. F.N. contributed to the establishment of CRISPR-based gene overexpression by feeding experiment. J.R. supervised the HEK293 proteomics and generation of the human mitoUFD cell line. T.H. supervised the experimental design and data interpretation. Q.L. and T.H. wrote the paper. All authors discussed the results and commented on the paper.

## Funding

## Competing interests

The authors declare no competing interests.

## Additional information

**Extended data** is available for this paper at https://doi.org/10.1038/s41556-025-01799-3.

**Correspondence and requests for materials** should be addressed to Thorsten Hoppe.

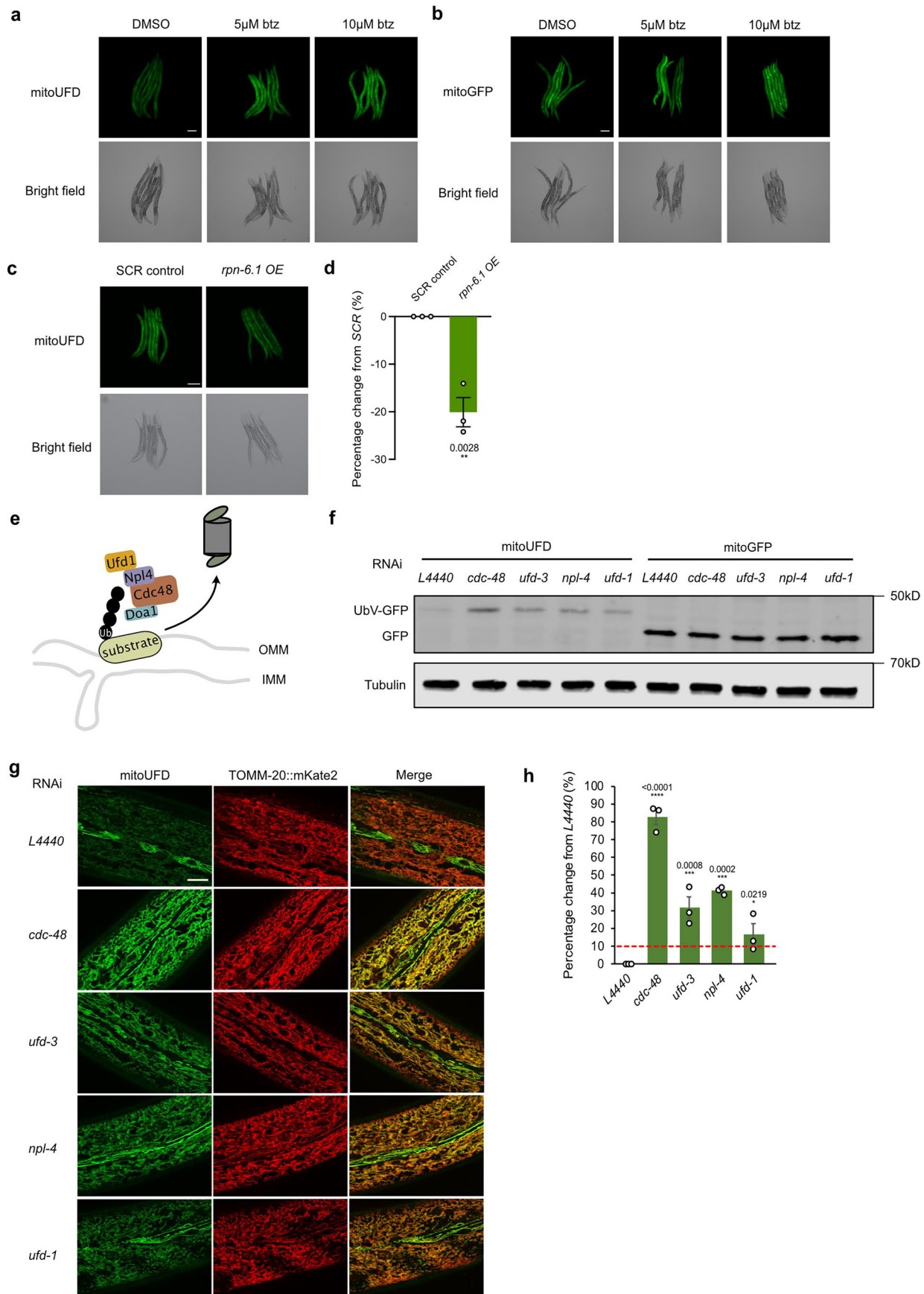

**Extended Data Fig. 1 | See next page for caption.**

**Extended Data Fig. 1 | Inhibition of UPS and CDC-48 complex impairs mitoUFD degradation. a-b.** Whole worm images showing that inhibition of proteasome by bortezomib (btz) treatment increases GFP level in worms expressing mitoUFD (a) but not mitoGFP (b). **c.** Whole worm images showing that *rpn-6.1* OE enhances mitoUFD degradation. **d.** ImageJ quantification showing that *rpn-6.1* OE enhances the degradation of mitoUFD. Mean ± SEM, n = 3 independent biological replicates with a of 18-24 animals. ImageJ quantification. Unpaired two-tailed t-test. **e.** Model of Cdc48 dependent OMM protein degradation by the 26S proteasome as described in yeast. **f.** Western blot analysis showing that

RNAi knockdown of *cdc-48* as well as its cofactors *ufd-3*, *npl-4* or *ufd-1* stabilizes mitoUFD but not mitoGFP. **g.** Confocal images showing that RNAi knockdown of *cdc-48* as well as its cofactors *ufd-3*, *npl-4*, or *ufd-1* stabilizes mitoUFD at OMM. Scale bar = 10 μm. **h.** Worm sorter quantification showing that RNAi knockdown of *cdc-48* as well as its cofactors *ufd-3*, *npl-4*, or *ufd-1* increases mitoUFD levels. Mean ± SEM, n = 3 independent biological replicates with 50-250 animals in each replicate. One-way ANOVA with Holm-Sidak correction for multiple comparisons with L4440 control. Source numerical data and unprocessed blots are available in Source data.

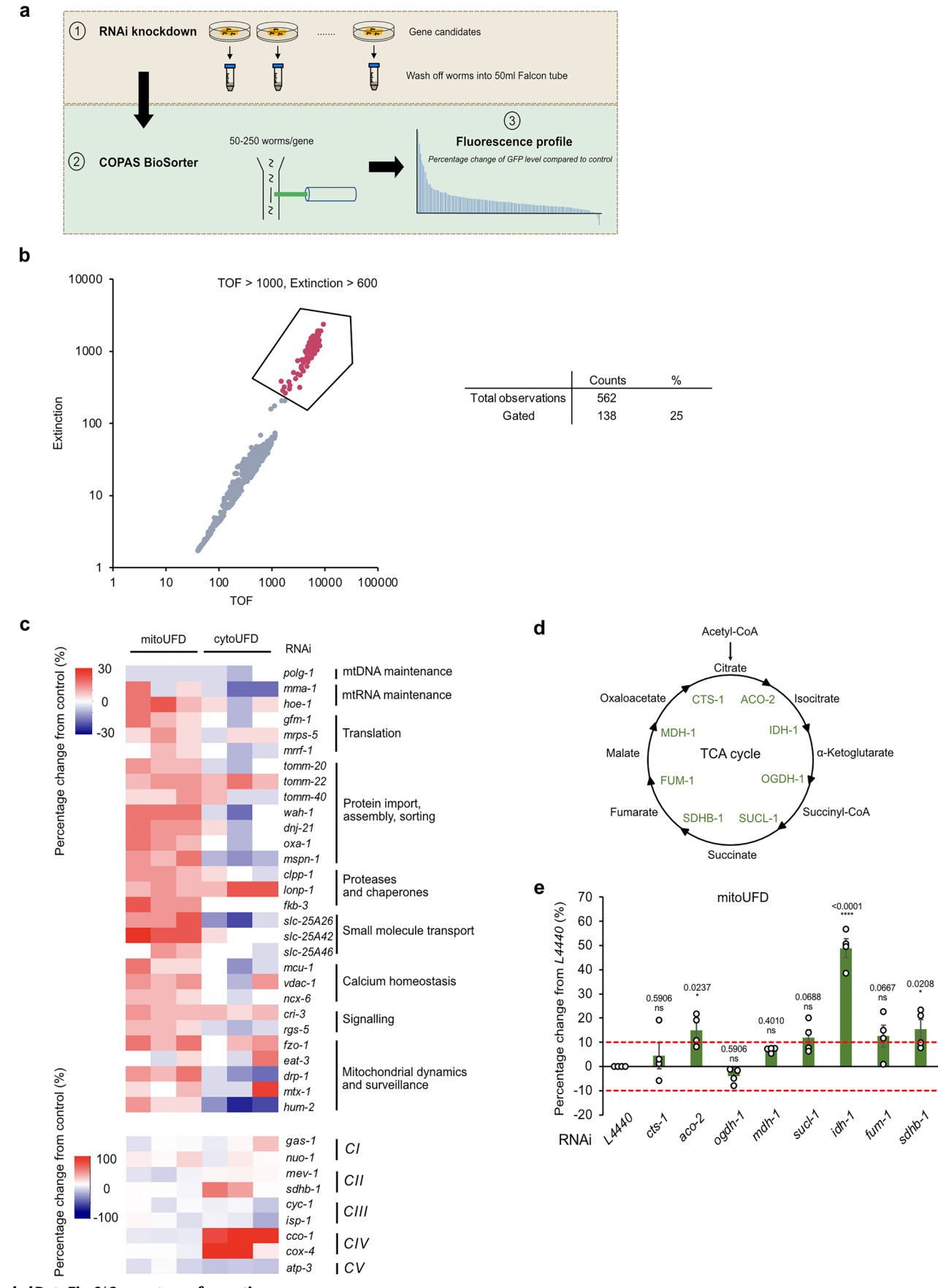

**Extended Data Fig. 2 | See next page for caption.**

**Extended Data Fig. 2 | TCA cycle affects OMM proteostasis. a**. Experimental procedure of BioSorter quantification of GFP levels in worms following RNAi knockdown of genes of interest. **b**. Representative profile of BioSorter quantification of whole worm fluorescence intensity. The red dots were gated for day 1 adults for fluorescence quantification based on TOF > 1000 and Extinction > 600. **c**. Changes in mitoUFD and cytoUFD levels after RNAi knockdown of mitochondrial genes quantified by BioSorter. n = 3 independent biological replicates with 50-250 animals in each replicate. **d**. TCA cycle and the *C. elegans* orthologues of key enzymes. **e**. RNAi knockdown of enzymes in TCA cycle in worms expressing mitoUFD. Mean ± sem, n = 4 independent biological replicates with total animal number of 24-32. ImageJ quantification. One way ANOVA with Holm-Sidak correction for multiple comparisons. Source numerical data are available in Source data.

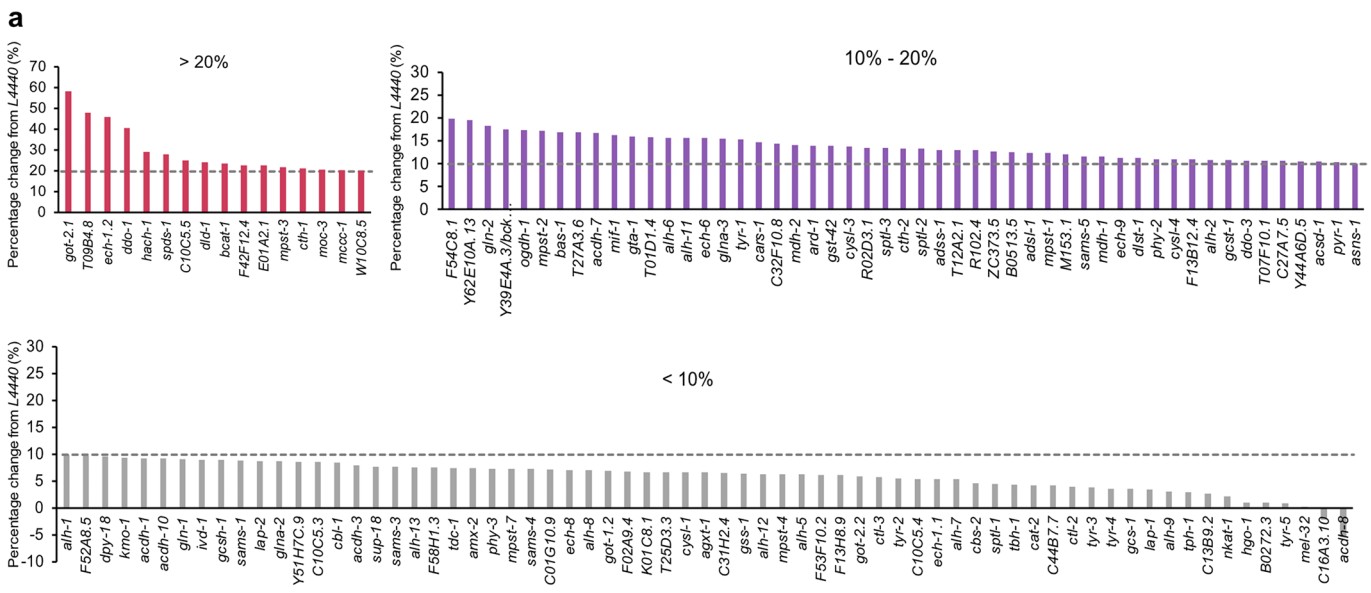

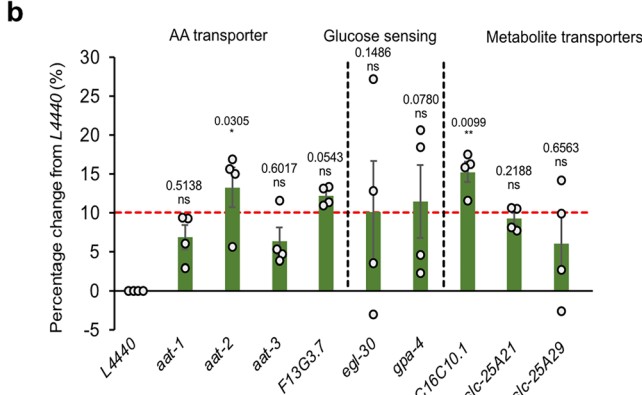

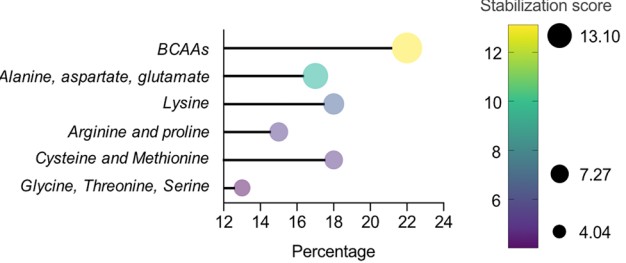

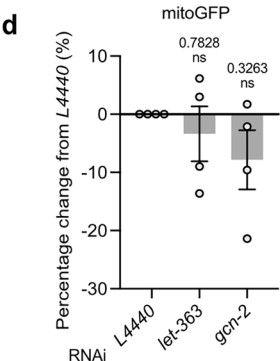

**Extended Data Fig. 3 | See next page for caption.**

**Extended Data Fig. 3 | RNAi screen for genes involved in amino acid metabolism, amino acid transport, glucose sensing and metabolite transport.** **a**. RNAi screen with the mitoUFD substrate for genes encoding enzymes for amino acid metabolism. Fluorescence in worms was quantified by worm sorter. 50-250 animals were measured for each gene. **b**. RNAi knockdown in the mitoUFD strain for genes involved in amino acid transportation, glucose sensing, and metabolite transportation at IMM. Mean ± sem, n = 4 independent biological replicates with total animal number of 24-32. ImageJ quantification. One way ANOVA with Dunnett correction for multiple comparisons. **c**. For the genes that stabilized mitoUFD of more than 20% in the targeted RNAi screen for 135 amino acid metabolic genes, the involved amino acids, ratio (hits/total genes per category) and their stabilization scores are shown based on pathway analysis. **d**. RNAi knockdown in the mitoGFP strain for genes involved in leucine sensing. Mean ± sem, n = 4 independent biological replicates with total animal number of 24-32. ImageJ quantification. One-way ANOVA with Dunnett correction for multiple comparisons with *L4440* RNAi control. Source numerical data are available in Source data.

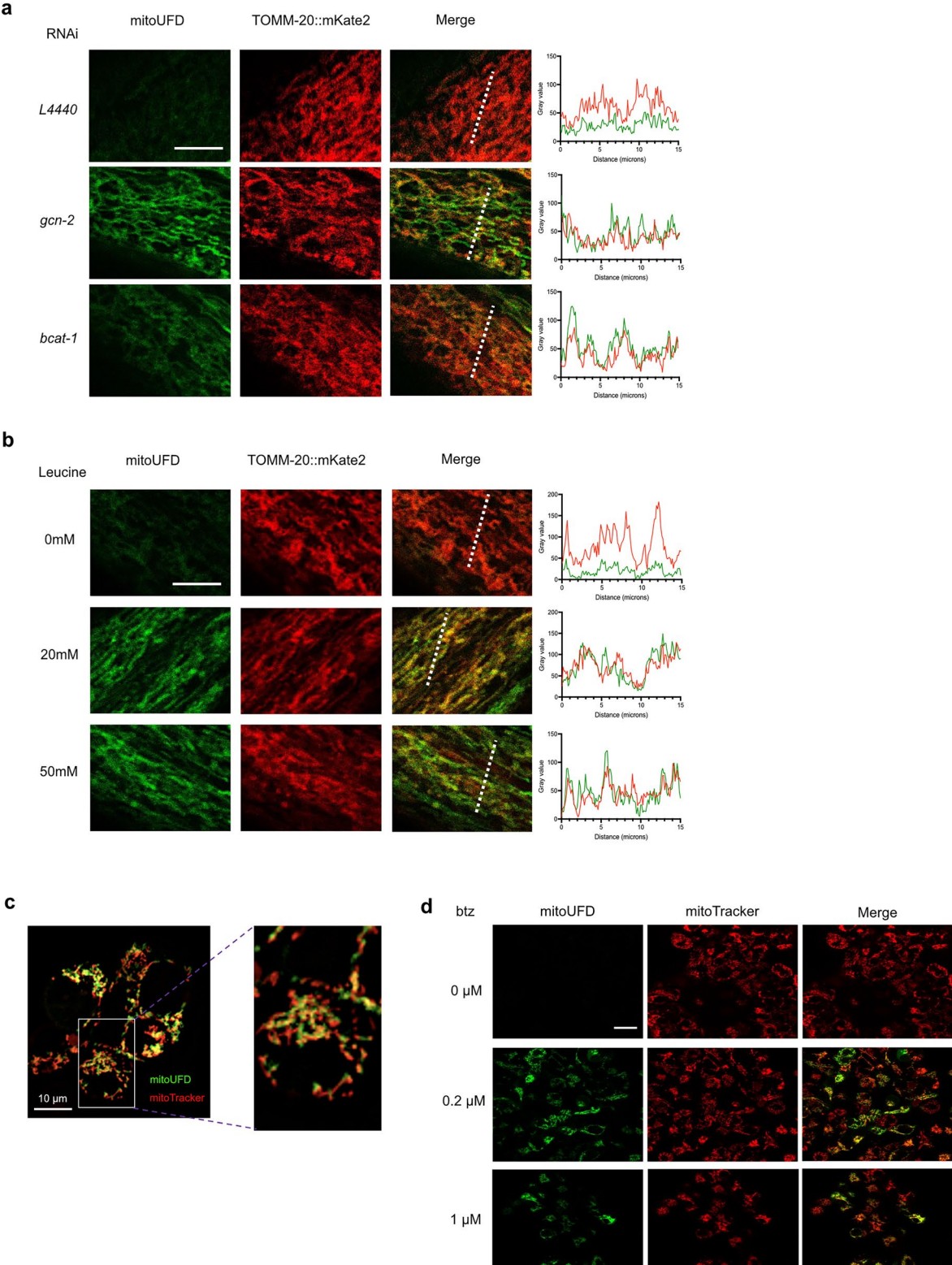

**Extended Data Fig. 4 | Leucine affects mitoUFD degradation. a**. Confocal imaging showing mitoUFD stabilisation upon RNAi knockdown of *gcn-2* and *bcat-1*. Scale bar = 10 μm. **b**. Confocal imaging showing mitoUFD stabilization after 3 h leucine treatment at 20 mM and 50 mM. Scale bar = 10 μm. **c**. Confocal imaging showing mitoUFD expression in HEK293 cells. mitoUFD in green and mitoTracker in red. Scale = 10 μm. **d**. Confocal imaging showing that bortezomib treatment at 0.2 μM and 1 μM stabilise mitoUFD.

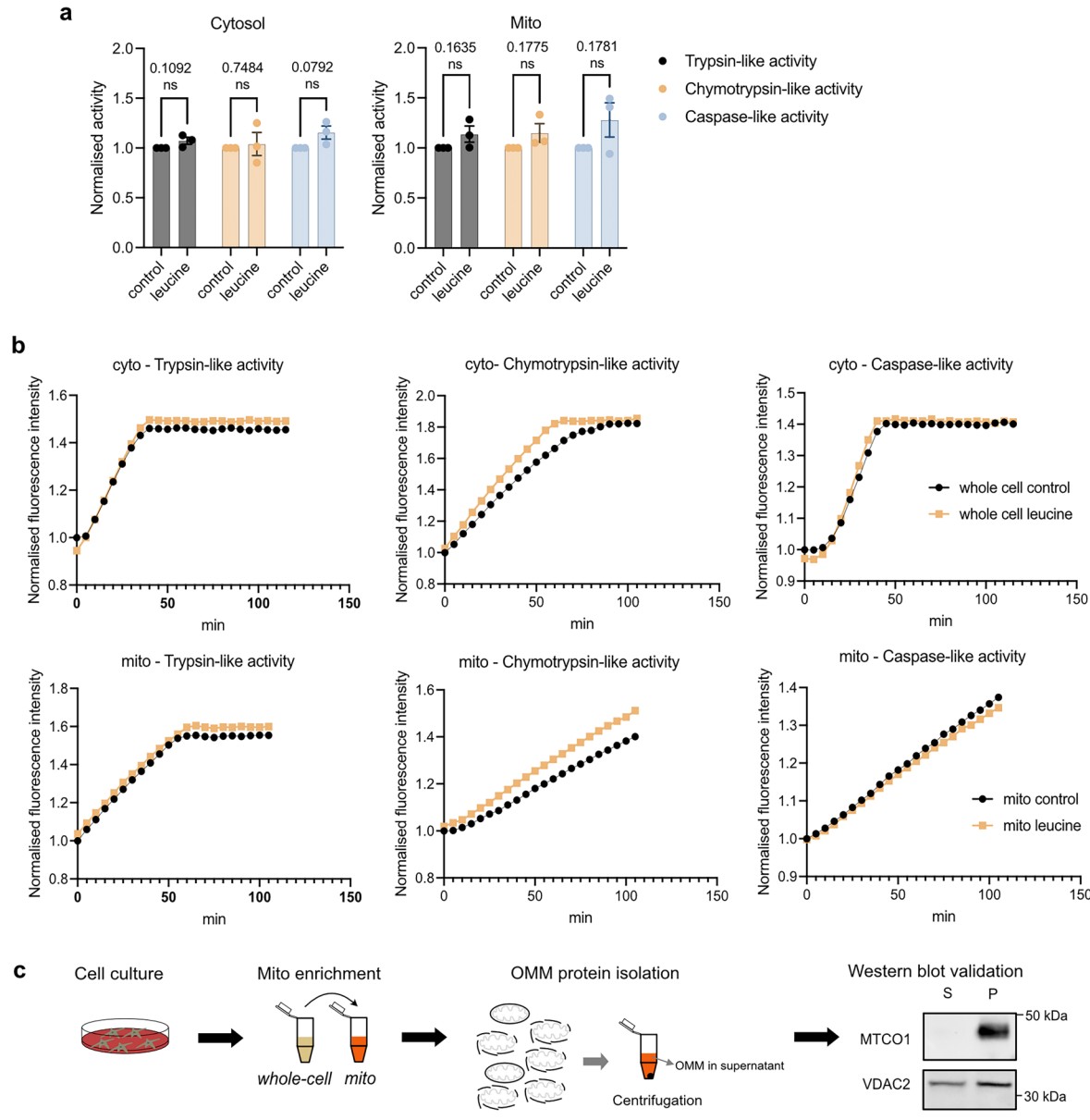

**Extended Data Fig. 5 | Proteasomal activity is not affected by leucine.**
**a-b**. Activity of cytosolic proteasome or proteasome associated with mitochondria. Mean ± sem, n = 3 independent biological replicates, unpaired two-tailed t-test. **c**. Experimental procedure of OMM protein isolation.

Mitochondria were enriched followed by OMM protein isolation. The isolated OMM proteins were confirmed by the presence of the OMM protein VDAC2 and the absence of IMM protein MTCO1. Source numerical data and unprocessed blots are available in Source data.

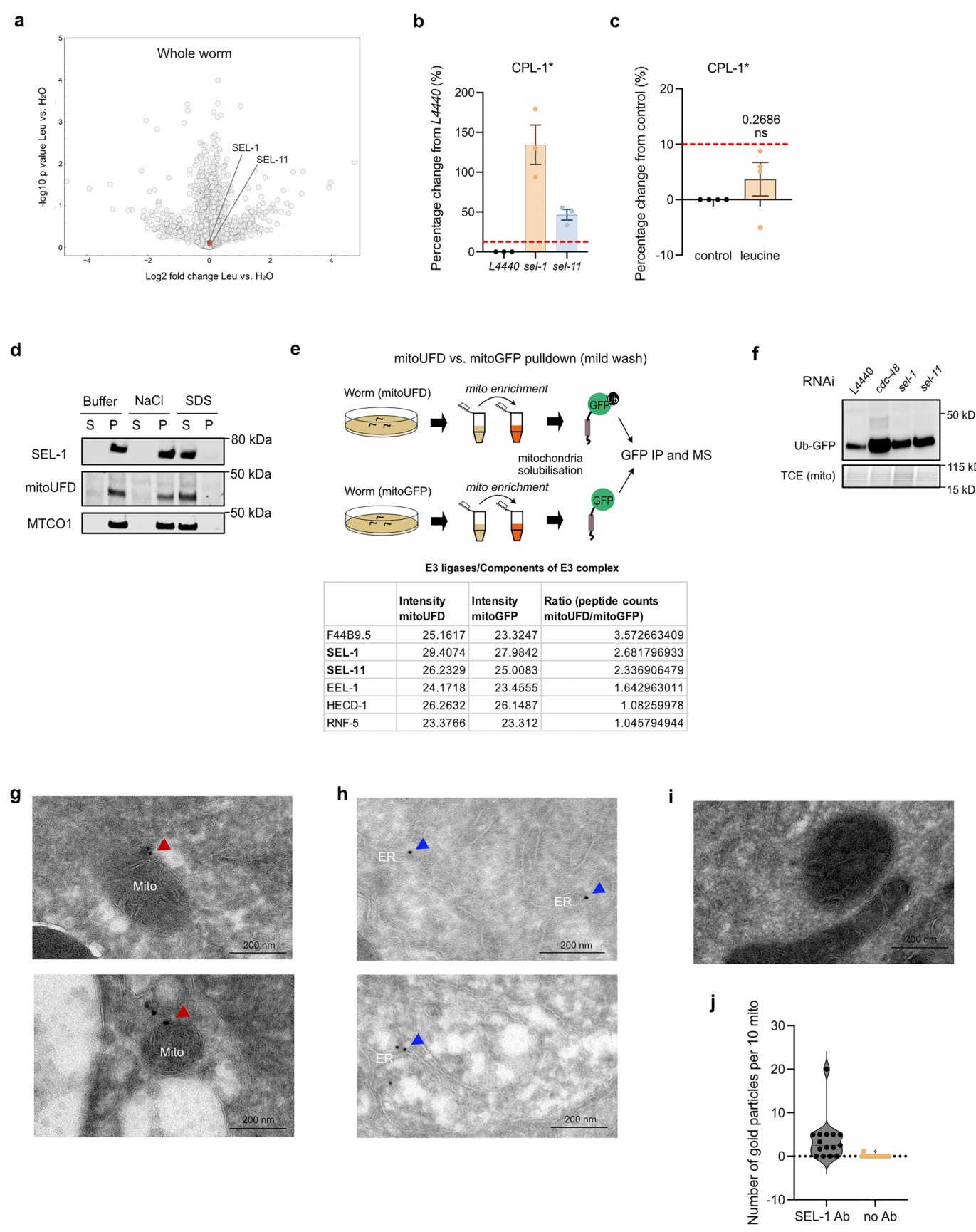

**Extended Data Fig. 6 | See next page for caption.**

**Extended Data Fig. 6 | Additional evidence supporting the role of SEL-1 in OMM protein degradation. a**. Volcano plots showing proteomic analysis of leucine effect on whole worm. Different from enriched mitochondria, SEL-1 protein level was not affected by leucine treatment in whole worm samples. P values were determined by unpaired two-tailed t-test. **b**. SEL-1 and SEL-11 mediated the degradation of ERAD substrate CPL-1*. Mean ± sem, n = 3 independent biological replicates with total animal number of 18-24. ImageJ quantification. **c**. Leucine supplementation did not alter the levels of CPL-1*. Mean ± sem, n = 4 independent biological replicates with total animal number of 24-32. ImageJ quantification. Unpaired two-tailed t test. ns, p > 0.05. **d**. Fractionation assay in worms showed that SEL-1 is membrane bound and co-fractionate with MTCO1 and mitoUFD. **e**. Pulldown of mitoUFD and mitoGFP (control) showed that SEL-1 and SEL-11 bind to mitoUFD and mitoGFP, but the amount of SEL-1 and SEL-11 that binds to mitoUFD is more than two-fold compared to mitoGFP. **f**. Western blot analysis of mitoUFD substrate with enriched mitochondrial from worms treated with *L4440, cdc-48, sel-1* and *sel-11* RNAi. **g-i**. Immunogold analysis showing OMM (g) and ER (h) localization of SEL-1 in worms. No specific staining was observed in control sections lacking primary antibody (i). **j**. Quantification of gold particles on the mitochondrial surface in SEL-1 antibody-treated and no-antibody control samples. Source numerical data and unprocessed blots are available in Source data.

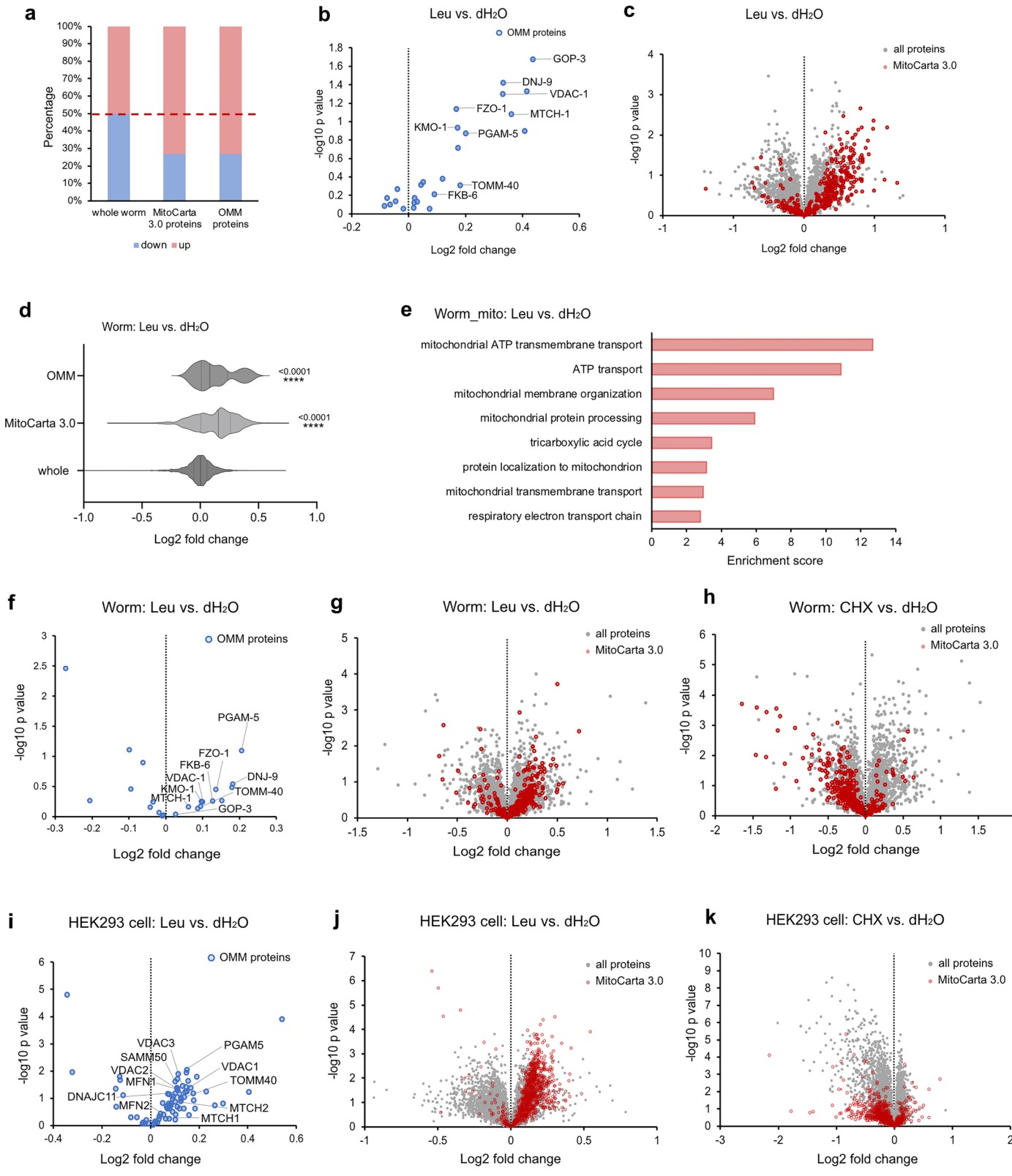

**Extended Data Fig. 7 | See next page for caption.**

**Extended Data Fig. 7 | Leucine effect on mitochondrial proteome.**
**a.** Percentage of proteins increased or decreased in whole worm proteome, MitoCarta 3.0 proteins, and OMM proteins upon leucine treatment. **b-c.** Volcano plots showing changes of OMM proteins (b) and whole worm and MitoCarta 3.0 proteins (c) upon leucine treatment. P values were determined by unpaired two-tailed t-test. OMM (b) and mitochondrial (c) proteins show the tendency to be more abundant in leucine-treated worms **d.** Log2 fold change (FC) of whole worm proteome, MitoCarta 3.0 proteins, and OMM proteins comparing Leu with dH$_2$O control treatment. Two-tailed Mann-Whitney U test between whole worm proteins and other protein groups. **e.** Enriched GO terms (p < 0.001 and FDR < 0.2 cutoff) for upregulated proteins of worm mito proteome upon leucine treatment. **f-g.** Volcano plot showing changes of OMM proteins (f) and whole worm and MitoCarta 3.0 proteins (g) upon leucine treatment. **h.** Volcano plot showing changes of whole worm and MitoCarta 3.0 proteins upon CHX treatment. **i-j.** Volcano plot showing changes of OMM proteins (i) and whole cell and MitoCarta 3.0 proteins (j) upon leucine treatment. **k.** Volcano plot showing changes of whole cell and MitoCarta 3.0 proteins upon CHX treatment. P values were determined by unpaired two-tailed t-test (f-k). Source numerical data are available in Source data.

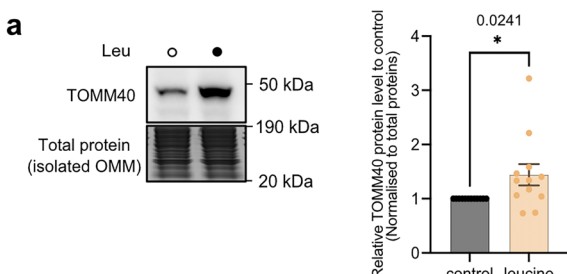

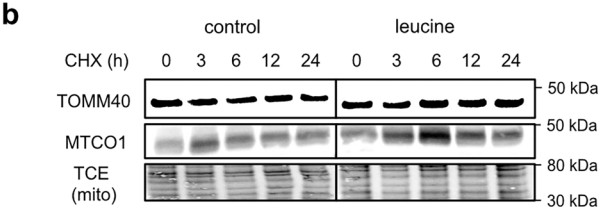

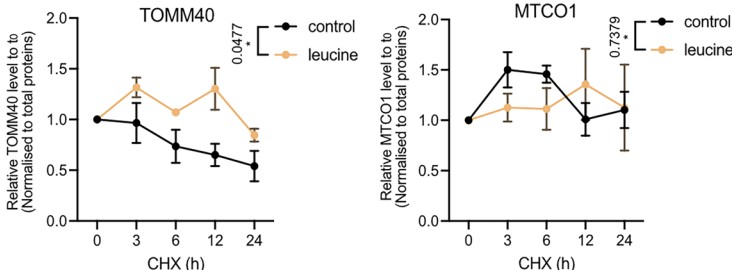

**Extended Data Fig. 8 | Validation of the OMM candidate TOMM40 from the proteomics. a**. Western blot for the OMM protein TOMM40 in HEK293 cells with and without 1 mM leucine treatment for 3 h. Mean ± sem, n = 12 independent biological replicates, unpaired two-tailed t test. **b**. CHX chase assay for the OMM protein TOMM40. The inner mitochondria membrane localised MTCO1 as control. Cells treated with 0 mM and 1 mM leucine were compared. Mean ± sem, n = 4 independent biological replicates, Two-way ANOVA, p value indicates leucine treatment effect. Source numerical data and unprocessed blots are available in Source data.

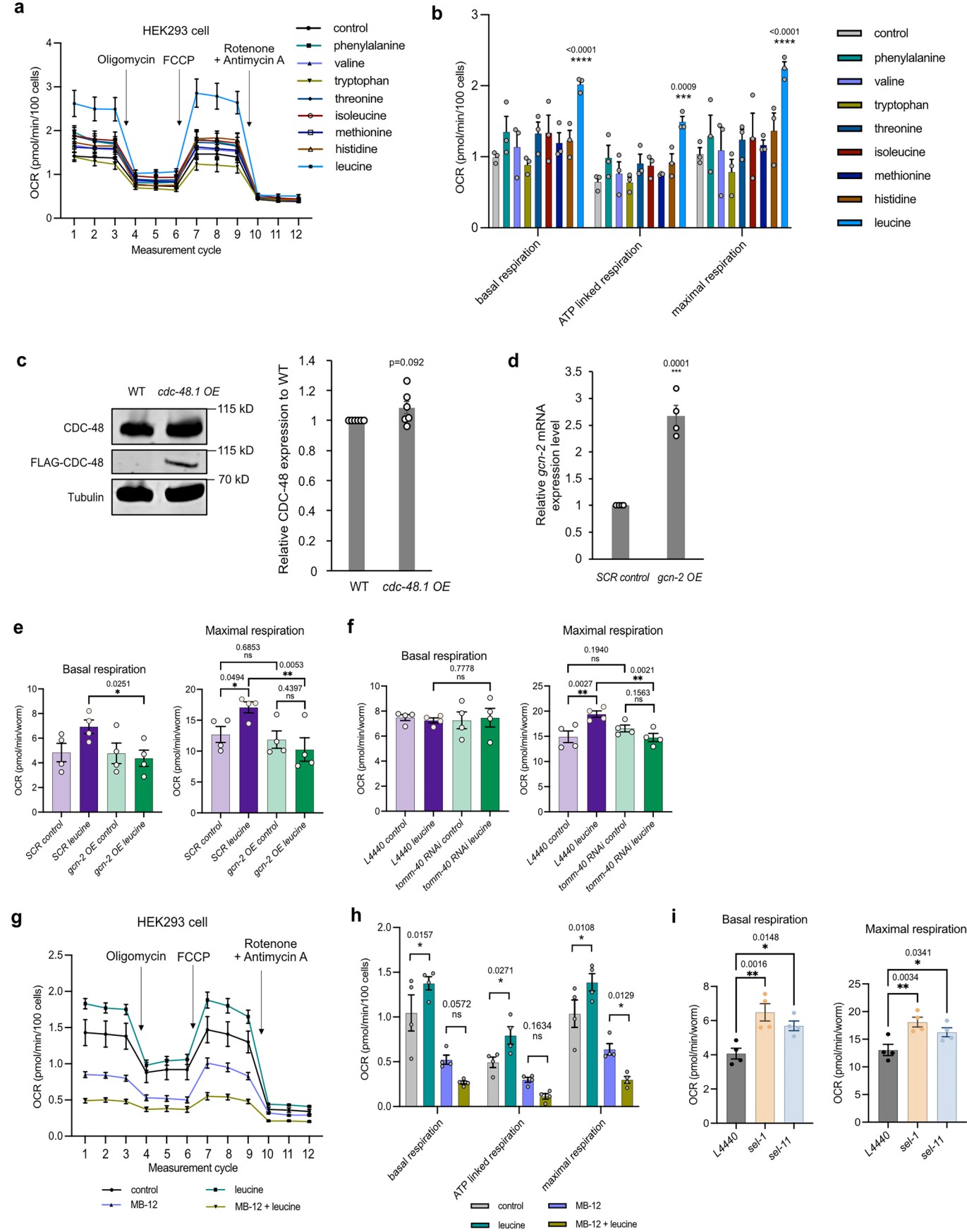

**Extended Data Fig. 9 | See next page for caption.**

**Extended Data Fig. 9 | Leucine enhances mitochondrial respiration via ubiquitin-dependent remodeling of OMM proteome by SEL-1-SEL-11 and mitochondrial protein import. a**. Average $O_2$ consumption rate by Seahorse analysis in HEK293 cells treated with essential amino acids at 1 mM for 3 hours. 3000 cells were seeded the day before assay. Mean ± sem, n = 3 independent biological replicates. **b**. Statistical analysis of basal respiration, ATP linked respiration and maximal respiration rates from (a). Two-way ANOVA with Fisher's LSD test for multiple comparisons. **c-d**. Validation of *cdc-48.1* (c) and *gcn-2* (d) overexpression by western blot and/or qPCR. Mean ± sem, unpaired two-tailed t-test. **e-f**. Statistical analysis for Fig. 5h, i. **g-h**. Seahorse analysis in HEK293 cells treated with leucine or MB-12. 3000 cells were seeded the day before assay. Mean ± sem, n = 4 independent biological replicates, two-way ANOVA with Fisher's LSD test for multiple comparisons. **i**. Statistical analysis for Fig. 5j. Source numerical data and unprocessed blots are available in Source data.

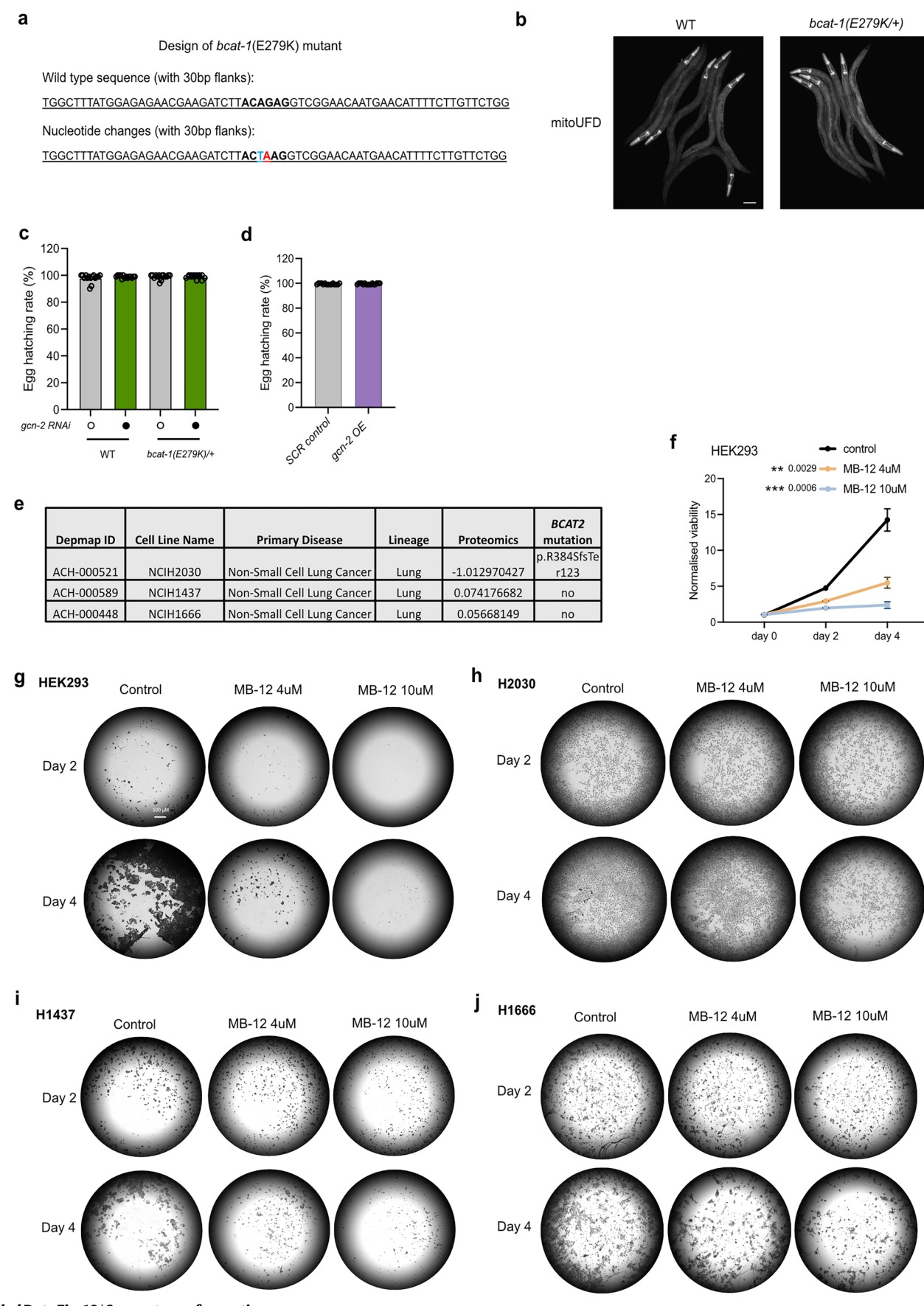

**Extended Data Fig. 10 | See next page for caption.**

**Extended Data Fig. 10 | Additional evidence supporting the link between BCAA catabolism and OMM proteostasis in fertility and cell proliferation under OMM perturbation. a**. Design of *bcat-1(E279K)* CRISPR editing strain. **b**. Whole worm imaging of mitoUFD in WT and *bcat-1(E279K)/+* worms. Scale bar = 10 μm. **c**. Egg hatch rate for Fig. 5l. n = 15 (WT_*L4440*), 14 (WT_*gcn-2* RNAi), 15 (*bcat-1/+_L4440*), 14 (*bcat-1/+_gcn-2* RNAi) animals. **d**. Egg hatch rate for Fig. 5m. n = 15 (SCR control), 16 (*gcn-2* OE) animals. **e**. Information of lung cancer cell lines from DepMap. **f**. Cell viability of HEK293 over four days with 0 μM, 4 μM and 10 μM MitoBlock-12 (MB-12) treatment. Two-way repeated ANOVA. Mean ± sem, n = 3 independent biological replicates with 3 technical replicates each day each condition. **g-j**. Cell images of HEK293 (g), H2030 (h), H1437 (i) and H1666 (j) at day 2 and day 4 after seeding. 3000 cells seeded on day 0 for HEK293, H2030 and H1437, 6000 cells seeded on day 0 for H1666. Source numerical data are available in Source data.

# Reporting Summary

## Statistics

For all statistical analyses, confirm that the following items are present in the figure legend, table legend, main text, or Methods section.

| n/a | Confirmed | |
|---|---|---|
| ☐ | ☒ | The exact sample size (*n*) for each experimental group/condition, given as a discrete number and unit of measurement |
| ☐ | ☒ | A statement on whether measurements were taken from distinct samples or whether the same sample was measured repeatedly |
| ☐ | ☒ | The statistical test(s) used AND whether they are one- or two-sided<br>*Only common tests should be described solely by name; describe more complex techniques in the Methods section.* |
| ☒ | ☐ | A description of all covariates tested |
| ☐ | ☒ | A description of any assumptions or corrections, such as tests of normality and adjustment for multiple comparisons |
| ☐ | ☒ | A full description of the statistical parameters including central tendency (e.g. means) or other basic estimates (e.g. regression coefficient) AND variation (e.g. standard deviation) or associated estimates of uncertainty (e.g. confidence intervals) |
| ☐ | ☒ | For null hypothesis testing, the test statistic (e.g. *F*, *t*, *r*) with confidence intervals, effect sizes, degrees of freedom and *P* value noted<br>*Give P values as exact values whenever suitable.* |
| ☒ | ☐ | For Bayesian analysis, information on the choice of priors and Markov chain Monte Carlo settings |
| ☒ | ☐ | For hierarchical and complex designs, identification of the appropriate level for tests and full reporting of outcomes |
| ☒ | ☐ | Estimates of effect sizes (e.g. Cohen's *d*, Pearson's *r*), indicating how they were calculated |

*Our web collection on statistics for biologists contains articles on many of the points above.*

## Software and code

Policy information about availability of computer code

| Data collection | Zeiss 2.3 blue edition<br>Image Studio v5.2.5 (LI-COR Biosciences)<br>FlowPilot-ProTM (v1)<br>Digital Micrograph (Gatan)<br>Seahorse Wave 2.6.1 |
|---|---|
| Data analysis | Image Studio v5.2.5 (LI-COR Biosciences)<br>Image Lab v6.1.0 build 7<br>ImageJ v2.1.0/1.53c<br>R3.6.2 and R Studio v1.2.5033<br>Prism v10.1.1<br>Perseus 2.0.10.0<br>Seahorse Wave 2.6.1 |

For manuscripts utilizing custom algorithms or software that are central to the research but not yet described in published literature, software must be made available to editors and reviewers. We strongly encourage code deposition in a community repository (e.g. GitHub). See the Nature Portfolio guidelines for submitting code & software for further information.

## Data

Policy information about availability of data

All manuscripts must include a data availability statement. This statement should provide the following information, where applicable:
- Accession codes, unique identifiers, or web links for publicly available datasets
- A description of any restrictions on data availability
- For clinical datasets or third party data, please ensure that the statement adheres to our policy

Source data of this study are available within the paper, its Supplementary Information, and public repository. The mass spectrometry proteomics data have been deposited to the ProteomeXchange Consortium via the PRIDE partner repository with the dataset identifiers PXD051398, PXD051401, PXD051403, PXD062734, PXD062736. Any other information related to the findings of this manuscript are available from the corresponding author upon request. Plasmids and C. elegans strains generated in this study will be distributed to researchers upon request.

## Research involving human participants, their data, or biological material

Policy information about studies with human participants or human data. See also policy information about sex, gender (identity/presentation), and sexual orientation and race, ethnicity and racism.

| | |
|---|---|
| Reporting on sex and gender | *Use the terms sex (biological attribute) and gender (shaped by social and cultural circumstances) carefully in order to avoid confusing both terms. Indicate if findings apply to only one sex or gender; describe whether sex and gender were considered in study design; whether sex and/or gender was determined based on self-reporting or assigned and methods used. Provide in the source data disaggregated sex and gender data, where this information has been collected, and if consent has been obtained for sharing of individual-level data; provide overall numbers in this Reporting Summary. Please state if this information has not been collected. Report sex- and gender-based analyses where performed, justify reasons for lack of sex- and gender-based analysis.* |
| Reporting on race, ethnicity, or other socially relevant groupings | *Please specify the socially constructed or socially relevant categorization variable(s) used in your manuscript and explain why they were used. Please note that such variables should not be used as proxies for other socially constructed/relevant variables (for example, race or ethnicity should not be used as a proxy for socioeconomic status). Provide clear definitions of the relevant terms used, how they were provided (by the participants/respondents, the researchers, or third parties), and the method(s) used to classify people into the different categories (e.g. self-report, census or administrative data, social media data, etc.) Please provide details about how you controlled for confounding variables in your analyses.* |
| Population characteristics | *Describe the covariate-relevant population characteristics of the human research participants (e.g. age, genotypic information, past and current diagnosis and treatment categories). If you filled out the behavioural & social sciences study design questions and have nothing to add here, write "See above."* |
| Recruitment | *Describe how participants were recruited. Outline any potential self-selection bias or other biases that may be present and how these are likely to impact results.* |
| Ethics oversight | *Identify the organization(s) that approved the study protocol.* |

Note that full information on the approval of the study protocol must also be provided in the manuscript.

# Field-specific reporting

Please select the one below that is the best fit for your research. If you are not sure, read the appropriate sections before making your selection.

☒ Life sciences  ☐ Behavioural & social sciences  ☐ Ecological, evolutionary & environmental sciences

For a reference copy of the document with all sections, see nature.com/documents/nr-reporting-summary-flat.pdf

# Life sciences study design

All studies must disclose on these points even when the disclosure is negative.

| | |
|---|---|
| Sample size | No statistical method was used to predetermine sample sizes. Sample size determination was done according to previous literature and standard C. elegans approaches. Exact sample sizes are stated in the figure legends and at least three biological replicates were performed for each experiment. |
| Data exclusions | No data was excluded from the analysis. |
| Replication | All experimental replications were indicated in the figure legends. For proteomic analysis, four biological replicates were performed for each condition each experiment. All other experiments were performed with at least three biological replicates. |
| Randomization | Randomization was not applicable or relevant in our study. Experiments were performed with worms or cells clearly grouped based on genotypes and treatments. |

| Blinding | The investigators were not blinded as this is not necessary in our study. All experiments in this study were automated quantified by software with defined settings. |

# Reporting for specific materials, systems and methods

We require information from authors about some types of materials, experimental systems and methods used in many studies. Here, indicate whether each material, system or method listed is relevant to your study. If you are not sure if a list item applies to your research, read the appropriate section before selecting a response.

## Materials & experimental systems

| n/a | Involved in the study |
|-----|----------------------|
| ☐ | ☒ Antibodies |
| ☐ | ☒ Eukaryotic cell lines |
| ☒ | ☐ Palaeontology and archaeology |
| ☐ | ☒ Animals and other organisms |
| ☒ | ☐ Clinical data |
| ☒ | ☐ Dual use research of concern |
| ☒ | ☐ Plants |

## Methods

| n/a | Involved in the study |
|-----|----------------------|
| ☒ | ☐ ChIP-seq |
| ☐ | ☒ Flow cytometry |
| ☒ | ☐ MRI-based neuroimaging |

## Antibodies

| Antibodies used | Living Colors A.v. GFP Monoclonal Antibody [JL-8] Clontech Cat. # 632380<br>Monoclonal alpha Tubulin antibody [EP1332Y] Abcam Cat. # ab52866; RRID: AB_869989<br>VDAC2 Polyclonal antibody Proteintech Cat. # 11663-1-AP<br>TOMM40 Monoclonal antibody Proteintech Cat. # 66658-1-Ig<br>Anti-MTCO1 antibody [1D6E1A8] Abcam Cat. # ab14705<br>Anti-SEL1L (N-terminal) antibody produced in rabbit Sigma-Aldrich Cat. # S3699<br>SYVN1 (D3O2A) Rabbit mAb Cell Signalling Cat. # 14773<br>Anti-CDC-48.1 Antibody raised in rabbits Hoppe-lab/Biogenes Berlin Custom antibody<br>Rabbit ANTI-FLAG polyclonal Sigma-Aldrich Cat. # F 7425<br>Anti-Ubiquitin Antibody, clone P4D1-A11 Sigma-Aldrich Cat. # 05-944<br>Anti-SEL-1 Antibody Jarosch-lab, Berlin Custom antibody<br>IRDye® 680RD Donkey anti-Rabbit IgG Secondary Antibody Li-Cor Cat. # 926-68073; RRID: AB_10954442<br>IRDye® 800CW Donkey anti-Mouse IgG Secondary Antibody Li-Cor Cat. # 926-32212; RRID: AB_621847<br>Peroxidase AffiniPure™ Goat Anti-Mouse IgG + IgM (H+L) Jackson ImmunoResearch Laboratories RRID: AB_2338503<br>Peroxidase AffiniPure™ Mouse Anti-Rabbit IgG (H+L) Jackson ImmunoResearch Laboratories RRID: AB_2339150 |
|-----------------|----|
| Validation | Anti-Ubiquitin Antibody, clone P4D1-A11 Sigma-Aldrich Cat. # 05-944: Routinely evaluated by immunoblot of human recombinant ubiquitin (Catalog # 12-558), and ubiquitinated proteins in acid extracts from HeLa cells. https://www.sigmaaldrich.com/DE/en/product/mm/05944?srsltid=AfmBOoouiDBheITfQQdOx4Qpe_KNPIpMTro5kRvpPNbq4GA7i3W1Xrlc. Dilution 1:3500<br>Rabbit ANTI-FLAG polyclonal Sigma-Aldrich Cat. # F 7425: The antibody recognizes the FLAG epitope located on FLAG-tagged fusion proteins at the N-terminus or C-terminus, applying dot blot, immunoblotting, immunoprecipitation and immunocytochemistry assays. https://www.sigmaaldrich.com/DE/en/product/sigma/f7425?srsltid=AfmBOorLqBO_BPwLkbHGhqD-UHHJJa6wYNmmt5vwdPjqCnz7aEB7IDG6. Dilution 1:2500<br>Anti-CDC-48.1 antibody: Franz et al., Mol.Cell., 2011, PMID: 21981920. Dilution 1:5000<br>Rabbit monoclonal alpha Tubulin antibody [EP1332Y]: Validated in IHC-P, WB, ICC/IF, Flow Cyt (Intra) and tested in Mouse, Rat, Pig, Human, Drosophila melanogaster samples. https://www.abcam.com/en-kr/products/primary-antibodies/alpha-tubulin-antibody-ep1332y-microtubule-marker-ab52866. Dilution 1:5000<br>Living Colors A.v. GFP Monoclonal Antibody [JL-8]: Validated by GFP positive and negative worms. Dilution 1:5000<br>Anti-SEL-1 Antibody Jarosch-lab, Berlin Custom antibody: Validated in previous publications and also with sel-1 mutants by western blot analysis. Dilution 1:8000<br>VDAC2 Polyclonal antibody Proteintech Cat. # 11663-1-AP: Positive WB detected in mouse brain tissue, human heart tissue, mouse heart tissue, rat brain tissue, rat heart tissue. https://www.ptglab.com/products/VDAC2-Antibody-11663-1-AP.htm?srsltid=AfmBOoq1sDWHKDPtAxDWXXI2o1bFS64QDEbTuqB81IL6ldxh4ZnjnSl5#product-information. Dilution 1:1000<br>TOMM40 Monoclonal antibody Proteintech Cat. # 66658-1-Ig: Positive WB detected in LNCaP cells, HeLa cells, HEK-293 cells, HepG2 cells, human brain tissue, pig brain tissue, Jurkat cells, HSC-T6 cells, PC-12 cells, NIH/3T3 cells, RAW264.7 cells, K-562 cells, rat brain tissue, mouse brain tissue. https://www.ptglab.com/products/TOMM40-Antibody-66658-1-Ig.htm?srsltid=AfmBOoqEk6yp-usG10-gXtjMXVVWRK6AJ01Xdrs2v3yHw3i8egcp4xWG. Dilution 1:2000<br>Anti-MTCO1 antibody [1D6E1A8] Abcam Cat. # ab14705: validated for use in Flow Cyt, ICC, IHC-P, WB in human, recombinant fragment samples. https://www.abcam.com/en-us/products/primary-antibodies/mtco1-antibody-1d6e1a8-ab14705?srsltid=AfmBOorQMKhyOAsSZXwdPIWeChiMCTcGuPXMMXflLu2_z9wsf1kovD9x#tab=datasheet. Dilution 1:2000<br>Anti-SEL1L (N-terminal) antibody produced in rabbit Sigma-Aldrich Cat. # S3699: validated in immunoblotting. immunoprecipitation, immunofluorescence. https://www.sigmaaldrich.com/DE/en/product/sigma/s3699?srsltid=AfmBOooG_AlTRKakLSP3L3ITqDI7_ymXpfdUABkDZaz___PJU6YRZBVZ. Dilution 1:1000<br>SYVN1 (D3O2A) Rabbit mAb Cell Signalling Cat. # 14773: This antibody has been validated by western blot analysis of extracts from various cell lines. https://www.cellsignal.com/products/primary-antibodies/syvn1-d3o2a-rabbit-mab/14773?srsltid=AfmBOoo5Rm1t7crOVb4m9aAegx1XePxc8NBA_2_XZrSl3HnvcGF-sqe7. Dilution 1:2000. |

# Eukaryotic cell lines

Policy information about cell lines and Sex and Gender in Research

| | |
|---|---|
| Cell line source(s) | Human: Flp-In T-REx-293-cell line Invitrogen Cat# R78007<br>Human: HEK293 Sigma-Aldrich Cat# 85120602<br>Human: NCI-H2030 [H2030] ATCC CRL-5914™<br>Human: NCI-H1437 [H1437] ATCC CRL-5872™<br>Human: NCI-H1666 [H1666] ATCC CRL-5885™ |
| Authentication | No additional authentication was performed. |
| Mycoplasma contamination | The cell line used in this study was confirmed negative for Mycoplasma contamination. |
| Commonly misidentified lines<br>(See ICLAC register) | No commonly misidentified cell line was used in this study. |

# Animals and other research organisms

Policy information about studies involving animals; ARRIVE guidelines recommended for reporting animal research, and Sex and Gender in Research

| | |
|---|---|
| Laboratory animals | Caenorhabditis elegans strains of different genotypes are listed in the Supplementary Table. Animal age is indicated in the Method section. |
| Wild animals | No wild animals were used in this study. |
| Reporting on sex | All experiments were performed with hermaphrodite C. elegans. |
| Field-collected samples | No field-collected samples in this study. |
| Ethics oversight | No ethical approval or guidance is required. |

Note that full information on the approval of the study protocol must also be provided in the manuscript.

# Plants

| | |
|---|---|
| Seed stocks | *Report on the source of all seed stocks or other plant material used. If applicable, state the seed stock centre and catalogue number. If plant specimens were collected from the field, describe the collection location, date and sampling procedures.* |
| Novel plant genotypes | *Describe the methods by which all novel plant genotypes were produced. This includes those generated by transgenic approaches, gene editing, chemical/radiation-based mutagenesis and hybridization. For transgenic lines, describe the transformation method, the number of independent lines analyzed and the generation upon which experiments were performed. For gene-edited lines, describe the editor used, the endogenous sequence targeted for editing, the targeting guide RNA sequence (if applicable) and how the editor was applied.* |
| Authentication | *Describe any authentication procedures for each seed stock used or novel genotype generated. Describe any experiments used to assess the effect of a mutation and, where applicable, how potential secondary effects (e.g. second site T-DNA insertions, mosiacism, off-target gene editing) were examined.* |

# Flow Cytometry

## Plots

Confirm that:

☒ The axis labels state the marker and fluorochrome used (e.g. CD4-FITC).

☒ The axis scales are clearly visible. Include numbers along axes only for bottom left plot of group (a 'group' is an analysis of identical markers).

☒ All plots are contour plots with outliers or pseudocolor plots.

☒ A numerical value for number of cells or percentage (with statistics) is provided.

## Methodology

| | |
|---|---|
| Sample preparation | Synchronized day 1 adults by timed-egg laying |
| Instrument | BioSorter (Union Biometrica) |
| Software | FlowPilot-ProTM (v1) |

Cell population abundance 50-250 gated worms were collected from each independent biological replicate

Gating strategy Gating of day 1 adults was based on TOF > 1000 and Extinction > 600.

☒ Tick this box to confirm that a figure exemplifying the gating strategy is provided in the Supplementary Information.

