## [Peer Review File · Nature Cell Biology]

Leucine inhibits degradation of outer mitochondrial membrane proteins to adapt mitochondrial respiration

Corresponding Author: Professor Thorsten Hoppe

Version 0:

Reviewer comments:

Reviewer #1

(Remarks to the Author)

In the manuscript by Li et al., the availability of leucine affects the turnover rates of the OMM proteins and thereby provides a protective mechanism for maintaining mitochondrial function. The work applies *C. elegans* as a main research model and genetic approaches combined with detecting fluorescently tagged uncleavable ubiquitin targeted to the OMM. This is a well-written study, and, in most cases, data support conclusions. However, many conclusions are based on insufficient experimental evidence (see Specific Comments below). Furthermore, while this work opens some new research avenues, it needs to propose a clear mechanism by which the OMM protein stability is affected by the OXPHOS activity, other aspects of mitochondrial homeostasis (fusion, fission, etc.), and isoleucine availability. The ample genetic evidence supports the role of leucine availability in controlling the expression of the mitochondrial model substrate UbV-GFP. However, many exciting observations need additional work to solidify each of these findings. The applicability of UbV-GFP for these studies should be further verified. The authors often state that they observe changes in UbV-GFP turnover but only analyze a steady state expression level under many distinct conditions. The turnover experiments were not included.

Furthermore, the ubiquitination status of this sensor (as well as the representative endogenous proteins) under many tested conditions should also be verified. Is this protein polyubiquitinated? Is its ubiquitination controlled by the OXPHOS activity and other pathways analyzed here?

While the study by Li et al. offers several insightful observations, it also raises significant concerns, particularly the lack of mechanistic depth in its current form. To enhance the study's suitability for publication, it is essential to conduct additional experiments that provide a more comprehensive understanding of the mechanisms underlying the observed effects.

Specific comments.

1. Figure 1. It is expected that in the presence of bortezomib or the absence of active proteasome, the mitochondrial substrate (UbV-GFP) would be ubiquitinated (Fig. 1d,f). Why do the authors show only one protein band? Is ubiquitination (higher molecular weight smear) undetectable in this system? The same applies to data showing CDC48-dependent extraction of UbV-GFP (Fig. 2). Please comment on this or add the ubiquitination data.
2. It is unclear whether HECD-1 acts directly on the mitochondrial UbV-GFP, or if this protein controls UbV-GFP degradation in the cytosol before inserting into the OMM. Many reports indicate that exogenous or abnormal mitochondrial proteins are degraded in the cytosol before their mitochondrial import. Please test this possibility. Generally, the direct versus indirect role of HECD-1 in mitochondrial proteostasis should be investigated in more detail.
3. Figure 3. The authors state that paraquat and rotenone affect UbV-GFP turnover. However, the data only show quantifications of steady-state protein levels. The authors should test the turnover rates of UbV-GFP in paraquat- and rotenone-treated cells. Furthermore, since paraquat is known to generate superoxide anion, verifying the degree to which antioxidative agents affect these changes would be interesting.
4. In Figure 4, it would be beneficial if the authors could provide original Western blot and imaging data for the most critical experiments.
5. Figure 5 of the manuscript reveals that leucine supplementation affects the expression levels of many mitochondrial proteins, including not only the OMM factors but also TCA and OXPHOS proteins. This finding suggests a potential role of leucine in mitochondrial biogenesis and somewhat challenges the specificity of the OMM protein turnover in this process. Could the authors comment on this possibility? Furthermore, including endogenous mitochondrial proteins as loading controls in Western blot experiments could address the total mitochondrial biogenesis versus specific changes in the OMM proteins.
6. Figure 5. Can the authors demonstrate that the stability of their target OMM proteins (e.g., Tom40 or VDAC1) is indeed affected by leucine treatment in worms and HEK cells? Such experiments are necessary to support the conclusion that leucine levels regulate the Ub-dependent degradation of these endogenous proteins. The endogenous protein cycloheximide chase can be easily done in mammalian cells for which antibodies are readily available. However, knowing the turnover rates of the UbV-GFP in worms would add additional depth.

Reviewer #2

(Remarks to the Author)

In this paper, the investigators develop a way to monitor outer mitochondrial membrane proteostasis by fusing a ubiquitinated GFP to the mitochondrial transmembrane domain of FIS-1 (mitoUFD) which allows them to monitor accumulation of mito-UFD as an indicator of decreased OMM protein turnover. They then use RNAi to screen the impact of knockdown of a range of proteins exploring different aspects of the regulation of OMM degradation and the impact of metabolism and different stressors. They eventually hone in on amino acid metabolism of which there was an enrichment in BCAA catabolism related genes in those genes that had the largest impact on mitoUFD accumulation when knocked down. Through a series of further experiments knockdown targeted at the mTOR pathway or *gcn-2*, they deduce that the impact of altered BCAA catabolism on mitoUFD accumulation may occur via GCN2 sensing of uncharged tRNA when leucine levels are low due to increased catabolism. To support this, they show that leucine supplementation phenocopied the *bcat-1* knockdown and any increase in mitoUFP with leucine is lost in *gcn-2* knockdown. They then perform mitochondrial proteomics in *C.elegans* and hek293 cells to identify which proteins change +/- leucine and +/- cyclohexamide, and find an increase in mitochondrial and OMM proteins with leucine supplementation that is unaffected by translational inhibitor cyclohexamide. Finally they find that leucine supplementation increases maximal respiration and this impact is eliminated in *gcn-2* or *cdc-48* overexpressors, or by *tomm-40* knockdown. Overall, the findings of this paper are really interesting but I think given the focus on leucine in the final few figures, title and abstract, some more amino acid controls in some of their experiments is required to see how specific some of the phenomena are to leucine versus other amino acids.

Main comments:

It is likely based on their RNAi screen of proteins related to amino acid metabolism, and the link to *gcn2* which is a general low amino acid sensor that increased levels of multiple other amino acids would have a similar impact and leucine may not be distinct in this capability. This should be explored at least at the level of testing whether supplementing other essential and non-essential amino acids impact mitoUFP accumulation.

The authors claim that leucine supplementation increases mitochondrial respiration by reducing OMM turnover as leucine supplementation to cells with overexpression of *gcn-2* or *cdc-48* which decrease mitoUFD do not have increased respiration in response to leucine. In addition they link this phenotype to mitochondrial protein import as *tomm-40* knockdown also prevents leucine induced increase in respiration.

Leucine can also impact respiration via its direct catabolism to acetyl-CoA. Is it possible that *gcn-2* OE or *tomm-40* kd also leads to decreased levels of BCAA catabolic enzymes in the mitochondria and that contributes to the decreased respiration? As a control, does another amino acid substrate such as glutamine increase respiration and is this compromised by *gcn-2* OE or *tomm-40* knockdown? Does knockdown of components of BCKDH such as ZK669.4 impact the leucine impact on respiration?

Reviewer #3

(Remarks to the Author)

The paper investigated how the regulation of outer mitochondrial membrane (OMM) proteostasis influences mitochondrial function. Through a fluorescent reporter assay and screening around 200 genes in *Caenorhabditis elegans*, researchers discovered that inhibiting the electron transport chain and the tricarboxylic acid cycle affected the degradation of OMM-anchored model substrates. Moreover, they observed that the loss of branched-chain amino acid transferase (*bcat-1*) and increased leucine levels inhibited the ubiquitin-dependent degradation of the model substrate. This led to a stabilization of OMM proteins, resulting in heightened mitochondrial protein content and improved respiration. The study suggests leucine is pivotal in connecting cellular nutrition with mitochondrial activity. While the genetic link between leucine and OMM protein levels is established, the precise mechanism connecting leucine and OMM protein turnover via ubiquitin-dependent pathways remains unclear, which limits enthusiasm for the manuscript in its current state.

Additionally, the paper described two pathways related to outer mitochondrial membrane (OMM) stability: mitochondrial function and leucine availability. However, it failed to establish a clear connection between these aspects. It would be pertinent to explore whether decreased mitochondrial function, induced by ETC and TCA RNAi, leads to elevated cellular leucine levels. If such an elevation occurs, it could potentially stabilize OMM proteins, as suggested by the proposed mechanism, thereby possibly prompting a compensatory increase in mitochondrial function. This hypothetical cascade might involve enhanced mitochondrial import efficiency and improved electron transport chain (ETC) activity. Nevertheless, the proposed sequence of events lacks robust experimental support and may oversimplify the intricate interactions among mitochondrial function, cellular metabolism, and protein turnover. Therefore, further experimentation is imperative to substantiate these hypotheses and elucidate the underlying mechanisms.

Main points:

1. The lack of RNAi efficiency testing throughout the study raises general concerns. This omission may impede the interpretation of certain negative results. For instance, Fig 1 and 2 employ *pdr-1* (*gk448*) and RNAi knockdown of other key mitophagy components to eliminate the effects of mitophagy and mitochondrial mass on mitoUFD level. However, the study overlooks examining mitochondrial protein content and RNAi efficiency to ensure that mitophagy levels and mitochondrial mass are indeed affected in these contexts.

2. In Fig 3, although the authors noted a decrease in mitoUFD level in response to paraquat, the outcomes of pharmacological and genetic inhibition of individual ETC complexes remain ambiguous. This uncertainty undermines the notion regarding the impact of mitochondrial respiration on mitoUFD turnover.
3. Whether supplementation of other amino acids, particularly the other two branched-chain amino acids (BCAAs), have a specific effect on mitoUFD levels? This aspect warrants further investigation to comprehensively understand the role of amino acid availability in regulating mitochondrial OMM protein turnover.
4. The proposed working model in Fig. 4j suggests that high levels of leucine reduce mitoUFD turnover by inactivating GCN-2. However, contradictory results are observed in Fig. 4h, where *gcn-2* RNAi + leucine, expected to further inactivate GCN-2, leads to lower mitoUFD levels compared to the L4440 control. This inconsistency raises questions regarding the proposed mechanism and warrants clarification.
5. The upregulation of most outer mitochondrial membrane (OMM) proteins mentioned in Fig 5 did not meet the commonly accepted 0.1 p-value cutoff typically set for omics studies. Moreover, no additional evidence from alternative methods was provided to confirm these changes. Even if leucine did reduce the degradation of endogenous OMM proteins, the lack of further verification regarding the involvement of UPS-mediated turnover limits the confidence in these findings.
6. In Fig 6, the authors noted that leucine supplementation increased both basal and maximal mitochondrial respiration, highlighting the necessity of intact TOM-mediated protein import. However, the study lacks direct evidence demonstrating that mitochondrial import efficiency was indeed elevated to transport specific target proteins.
7. Consequently, despite the inhibition of outer mitochondrial membrane (OMM) protein turnover by leucine, as shown in the previous section, it remains unclear how leucine exerts its effect on OMM protein levels, subsequently increasing mitochondrial import efficiency and respiration. The existing gaps between these conclusions and the lack of experimental support underscore the study's reliance on correlations rather than robust causal relationships.

Minor Concerns:

1. Line 68 states that "mitoUFD was ubiquitously expressed in all *C. elegans* tissue," but no corresponding result is presented to support this claim.
2. From Fig. 2, the authors primarily rely on the percentage change of BioSorter fluorescence intensity to indicate alterations in mitoUFD content. However, this measurement may be influenced by various factors such as mitochondrial mass. It may be beneficial to conduct further validations using fluorescence microscopy and Western Blot analysis.
3. The use of the term "crosstalk" in Line 94 might be overstated and potentially misleading, as it implies a relationship between two factors solely based on their mutual influence.
4. In Fig. 1e, f, and S1b, the "hecd-1" and "rpn-8" groups can be interchanged to align with their first mention in the text.
5. It is unclear which animal group serves as the control for calculating the percentage change in Fig. 3a.
6. The insets of Fig. 4b should include the Y-axis title to prevent confusion.
7. If applicable, additional p-values of non-significant groups can be displayed in the figures.
8. The Discussion section would benefit from a comprehensive evaluation of the study's limitations and suggestions for future research directions.

Decision Letter:

Dear Thorsten,

Thank you again for submitting your manuscript "Leucine increases mitochondrial respiration by reducing protein turnover at the outer mitochondrial membrane", to Nature Cell Biology and I am very sorry for the delay in communicating our decision to you. Your manuscript has now been seen by 3 referees, who are experts in mitochondria quality control (Referee #1); amino acid metabolism (Referee #2); and *C. elegans*, stress responses (Referee #3), and whose comments are pasted below. In light of their advice, we regret that we cannot offer to publish the study in Nature Cell Biology.

As you will see, although the reviewers found the work interesting, they raised serious concerns that question the strength of the data and of the novel conclusions that can be drawn at this stage. The reviewers were critical of the depth of mechanistic understanding and lack of sufficient investigation of other amino acids. We have discussed their concerns in detail editorially and find them valid and significant. Overall, we regrettably find that the dataset is too preliminary to pursue at the journal.

Although we cannot publish your paper, it may be appropriate for another journal in the Nature Portfolio. If you wish to explore the journals and transfer your manuscript please use our manuscript transfer portal. You will not have to re-supply manuscript metadata and files, unless you wish to make modifications. For more information, please see our [manuscript transfer FAQ](http://www.nature.com/authors/author_resources/transfer_manuscripts.html?WT.mc_id=EMI_NPG_1511_AUTHORTRANSF&WT.ec_id=AUTHOR) page.

We are very sorry that we could not be more positive on this occasion, but we thank you for the opportunity to consider this

work. We also hope that you find the reviewers' feedback below useful as you consider the next steps for the manuscript. Once again I also apologize for not being able to share our decision sooner.

With kind regards,

Reviewers' comments:

Reviewer #1 (Remarks to the Author):

In the manuscript by Li et al., the availability of leucine affects the turnover rates of the OMM proteins and thereby provides a protective mechanism for maintaining mitochondrial function. The work applies *C. elegans* as a main research model and genetic approaches combined with detecting fluorescently tagged uncleavable ubiquitin targeted to the OMM. This is a well-written study, and, in most cases, data support conclusions. However, many conclusions are based on insufficient experimental evidence (see Specific Comments below). Furthermore, while this work opens some new research avenues, it needs to propose a clear mechanism by which the OMM protein stability is affected by the OXPHOS activity, other aspects of mitochondrial homeostasis (fusion, fission, etc.), and isoleucine availability. The ample genetic evidence supports the role of leucine availability in controlling the expression of the mitochondrial model substrate UbV-GFP. However, many exciting observations need additional work to solidify each of these findings. The applicability of UbV-GFP for these studies should be further verified. The authors often state that they observe changes in UbV-GFP turnover but only analyze a steady state expression level under many distinct conditions. The turnover experiments were not included.

Furthermore, the ubiquitination status of this sensor (as well as the representative endogenous proteins) under many tested conditions should also be verified. Is this protein polyubiquitinated? Is its ubiquitination controlled by the OXPHOS activity and other pathways analyzed here?

While the study by Li et al. offers several insightful observations, it also raises significant concerns, particularly the lack of mechanistic depth in its current form. To enhance the study's suitability for publication, it is essential to conduct additional experiments that provide a more comprehensive understanding of the mechanisms underlying the observed effects.

Specific comments.

1. Figure 1. It is expected that in the presence of bortezomib or the absence of active proteasome, the mitochondrial substrate (UbV-GFP) would be ubiquitinated (Fig. 1d,f). Why do the authors show only one protein band? Is ubiquitination (higher molecular weight smear) undetectable in this system? The same applies to data showing CDC48-dependent extraction of UbV-GFP (Fig. 2). Please comment on this or add the ubiquitination data.
2. It is unclear whether HECD-1 acts directly on the mitochondrial UbV-GFP, or if this protein controls UbV-GFP degradation in the cytosol before inserting into the OMM. Many reports indicate that exogenous or abnormal mitochondrial proteins are degraded in the cytosol before their mitochondrial import. Please test this possibility. Generally, the direct versus indirect role of HECD-1 in mitochondrial proteostasis should be investigated in more detail.
3. Figure 3. The authors state that paraquat and rotenone affect UbV-GFP turnover. However, the data only show quantifications of steady-state protein levels. The authors should test the turnover rates of UbV-GFP in paraquat- and rotenone-treated cells. Furthermore, since paraquat is known to generate superoxide anion, verifying the degree to which antioxidative agents affect these changes would be interesting.
4. In Figure 4, it would be beneficial if the authors could provide original Western blot and imaging data for the most critical experiments.
5. Figure 5 of the manuscript reveals that leucine supplementation affects the expression levels of many mitochondrial proteins, including not only the OMM factors but also TCA and OXPHOS proteins. This finding suggests a potential role of leucine in mitochondrial biogenesis and somewhat challenges the specificity of the OMM protein turnover in this process. Could the authors comment on this possibility? Furthermore, including endogenous mitochondrial proteins as loading controls in Western blot experiments could address the total mitochondrial biogenesis versus specific changes in the OMM proteins.
6. Figure 5. Can the authors demonstrate that the stability of their target OMM proteins (e.g., Tom40 or VDAC1) is indeed affected by leucine treatment in worms and HEK cells? Such experiments are necessary to support the conclusion that leucine levels regulate the Ub-dependent degradation of these endogenous proteins. The endogenous protein cycloheximide chase can be easily done in mammalian cells for which antibodies are readily available. However, knowing the turnover rates of the UbV-GFP in worms would add additional depth.

Reviewer #2 (Remarks to the Author):

In this paper, the investigators develop a way to monitor outer mitochondrial membrane proteostasis by fusing a ubiquitinated GFP to the mitochondrial transmembrane domain of FIS-1 (mitoUFD) which allows them to monitor accumulation of mito-UFD as an indicator of decreased OMM protein turnover. They then use RNAi to screen the impact of knockdown of a range of proteins exploring different aspects of the regulation of OMM degradation and the impact of metabolism and different stressors. They eventually hone in on amino acid metabolism of which there was an enrichment in BCAA catabolism related genes in those genes that had the largest impact on mitoUFD accumulation when knocked down. Through a series of further experiments knockdown targeted at the mTOR pathway or *gcn-2*, they deduce that the impact of altered BCAA catabolism on mitoUFD accumulation may occur via GCN2 sensing of uncharged tRNA when leucine levels are low due to increased catabolism. To support this, they show that leucine supplementation phenocopied the *bcat-1* knockdown and any increase in mitoUFD with leucine is lost in *gcn-2* knockdown. They then perform mitochondrial

proteomics in *C. elegans* and HEK293 cells to identify which proteins change +/- leucine and +/- cyclohexamide, and find an increase in mitochondrial and OMM proteins with leucine supplementation that is unaffected by translational inhibitor cyclohexamide. Finally they find that leucine supplementation increases maximal respiration and this impact is eliminated in *gcn-2* or *cdc-48* overexpressors, or by *tomm-40* knockdown. Overall, the findings of this paper are really interesting but I think given the focus on leucine in the final few figures, title and abstract, some more amino acid controls in some of their experiments is required to see how specific some of the phenomena are to leucine versus other amino acids.

Main comments:

It is likely based on their RNAi screen of proteins related to amino acid metabolism, and the link to *gcn2* which is a general low amino acid sensor that increased levels of multiple other amino acids would have a similar impact and leucine may not be distinct in this capability. This should be explored at least at the level of testing whether supplementing other essential and non-essential amino acids impact mitoUFP accumulation.

The authors claim that leucine supplementation increases mitochondrial respiration by reducing OMM turnover as leucine supplementation to cells with overexpression of *gcn-2* or *cdc-48* which decrease mitoUFD do not have increased respiration in response to leucine. In addition they link this phenotype to mitochondrial protein import as *tomm-40* knockdown also prevents leucine induced increase in respiration.

Leucine can also impact respiration via its direct catabolism to acetyl-CoA. Is it possible that *gcn-2* OE or *tomm-40* kd also leads to decreased levels of BCAA catabolic enzymes in the mitochondria and that contributes to the decreased respiration? As a control, does another amino acid substrate such as glutamine increase respiration and is this compromised by *gcn-2* OE or *tomm-40* knockdown? Does knockdown of components of BCKDH such as ZK669.4 impact the leucine impact on respiration?

Reviewer #3 (Remarks to the Author):

The paper investigated how the regulation of outer mitochondrial membrane (OMM) proteostasis influences mitochondrial function. Through a fluorescent reporter assay and screening around 200 genes in *Caenorhabditis elegans*, researchers discovered that inhibiting the electron transport chain and the tricarboxylic acid cycle affected the degradation of OMM-anchored model substrates. Moreover, they observed that the loss of branched-chain amino acid transferase (*bcat-1*) and increased leucine levels inhibited the ubiquitin-dependent degradation of the model substrate. This led to a stabilization of OMM proteins, resulting in heightened mitochondrial protein content and improved respiration. The study suggests leucine is pivotal in connecting cellular nutrition with mitochondrial activity. While the genetic link between leucine and OMM protein levels is established, the precise mechanism connecting leucine and OMM protein turnover via ubiquitin-dependent pathways remains unclear, which limits enthusiasm for the manuscript in its current state.

Additionally, the paper described two pathways related to outer mitochondrial membrane (OMM) stability: mitochondrial function and leucine availability. However, it failed to establish a clear connection between these aspects. It would be pertinent to explore whether decreased mitochondrial function, induced by ETC and TCA RNAi, leads to elevated cellular leucine levels. If such an elevation occurs, it could potentially stabilize OMM proteins, as suggested by the proposed mechanism, thereby possibly prompting a compensatory increase in mitochondrial function. This hypothetical cascade might involve enhanced mitochondrial import efficiency and improved electron transport chain (ETC) activity. Nevertheless, the proposed sequence of events lacks robust experimental support and may oversimplify the intricate interactions among mitochondrial function, cellular metabolism, and protein turnover. Therefore, further experimentation is imperative to substantiate these hypotheses and elucidate the underlying mechanisms.

Main points:

1. The lack of RNAi efficiency testing throughout the study raises general concerns. This omission may impede the interpretation of certain negative results. For instance, Fig 1 and 2 employ *pdr-1(gk448)* and RNAi knockdown of other key mitophagy components to eliminate the effects of mitophagy and mitochondrial mass on mitoUFD level. However, the study overlooks examining mitochondrial protein content and RNAi efficiency to ensure that mitophagy levels and mitochondrial mass are indeed affected in these contexts.
2. In Fig 3, although the authors noted a decrease in mitoUFD level in response to paraquat, the outcomes of pharmacological and genetic inhibition of individual ETC complexes remain ambiguous. This uncertainty undermines the notion regarding the impact of mitochondrial respiration on mitoUFD turnover.
3. Whether supplementation of other amino acids, particularly the other two branched-chain amino acids (BCAAs), have a specific effect on mitoUFD levels? This aspect warrants further investigation to comprehensively understand the role of amino acid availability in regulating mitochondrial OMM protein turnover.
4. The proposed working model in Fig. 4j suggests that high levels of leucine reduce mitoUFD turnover by inactivating GCN-2. However, contradictory results are observed in Fig. 4h, where *gcn-2* RNAi + leucine, expected to further inactivate GCN-2, leads to lower mitoUFD levels compared to the L4440 control. This inconsistency raises questions regarding the proposed mechanism and warrants clarification.

5. The upregulation of most outer mitochondrial membrane (OMM) proteins mentioned in Fig 5 did not meet the commonly accepted 0.1 p-value cutoff typically set for omics studies. Moreover, no additional evidence from alternative methods was provided to confirm these changes. Even if leucine did reduce the degradation of endogenous OMM proteins, the lack of further verification regarding the involvement of UPS-mediated turnover limits the confidence in these findings.

6. In Fig 6, the authors noted that leucine supplementation increased both basal and maximal mitochondrial respiration, highlighting the necessity of intact TOM-mediated protein import. However, the study lacks direct evidence demonstrating that mitochondrial import efficiency was indeed elevated to transport specific target proteins.

7. Consequently, despite the inhibition of outer mitochondrial membrane (OMM) protein turnover by leucine, as shown in the previous section, it remains unclear how leucine exerts its effect on OMM protein levels, subsequently increasing mitochondrial import efficiency and respiration. The existing gaps between these conclusions and the lack of experimental support underscore the study's reliance on correlations rather than robust causal relationships.

Minor Concerns:

1. Line 68 states that "mitoUFD was ubiquitously expressed in all *C. elegans* tissue," but no corresponding result is presented to support this claim.
2. From Fig. 2, the authors primarily rely on the percentage change of BioSorter fluorescence intensity to indicate alterations in mitoUFD content. However, this measurement may be influenced by various factors such as mitochondrial mass. It may be beneficial to conduct further validations using fluorescence microscopy and Western Blot analysis.
3. The use of the term "crosstalk" in Line 94 might be overstated and potentially misleading, as it implies a relationship between two factors solely based on their mutual influence.
4. In Fig. 1e, f, and S1b, the "hecd-1" and "rpn-8" groups can be interchanged to align with their first mention in the text.
5. It is unclear which animal group serves as the control for calculating the percentage change in Fig. 3a.
6. The insets of Fig. 4b should include the Y-axis title to prevent confusion.
7. If applicable, additional p-values of non-significant groups can be displayed in the figures.
8. The Discussion section would benefit from a comprehensive evaluation of the study's limitations and suggestions for future research directions.

**For Nature Portfolio general information and news for authors, see <http://npg.nature.com/authors>.

Version 1:

Decision Letter:

Dear Professor Hoppe,

Dear Thorsten,

We have decided to reverse the decision and seek further input from the reviewers. Please see below for additional information you must provide, and we will send the revision and rebuttal back to the referees. Please do not hesitate to contact me if you have any further questions or concerns.

Thank you for your email asking us to reconsider our decision on your manuscript, "Metabolic adoption of mitochondrial respiration by leucine-controlled protein degradation". We are always willing to hear the authors' perspective, but we must first prioritize decisions on new submissions. We appreciate your patience while we considered this appeal.

I have now discussed your manuscript, and the referees' comments and your rebuttal, in detail with my colleagues, and we would be willing to reconsider a revised manuscript provided the following issues can be addressed, and that nothing similar is accepted for publication at Nature Cell Biology or published elsewhere in the meantime.

Please pay close attention to our guidelines on statistical and methodological reporting (listed below) as failure to do so may delay the reconsideration of the revised manuscript. In particular please provide:

- a Supplementary Table including all numerical source data in Excel format, with data for different figures provided as different sheets within a single Excel file. The file should include source data giving rise to graphical representations and

statistical descriptions in the paper and for all instances where the figures present representative experiments of multiple independent repeats, the source data of all repeats should be provided.

On resubmission please provide the completed Reporting Summary (found here <https://www.nature.com/documents/nr-reporting-summary.pdf>). This is essential for reconsideration of the manuscript and this document will be available to editors and referees in the event of peer review. For more information see below. Please also ensure that the presentation of statistical information in the revised submission complies with Nature Cell Biology's statistical guidelines (see below).

Please use the link below to submit the complete manuscript files, and include a point-by-point response to the complete reviewer comments, verbatim as provided in their reports.

Link Redacted

Please let us know how you wish to proceed and when we can expect your revised manuscript.

With kind regards,

GUIDELINES FOR EXPERIMENTAL AND STATISTICAL REPORTING

REPORTING REQUIREMENTS – To improve the quality of methods and statistics reporting in our papers we have recently revised the reporting checklist we introduced in 2013. We are now asking all life sciences authors to complete a reporting summary (found here <https://www.nature.com/documents/nr-reporting-summary.pdf>) that collects information on experimental design and reagents. This document is available to referees to aid the evaluation of the manuscript. Please note that this form is a dynamic 'smart pdf' and must therefore be downloaded and completed in Adobe Reader. We will then flatten it for ease of use by the reviewers. If you would like to reference the guidance text as you complete the template, please access these flattened versions at <http://www.nature.com/authors/policies/availability.html>.

Version 2:

Reviewer comments:

Reviewer #1

(Remarks to the Author)

* In the revised manuscript, the authors include several new experiments that address most of the issues from the original submission. Furthermore, they established new connections and extended endogenous protein analyses, as well as added validation in cultured mammalian model cells. The well written work is easy to follow and describes novel interesting findings. I believe that the current version of this manuscript could be accepted for publication without any additional experimental work.

Reviewer #2

(Remarks to the Author)

The authors have adequately addressed my queries.

Reviewer #3

(Remarks to the Author)

While the revised version of the manuscript demonstrates improved data quality with the addition of new experiments, several critical issues remain unresolved, and some data from the previous submission have been removed, which I find concerning.

It is somewhat disappointing that the authors removed the data related to mitochondrial ETC inhibitors, particularly given the original proposal that decreased mitochondrial activity could serve as a trigger for the increased accumulation of OMM proteins to compensate for impaired mitochondrial function. This crucial experiment had the potential to establish a clear link between mitochondrial dysfunction and the turnover of OMM proteins. Its removal weakens the significance of the overall discovery, and the interpretation of the results lacks the necessary context to justify the conclusions drawn.

The manuscript still struggles to explain why specifically leucine, and no other amino acids, is the key modulator of OMM protein turnover. Despite the increased experimental effort to dissect the mechanism through the GCN2-SEL1 axis, it remains puzzling why leucine specifically regulates the turnover of OMM proteins.

The authors have made considerable efforts to elucidate the mechanism through which leucine mediates OMM protein turnover via the GCN2-SEL1 axis. However, the rationale for specifically targeting mitochondrial-localized SEL1 remains unclear. The mechanism behind this specificity should be explored in more depth to understand why the mitochondria are the primary site of action for SEL1 in this process. Additionally, the immunogold analysis presented to support the mitochondrial localization of SEL1 is not very convincing. The quality of the images raises concerns, and the arrows indicating SEL1 localization appear to point to regions where the mitochondria and ER come into close contact. This weakens the argument that SEL1 is truly localized to the mitochondria.

Decision Letter:

Our ref: NCB-A54011B

18th August 2025

Dear Dr. Hoppe,

Thank you for submitting your revised manuscript "Metabolic adoption of mitochondrial respiration by leucine-controlled protein degradation" (NCB-A54011B). It has now been seen by the original referees and their comments are below. The reviewers find that the paper has improved in revision, and therefore we'll be happy in principle to publish it in Nature Cell Biology, pending minor revisions to satisfy the referees' final requests and to comply with our editorial and formatting guidelines.

Thank you again for your interest in Nature Cell Biology Please do not hesitate to contact me if you have any questions.

Sincerely,

Reviewer #1 (Remarks to the Author):

* In the revised manuscript, the authors include several new experiments that address most of the issues from the original submission. Furthermore, they established new connections and extended endogenous protein analyses, as well as added validation in cultured mammalian model cells. The well written work is easy to follow and describes novel interesting findings. I believe that the current version of this manuscript could be accepted for publication without any additional experimental work.

Reviewer #2 (Remarks to the Author):

The authors have adequately addressed my queries.

Reviewer #3 (Remarks to the Author):

While the revised version of the manuscript demonstrates improved data quality with the addition of new experiments, several critical issues remain unresolved, and some data from the previous submission have been removed, which I find concerning.

It is somewhat disappointing that the authors removed the data related to mitochondrial ETC inhibitors, particularly given the original proposal that decreased mitochondrial activity could serve as a trigger for the increased accumulation of OMM proteins to compensate for impaired mitochondrial function. This crucial experiment had the potential to establish a clear link between mitochondrial dysfunction and the turnover of OMM proteins. Its removal weakens the significance of the overall discovery, and the interpretation of the results lacks the necessary context to justify the conclusions drawn.

The manuscript still struggles to explain why specifically leucine, and no other amino acids, is the key modulator of OMM protein turnover. Despite the increased experimental effort to dissect the mechanism through the GCN2-SEL1 axis, it remains puzzling why leucine specifically regulates the turnover of OMM proteins.

The authors have made considerable efforts to elucidate the mechanism through which leucine mediates OMM protein turnover via the GCN2-SEL1 axis. However, the rationale for specifically targeting mitochondrial-localized SEL1 remains unclear. The mechanism behind this specificity should be explored in more depth to understand why the mitochondria are the primary site of action for SEL1 in this process. Additionally, the immunogold analysis presented to support the mitochondrial localization of SEL1 is not very convincing. The quality of the images raises concerns, and the arrows indicating SEL1 localization appear to point to regions where the mitochondria and ER come into close contact. This weakens the argument that SEL1 is truly localized to the mitochondria.

Version 3:

Decision Letter:

Dear Dr Hoppe,

I am pleased to inform you that your manuscript, "Leucine inhibits degradation of outer mitochondrial membrane proteins to adapt mitochondrial respiration", has now been accepted for publication in Nature Cell Biology.

Please note that *Nature Cell Biology* is a Transformative Journal (TJ). Authors may publish their research with us through the traditional subscription access route or make their paper immediately open access through payment of an article-processing charge (APC). Authors will not be required to make a final decision about access to their article until it has been accepted. [Find out more about Transformative Journals](https://www.springernature.com/gp/open-research/transformative-journals)

Authors may need to take specific actions to achieve compliance with funder and institutional open access mandates. If your research is supported by a funder that requires immediate open access (e.g. according to [Plan S principles](https://www.springernature.com/gp/open-science/plan-s-compliance) or the [NIH public access policy](https://www.springernature.com/gp/open-science/us-federal-agency-compliance)) then you should select the gold OA route, and we will direct you to the compliant route where possible. Because authors warrant under our subscription licensing terms that they haven't committed to licensing any version of their article under a licence inconsistent with the terms of our agreement – including the applicable embargo period – publication under the subscription model isn't suitable for authors whose funders require no embargo.

If you have not already done so, we strongly recommend that you upload the step-by-step protocols used in this manuscript to protocols.io (<https://protocols.io>), an open online resource that allows researchers to share their detailed experimental know-how. All uploaded protocols are made freely available and are assigned DOIs for ease of citation. Protocols and Nature Portfolio journal papers in which they are used can be linked to one another, and this link is clearly and prominently visible in the online versions of both. Authors who performed the specific experiments can act as primary authors for the Protocol as they will be best placed to share the methodology details, but the Corresponding Author of the present research paper should be included as one of the authors. By uploading your Protocols onto protocols.io, you are enabling researchers to more readily reproduce or adapt the methodology you use, as well as increasing the visibility of your protocols and papers. You can also establish a dedicated workspace to collect your lab Protocols. Further information can be found at <https://www.protocols.io/help/publish-articles>.

Nature Cell Biology encourages authors presenting evidence for cell, biological, molecular, and genetic interactions to consider communicating these findings using Biofactoid (<https://biofactoid.org/>). This tool helps users share a searchable representation of interactions (e.g. binding, gene expression, post-translational modification) between genes, gene products, or chemicals. Information added to Biofactoid, with author attribution, is shared on social media and public databases, such as Pathway Commons, where it can be discovered and analyzed in the context of a large and growing corpus of knowledge.

With kind regards,

** Visit the Springer Nature Editorial and Publishing website at http://editorial-jobs.springernature.com?utm_source=ejP_NCB_email&utm_medium=ejP_NCB_email&utm_campaign=ejp_NCB for more information about our career opportunities. If you have any questions please click [here](mailto:editorial.publishing.jobs@springernature.com).

Point-by-point response to reviewer comments

We would like to thank all the reviewers for taking the time to read our manuscript and for their valuable comments and suggestions for improvement. Our responses are written in blue. To streamline the presentation of key findings and incorporate several important new results, we have made extensive revisions throughout the manuscript. Accordingly, the figure numbers referenced below correspond to those in the revised version of the manuscript in the resubmission.

Reviewer #1:

In the manuscript by Li et al., the availability of leucine affects the turnover rates of the OMM proteins and thereby provides a protective mechanism for maintaining mitochondrial function. The work applies *C. elegans* as a main research model and genetic approaches combined with detecting fluorescently tagged uncleavable ubiquitin targeted to the OMM. This is a well-written study, and, in most cases, data support conclusions. However, many conclusions are based on insufficient experimental evidence (see Specific Comments below).

We thank the reviewer for carefully reading the manuscript and providing constructive suggestions for improvement. According to the reviewer's comments, we have made significant revisions. Specifically, we have: 1) conducted additional verifications of the mitoUFD model substrate in *C. elegans* and human cells to further confirm its applicability in studying the ubiquitin-dependent degradation of OMM proteins; 2) identified a clear mechanism by which leucine regulates the ubiquitin-dependent degradation of OMM proteins. We show that leucine levels modulate the abundance of the E3 ubiquitin ligase HRD1 cofactor, SEL-1/SEL1L, at the OMM. We will discuss the detailed experiments below in correspondence with the specific comments.

Furthermore, while this work opens some new research avenues, it needs to propose a clear mechanism by which the OMM protein stability is affected by the OXPHOS activity, other aspects of mitochondrial homeostasis (fusion, fission, etc.), and isoleucine availability.

We thank the reviewer for raising this important point. In the previous version of the manuscript, we reported that pharmacological inhibition of Complex I (CI), but not other ETC complexes, reduced mitoUFD levels. However, in new experiments, we found that RNAi-mediated knockdown of ETC subunits, including the CI subunit NUO-1, produced the opposite effect—slightly increasing mitoUFD levels (**Extended Data Fig. 2b**). These contrasting results may reflect differences in the extent and mode of ETC inhibition. Notably, RNAi knockdown of ETC subunits strongly disrupts cytosolic proteostasis (also shown by our lab before in Segref et al., 2014¹), and this disruption appears more pronounced than its effect on OMM proteostasis (**Extended Data Fig. 2b**). Due to the complexity of ETC activity and its broader implications for coordinating cytosolic and mitochondrial proteostasis, we plan to explore this question further in follow-up studies.

Taken from *Extended Data Fig. 2b*.

In the revised manuscript, we focused exclusively on the leucine-controlled regulation of OMM proteostasis and the subsequent adaptation of mitochondrial respiratory activity. Through additional experiments, we now provide a clear mechanism: elevated leucine levels downregulate the E3 ubiquitin ligase cofactor SEL-1/SEL1L-dependent OMM protein degradation. Consequently, stabilized OMM proteins, including components of the TOM complex, lead to an increased content of mitochondrial proteins, including those involved in the TCA cycle, OXPHOS, and amino acid metabolism. We will discuss the detailed experiments below in correspondence with the specific comments.

The ample genetic evidence supports the role of leucine availability in controlling the expression of the mitochondrial model substrate UbV-GFP. However, many exciting observations need additional work to solidify each of these findings. The applicability of UbV-GFP for these studies should be further verified. The authors often state that they observe changes in UbV-GFP turnover but only analyze a steady state expression level under many distinct conditions. The turnover experiments were not included. Furthermore, the ubiquitination status of this sensor (as well as the representative endogenous proteins) under many tested conditions should also be verified. Is this protein polyubiquitinated? Is its ubiquitination controlled by the OXPHOS activity and other pathways analyzed here?

We appreciate the reviewer's agreement with our genetic analysis and support of our conclusion.

1. We agreed that more research was needed to validate the applicability of the mitoUFD model substrate, so we performed the following experiments.

- We performed a chase assay of mitoUFD and compared its stability with that of the control mitoGFP (which lacks the ubiquitin fusion as the degradation signal) expressed in *C. elegans* (Fig. 1e, f). The cycloheximide chase assay revealed that the levels of the control mitoGFP remained stable over nine hours, while those of the mitoUFD showed fast turnover.

Taken from **Fig. 1e, f**.

- We performed a mitoUFD pulldown from enriched mitochondria, followed by mass spectrometry analysis to identify the binding partners of the mitoUFD substrate (**Fig. 1g**). By pulling down the mitoUFD substrate, we identified its binding partners: CDC-48, ubiquitin-associated proteins, and proteasome subunits.

Taken from **Fig. 1g**.

- We performed TUBE-IP on enriched and lysed mitochondria to confirm the polyubiquitylation of the mitoUFD substrate (**Fig. 1h**).

Taken from **Fig. 1h**.

- We established a stable human cell line that expresses the mitoUFD substrate. Upon bortezomib treatment, this substrate was ubiquitylated and targeted for proteasomal degradation in HEK293 cells (**Fig. 2i**, **Extended Data Fig. 5b**).

Taken from **Fig. 2i** and **Extended Data Fig. 5b**.

2. To validate the increased stability of the mitochondrial UFD substrate, we conducted a chase assay to examine leucine effect. Indeed, the mitoUFD substrate exhibited greater stability when exposed to an excess of leucine (**Fig. 2k**).

Taken from **Fig. 2k**.

3. To examine the ubiquitylation state, we isolated endogenous OMM proteins and found that the level of global ubiquitylation decreased significantly with the addition of leucine (**Fig. 3a**). Furthermore, our results are consistent with our genetic analysis and show that GCN2 mediates the reduction of OMM protein ubiquitylation in response to leucine (**Fig. 3b**).

Taken from Fig. 3a, b.

While the study by Li et al. offers several insightful observations, it also raises significant concerns, particularly the lack of mechanistic depth in its current form. To enhance the study's suitability for publication, it is essential to conduct additional experiments that provide a more comprehensive understanding of the mechanisms underlying the observed effects.

We thank the reviewer for recognizing the impact of our work, as well as for providing valuable suggestions for improvement. During the revision process, we identified a clear mechanism by which leucine regulates the ubiquitylation and degradation of OMM proteins via the E3 ubiquitin ligase cofactor, SEL1L. We have revised our proposed model accordingly (Fig. 5s): High levels of leucine inhibit GCN2 and reduce SEL1L-HRD1-dependent degradation of OMM proteins, thereby stabilizing key components of the protein import machinery, such as TOMM40. This leads to an expansion of the mitochondrial proteome, including the respiratory chain, the TCA cycle, amino acid metabolism and metabolite transport. Consequently, mitochondrial respiratory capacity and cell viability are enhanced.

Taken from Fig. 5s.

Specific comments.

1. Figure 1. It is expected that in the presence of bortezomib or the absence of active proteasome, the mitochondrial substrate (UbV-GFP) would be ubiquitinated (Fig. 1d,f). Why do the authors show only one protein band? Is ubiquitination (higher molecular weight smear) undetectable in this system? The same applies to data showing CDC48-dependent extraction of UbV-GFP (Fig. 2). Please comment on this or add the ubiquitination data.

We thank the reviewer for pointing out this important issue. We have now added results demonstrating the ubiquitylation of the mitoUFD substrate in both control and *cdc-48* RNAi knockdown worms. *cdc-48* RNAi animals accumulated polyubiquitylated forms of the mitoUFD substrate (Fig. 1h).

Taken from Fig. 1h.

Additionally, by pulldown of the mitoUFD substrate, we identified CDC-48 as its binding partner, as well as some ubiquitin-associated/related proteins and subunits of the proteasome (Fig. 1g).

Taken from Fig. 1g.

Moreover, in the previous version of the manuscript, we showed that knocking down the expression of *cdc-48*/CDC48 or its cofactors *ufd-3*/DOA1, *npl-4*/NPL4, and *ufd-1*/UFD1

increased the levels of the mitoUFD substrate but not the levels of the mitoGFP substrate at the mitochondrial surface (Extended Data Fig. 1f-h). These results together suggest that CDC-48 is likely involved in the extraction of the mitoUFD substrate from the OMM for proteasomal degradation.

Additionally, we expressed the same mitoUFD substrate in human HEK293 cells and observed clear ubiquitylated mitoUFD upon bortezomib treatment (**Fig. 2i**).

Taken from Fig. 2i.

Under standard Western blot conditions, the ubiquitylated form of the mitoUFD substrate was visible in HEK293 cells treated with bortezomib. Ubiquitylated forms of the mitoUFD substrate were difficult to detect in *C. elegans*, even with bortezomib treatment. However, the ubiquitylated form was clearly visible when TUBE1 agarose was used to pull down the polyubiquitylated substrate from the enriched mitochondria.

2. It is unclear whether HECD-1 acts directly on the mitochondrial UbV-GFP, or if this protein controls UbV-GFP degradation in the cytosol before inserting into the OMM. Many reports indicate that exogenous or abnormal mitochondrial proteins are degraded in the cytosol before their mitochondrial import. Please test this possibility. Generally, the direct versus indirect role of HECD-1 in mitochondrial proteostasis should be investigated in more detail.

We thank the reviewer for this comment. To make the content more focused and coherent, we removed the initial results regarding HECD-1 from the revised manuscript. Instead, we focused on the SEL-1(SEL1L)-SEL-11(HRD1) E3 ubiquitin ligase complex. Our proteomic analysis revealed that SEL-1 is regulated by leucine treatment, whereas HECD-1 is not (Fig. 3). Specifically, we demonstrate:

1. The SEL-1(SEL1L) level is specifically reduced in the mitochondrial fraction in worms upon leucine supplementation (**Fig. 3c,d**),

Taken from Fig. 3c, d.

and a similar effect is observed for SEL1L in human cells (Fig. 3h).

Taken from Fig. 3h.

Consistent with the specific downregulation of SEL-1 levels in the mitochondrial fraction of worms, we found that the levels of the endoplasmic reticulum-associated degradation (ERAD) substrate CPL-1*, which is degraded by SEL-1 and SEL-11², were not stabilized by leucine supplementation (Extended Data Fig. 8b, c).

Taken from Extended Data Fig. 8b, c.

2. SEL-1 is a membrane-bound protein in worms that co-fractionates with the mitoUFD substrate (Extended Data Fig. 8d). SEL-1 and SEL-11 bind more than twice to mitoUFD compared to the mitoGFP control (Extended Data Fig. 8e), which suggests that the SEL-1-SEL-11 E3 ligase complex interacts directly with and ubiquitylates OMM proteins.

d**e**
Taken from Extended Data Fig. 8d, e.

3. RNAi knockdown of SEL-1 or SEL-11 stabilized the mitoUFD substrate at the OMM in *C. elegans* (Fig. 3f, g).

Taken from Fig. 3f, g.

4. RNAi knockdown of SEL-1 or SEL-11 is sufficient to enhance mitochondrial respiratory activity in worms (Fig. 5j, Extended Data Fig. 12i).

Taken from Fig. 5j and Extended Data Fig. 12i.

Overall, our results suggest that high concentrations of leucine decrease SEL-1-SEL-11-dependent degradation of OMM proteins and increase mitochondrial respiration.

We discussed our results as follow in the discussion part: ‘The identification of the SEL1-SEL-11/SEL-1L-HRD1 complex acting specifically at the OMM and its modulation by leucine availability adds an important new dimension to our understanding of mitochondrial proteostasis. Recent studies have shown that the SEL1L-HRD1 complex binds to OMM proteins such as SAMM50, PGAM5, and VDAC1 and might regulate the stability of MTCH2³, implying a role for ER-associated E3 ligases and cofactors in OMM protein degradation. Furthermore, SEL1L-HRD1 has been shown to regulate mitochondrial dynamics by modulating ER-mitochondria contacts in brown adipocytes, which is essential for metabolic adaptation to cold challenge⁴. Other ERAD components, such as Cdc48/p97 and Ubx2, have also been implicated in mitochondrial protein quality control⁵⁻¹¹, and there is evidence that Ubx2 localizes to both the ER and OMM¹⁰.’

3. Figure 3. The authors state that paraquat and rotenone affect UbV-GFP turnover. However, the data only show quantifications of steady-state protein levels. The authors should test the turnover rates of UbV-GFP in paraquat- and rotenone-treated cells. Furthermore, since paraquat is known to generate superoxide anion, verifying the degree to which antioxidative agents affect these changes would be interesting.

To avoid making the manuscript overly complex, we removed the section on paraquat and rotenone treatment and focused solely on leucine. We believe that paraquat and rotenone affect OMM proteostasis through mechanisms different from those of leucine-controlled protein degradation. We would like to further investigate this complex question in follow-up studies.

4. In Figure 4, it would be beneficial if the authors could provide original Western blot and imaging data for the most critical experiments.

We thank the reviewer for pointing this out. In the revised manuscript, the results that corresponded to the previous Fig. 4 are now in Fig. 2. We have now included confocal images regarding GCN-2 RNAi, BCAT-1 RNAi, and leucine treatment at 20 mM and 50 mM (**Extended Data Fig. 4a, b**). These confirm the OMM-specific accumulation of the mitoUFD substrate under these conditions.

Taken from Extended Data Fig. 4a, b.

5. Figure 5 of the manuscript reveals that leucine supplementation affects the expression levels of many mitochondrial proteins, including not only the OMM factors but also TCA and OXPHOS proteins. This finding suggests a potential role of leucine in mitochondrial biogenesis and somewhat challenges the specificity of the OMM protein turnover in this process. Could the authors comment on this possibility? Furthermore, including endogenous mitochondrial proteins as loading controls in Western blot experiments could address the total mitochondrial biogenesis versus specific changes in the OMM proteins.

We thank the reviewer for the comments and suggestions. Along with new experiments, we now present additional evidence supporting our finding that leucine regulates OMM protein degradation:

1. In our previous experiments, we demonstrated that increases in OMM and mitochondrial protein content are independent of protein translation (**Fig. 4c, h**).

Taken from Fig. 4c, h.

2. We performed a new proteomic analysis on animals that overexpress CDC-48.1. These animals' OMM proteins are degraded more actively than those of WT animals (Fig. 5a). When CDC-48.1 is overexpressed, the amount of OMM and mitochondrial proteins does not increase upon leucine treatment (Fig. 5c-f).

Taken from Fig. 5c-f.

Unlike WT animals (Extended Data Fig. 9e), we did not observe upregulation of GO terms related to OXPHOS and TCA activity in CDC-48.1 overexpression worms.

Taken from Extended Data Fig. 9e. Significantly enriched GO terms for upregulated proteins of WT worm Mito proteome upon leucine treatment.

Significantly enriched GO terms for upregulated proteins of CDC-48.1 OE worm Mito proteome upon leucine treatment.

These results further support our hypothesis that the degradation of OMM proteins, rather than mitochondrial biogenesis in general, contributes to the increase in OMM and mitochondrial proteome induced by leucine.

3. Moreover, in our previous Seahorse analyses, which measures oxygen consumption rates, an increase in mitochondrial respiration caused by leucine is prevented when OMM proteins are more efficiently degraded by either CDC-48 or GCN-2 overexpression (Fig. 5a, b, g, h).

4. We have now included the validation of the turnover rates of mitoUFD and TOMM40 using MTCO1 and total mitochondrial protein as controls (Fig. 2k, Extended Data Fig. 11a, b).

Taken from Fig. 2k.

Taken from *Extended Data Fig. 11a, b*.

These results together strongly suggest that leucine inhibits the degradation of OMM proteins, including subunits of the TOM complex. This increases protein import into mitochondria, including proteins involved in the TCA cycle and OXPHOS activity. Consequently, mitochondrial respiration is enhanced. Because in our experiments, leucine treatment is short-term (3-hour max), we hypothesize that modulation of OMM proteins is a rapid response that adjusts mitochondrial activity prior to anabolic metabolic processes such as mitochondrial biogenesis.

6. Figure 5. Can the authors demonstrate that the stability of their target OMM proteins (e.g., Tom40 or VDAC1) is indeed affected by leucine treatment in worms and HEK cells? Such experiments are necessary to support the conclusion that leucine levels regulate the Ub-dependent degradation of these endogenous proteins. The endogenous protein cycloheximide chase can be easily done in mammalian cells for which antibodies are readily available. However, knowing the turnover rates of the UbV-GFP in worms would add additional depth.

We thank the reviewer for pointing this out. As mentioned above, we have now added the chase assay result for mitoUFD model substrate in HEK293 cells (**Fig. 2k**), and we confirmed the increased protein stability of TOMM40 upon leucine supplementation by western blot, whereas the stability of the IMM protein MTCO1 is not affected by leucine treatment (**Extended Data Fig. 11a, b**).

Reviewer #2 (Remarks to the Author):

In this paper, the investigators develop a way to monitor outer mitochondrial membrane proteostasis by fusing a ubiquitinated GFP to the mitochondrial transmembrane domain of FIS-1(mitoUFD) which allows them to monitor accumulation of mito-UFD as an indicator of decreased OMM protein turnover. They then use RNAi to screen the impact of knockdown of a range of proteins exploring different aspects of the regulation of OMM degradation and the impact of metabolism and different stressors. They eventually hone in on amino acid metabolism of which there was an enrichment in BCAA catabolism related genes in those genes that had the largest impact on mitoUFD accumulation when knocked down. Through a series of further experiments knockdown targeted at the mTOR pathway or *gcn-2*, they deduce that the impact of altered BCAA catabolism on mitoUFD accumulation may occur via GCN2 sensing of uncharged tRNA when leucine levels are low due to increased catabolism. To support this, they show that leucine supplementation phenocopied the *beat-1* knockdown and any increase in mitoUFP with leucine is lost in *gcn-2* knockdown. They then perform mitochondrial proteomics in *C. elegans* and hek293 cells to identify which proteins change +/- leucine and +/- cycloheximide, and find an increase in mitochondrial and OMM proteins with leucine supplementation that is unaffected by translational inhibitor cycloheximide. Finally they find that leucine supplementation increases maximal respiration and this impact is eliminated in *gcn-2* or *cdc-48* overexpressors, or by *tomm-40* knockdown. Overall, the findings of this paper are really interesting but I think given the focus on leucine in the final few figures, title and abstract, some more amino acid controls in some of their experiments is required to see how specific some of the phenomena are to leucine versus other amino acids.

Main comments:

it is likely based on their RNAi screen of proteins related to amino acid metabolism, and the link to *gcn2* which is a general low amino acid sensor that increased levels of multiple other amino acids would have a similar impact and leucine may not be distinct in this capability. This should be explored at least at the level of testing whether supplementing other essential and non-essential amino acids impact mitoUFP accumulation.

We thank the reviewer for carefully reading our manuscript and recognizing its potential. We also appreciate the comments and suggestions regarding testing other amino acids.

To address this point:

1. We checked the other two branched-chain amino acids: isoleucine and valine. We did not detect a significant effect on mitoUFD regulation (**Fig. 2f**).

Taken from Fig. 2f.

2. We also performed a Seahorse analysis on several other essential amino acids and found that leucine uniquely and significantly enhances maximal mitochondrial respiration (Extended Data Fig. 12a, b).

Taken from Extended Data Fig. 12a, b.

The authors claim that leucine supplementation increases mitochondrial respiration by reducing OMM turnover as leucine supplementation to cells with overexpression of *gcn-2* or *cdc-48* which decrease mitoUFD do not have increased respiration in response to leucine. In addition, they link this phenotype to mitochondrial protein import as *tomm-40* knockdown also prevents leucine induced increase in respiration. Leucine can also impact respiration via its direct catabolism to acetyl-CoA. Is it possible that *gcn-2* OE or *tomm-40* kd also leads to decreased levels of BCAA catabolic enzymes in the mitochondria and that contributes to the decreased respiration? As a control, does another amino acid substrate such as glutamine increase respiration and is this compromised by *gcn-2* OE or *tomm-40* knockdown?

We thank the reviewer for pointing out that GCN-2 overexpression (OE) or TOMM-40 knockdown (KD) could lead to decreased levels of BCAA catabolic enzymes in mitochondria, contributing to respiratory adaptation. Our results strongly support this hypothesis, which likely occurs downstream of OMM protein degradation. In WT animals, many of the proteins involved in the TCA cycle, OXPHOS, and amino acid metabolism are upregulated independently of translation upon leucine supplementation (Fig. 4k, l).

k Worm_mito: Leu+CHX vs. CHX**l** HEK293 cell_mito: Leu+CHX vs. CHX
Taken from **Fig. 4k, l**.

This strongly suggests an increased import of mitochondrial proteins to support enhanced metabolic activity in mitochondria. However, upon GCN-2 OE or TOMM-40 KD, the increase in mitochondrial protein import by leucine is likely prevented because OMM proteins are actively degraded (in the case of GCN-2 OE) or import is directly disrupted (in the case of TOMM-40 KD). To further investigate this, we performed a new proteomic analysis of CDC-48 OE worms, whose OMM proteins were more efficiently degraded, similar to GCN-2 OE. In these animals, leucine was unable to increase OMM proteins or mitochondrial protein content (**Fig. 5c-f**).

Taken from **Fig. 5c-f**.

Unlike WT animals (**Fig. 4k, l**), proteins related to OXPHOS, TCA activity, or amino acid metabolism did not show significantly enriched upregulation in CDC-48 OE worms.

Significantly enriched GO terms for upregulated proteins of CDC-48.1 OE worm Mito proteome upon leucine treatment.

Accordingly, worms overexpressing CDC-48 do not exhibit an increase in respiration induced by leucine (Fig. 5b).

We also performed a Seahorse analysis with a 3-hour treatment of 1 mM glutamine but did not observe a significant effect on mitochondrial respiration in HEK293 cells.

Seahorse analysis for 3-hour treatment of 1 mM glutamine in HEK293 cells.

Does knockdown of components of BCKDH such as ZK669.4 impact the leucine impact on respiration?

We performed an RNAi knockdown to study the effects of the BCKDH component *ZK669.4* (*dbt-1*) in *C. elegans* but did not observe a significant effect on the accumulation of mitoUFD.

Changes of mitoUFD levels upon RNAi knockdown of dbt-1 and bcat-1 compared to control L4440 RNAi.

In addition, we tested another BCKDH component, DLD-1. Although DLD-1 increased mitoUFD levels, it also increased cytoUFD levels by approximately 200% (Fig. 2d, e). These results suggest that DLD-1 exerts a stronger influence on cytosolic proteostasis than on the OMM. Our lab previously observed this phenomenon with the knockdown of IVD-1, which acts downstream of BCKDH^{1,12}.

Taken from Fig. 2d, e.

Reviewer #3 (Remarks to the Author):

The paper investigated how the regulation of outer mitochondrial membrane (OMM) proteostasis influences mitochondrial function. Through a fluorescent reporter assay and screening around 200 genes in *Caenorhabditis elegans*, researchers discovered that inhibiting the electron transport chain and the tricarboxylic acid cycle affected the degradation of OMM-anchored model substrates. Moreover, they observed that the loss of branched-chain amino acid transferase (*bcat-1*) and increased leucine levels inhibited the ubiquitin-dependent degradation of the model substrate. This led to a stabilization of OMM proteins, resulting in heightened mitochondrial protein content and improved respiration. The study suggests leucine is pivotal in connecting cellular nutrition with mitochondrial activity. While the genetic link between leucine and OMM protein levels is established, the precise mechanism connecting leucine and OMM protein turnover via ubiquitin-dependent pathways remains unclear, which limits enthusiasm for the manuscript in its current state.

We thank the reviewer for recognizing the strengths of our genetic approach and suggesting ways to improve the manuscript. We have now undertaken a comprehensive set of new experiments that uncover a clear mechanism linking leucine availability to the ubiquitin-dependent degradation of OMM proteins. Specifically, we demonstrate that the E3 ubiquitin ligase cofactor SEL-1/SEL1L is essential for this process in both *C. elegans* and human cells. Leucine modulates SEL-1/SEL1L abundance at the OMM, thereby regulating the ubiquitylation and degradation of OMM proteins. We will discuss the corresponding experiments in detail below.

Additionally, the paper described two pathways related to outer mitochondrial membrane (OMM) stability: mitochondrial function and leucine availability. However, it failed to establish a clear connection between these aspects. It would be pertinent to explore whether decreased mitochondrial function, induced by ETC and TCA RNAi, leads to elevated cellular leucine levels. If such an elevation occurs, it could potentially stabilize OMM proteins, as suggested by the proposed mechanism, thereby possibly prompting a compensatory increase in mitochondrial function. This hypothetical cascade might involve enhanced mitochondrial import efficiency and improved electron transport chain (ETC) activity. Nevertheless, the

proposed sequence of events lacks robust experimental support and may oversimplify the intricate interactions among mitochondrial function, cellular metabolism, and protein turnover. Therefore, further experimentation is imperative to substantiate these hypotheses and elucidate the underlying mechanisms.

We appreciate the reviewer's suggestion of possible connections between leucine availability and the inhibition of the ETC and TCA. However, based on previous evidence, inhibition of the ETC or TCA did not elevate leucine levels¹³. In the previous version of the manuscript, we reported that pharmacological inhibition of Complex I (CI), but not other ETC complexes, reduced mitoUFD levels. However, in new experiments, we found that RNAi-mediated knockdown of ETC subunits, including the CI subunit NUO-1, produced the opposite effect—slightly increasing mitoUFD levels (**Extended Data Fig. 2b**). These contrasting results may reflect differences in the extent and mode of ETC inhibition. Notably, RNAi knockdown of ETC subunits strongly disrupts cytosolic proteostasis (also shown before in Segref et al., 2014¹), and this disruption appears more pronounced than its effect on OMM proteostasis (**Extended Data Fig. 2b**). Given the complexity and broader implications of how ETC activity may coordinate cytosolic and mitochondrial proteostasis, we plan to further explore this complex question in follow-up studies.

Taken from **Extended Data Fig. 2b**.

In the revised manuscript, we focused exclusively on the leucine-controlled regulation of OMM proteostasis and the subsequent adaptation of mitochondrial respiratory activity. Through additional experiments, we now provide a clear mechanism: elevated leucine levels downregulate the E3 ubiquitin ligase cofactor SEL-1/SEL1L-dependent OMM protein degradation. Consequently, stabilized OMM proteins, including TOM complex components, lead to increased mitochondrial protein content including proteins involved in the TCA cycle, OXPHOS, and amino acid metabolism. We will discuss the detailed experiments below in correspondence with the specific comments.

Main points:

1. The lack of RNAi efficiency testing throughout the study raises general concerns. This omission may impede the interpretation of certain negative results. For instance, Fig. 1 and 2 employ *pdr-1(gk448)* and RNAi knockdown of other key mitophagy components to eliminate the effects of mitophagy and mitochondrial mass on mitoUFD level. However, the study

overlooks examining mitochondrial protein content and RNAi efficiency to ensure that mitophagy levels and mitochondrial mass are indeed affected in these contexts.

We thank the reviewer for this comment. In the previous version of the manuscript, we validated key findings using both RNAi-mediated knockdown and corresponding loss-of-function mutants in *C. elegans*—for example, *cdc-48*, *hecd-1*, and *pdr-1*. In the case of the leucine effect on mitoUFD, we confirmed the result through two independent approaches: RNAi knockdown of BCAT-1 (the first enzyme in leucine catabolism) and direct leucine supplementation. That said, we acknowledge the reviewer’s concern regarding RNAi efficiency and its impact on interpreting certain negative results, particularly in the context of mitophagy. Given that mitophagy was not a central focus of our study, and to avoid potential confusion from results not directly related to our main findings, we have removed those data from the revised manuscript. Importantly, our revised data robustly support the conclusion that mitoUFD is primarily degraded through a UPS-dependent mechanism, as detailed below:

1. Treatment of worms with bortezomib inhibited proteasome activity, which stabilized the mitoUFD substrate, but not the control mitoGFP which lacks the ubiquitin fusion as the degradation signal (**Fig. 1c, d**);

Taken from **Fig. 1c, d**.

2. The overexpression of the proteasome subunit RPN-6.1 enhances the degradation of the mitoUFD substrate in worms (**Extended Data Fig. 1c,d**).

Taken from **Extended Data Fig. 1c,d**.

The RPN-6.1 proteasome subunit was also enriched in our new mass spectrometry analysis to identify the binding partners of the mitoUFD substrate in worms (**Fig. 1g**).

Taken from **Fig. 1g**.

2. In Fig 3, although the authors noted a decrease in mitoUFD level in response to paraquat, the outcomes of pharmacological and genetic inhibition of individual ETC complexes remain ambiguous. This uncertainty undermines the notion regarding the impact of mitochondrial respiration on mitoUFD turnover.

We thank the reviewer for this comment. As mentioned above, in the revised manuscript, we focused exclusively on the leucine-controlled regulation of mitochondrial activity and plan to further investigate how ETC activity coordinates mitochondrial and cytosolic proteostasis in follow-up studies.

3. Whether supplementation of other amino acids, particularly the other two branched-chain amino acids (BCAAs), have a specific effect on mitoUFD levels? This aspect warrants further investigation to comprehensively understand the role of amino acid availability in regulating mitochondrial OMM protein turnover.

We thank the reviewer for pointing this out. We have now examined the other two branched-chain amino acids, isoleucine and valine, and found no significant changes of mitoUFD level in worms (**Fig. 2f**).

Taken from Fig. 2f.

We also performed a Seahorse analysis on several other essential amino acids and found that leucine uniquely and significantly enhances maximal mitochondrial respiration (Extended Data Fig. 12a, b).

Taken from Extended Data Fig. 12a, b.

4. The proposed working model in Fig. 4j suggests that high levels of leucine reduce mitoUFD turnover by inactivating GCN-2. However, contradictory results are observed in Fig. 4h, where *gcn-2* RNAi + leucine, expected to further inactivate GCN-2, leads to lower mitoUFD levels compared to the L4440 control. This inconsistency raises questions regarding the proposed mechanism and warrants clarification.

We apologize for the confusion. We have now included a figure showing how the experiment was performed in Fig. 2h. We have also revised the Y-axis title to more clearly indicate the comparison.

Taken from Fig. 2h.

5. The upregulation of most outer mitochondrial membrane (OMM) proteins mentioned in Fig. 5 did not meet the commonly accepted 0.1 p-value cutoff typically set for omics studies. Moreover, no additional evidence from alternative methods was provided to confirm these changes. Even if leucine did reduce the degradation of endogenous OMM proteins, the lack of further verification regarding the involvement of UPS-mediated turnover limits the confidence in these findings.

We thank the reviewer for the comment. We have now added the chase assay results for the mitoUFD model substrate and the endogenous outer membrane protein TOMM40 in HEK293 cells. Indeed, we observe increased stability of both proteins with leucine supplementation (**Fig. 2k, Extended Data Fig. 11a, b**). As control, the stability of the IMM protein MTCO1 is unaffected by leucine treatment.

Taken from **Fig. 2k**.

Taken from **Extended Data Fig. 11a, b**.

To examine the ubiquitylation states of the endogenous OMM proteins, we now isolated the total OMM proteins and observed a significant reduction in the global ubiquitylation level with leucine supplementation (**Fig. 3a**).

Taken from Fig. 3a.

Additionally, our new analysis revealed that the SEL-1-SEL-11 E3 ubiquitin ligase complex is essential for mediating the degradation of OMM proteins in response to leucine in worms. Specifically, we demonstrate that:

1. The SEL-1 level is specifically reduced in the mitochondrial fraction upon leucine supplementation (Fig. 3c,d),

Taken from Fig. 3c, d.

and a similar effect is observed in human cells (Fig. 3h).

Taken from Fig. 3h.

Consistent with the specific downregulation of SEL-1 levels in the mitochondrial fraction, we found that the levels of the endoplasmic reticulum-associated degradation (ERAD) substrate CPL-1*, which is degraded by SEL-1 and SEL-11², were not affected by leucine supplementation in worms (Extended Data Fig. 8b, c).

Taken from *Extended Data Fig. 8b, c*.

2. SEL-1 is a membrane-bound protein in worms that co-fractionates with the mitoUFD substrate (*Extended Data Fig. 8d*). SEL-1 and SEL-11 bind more than twice to mitoUFD as to mitoGFP (*Extended Data Fig. 8e*), which suggests that the SEL-1-SEL-11 E3 ligase complex interacts directly with and ubiquitylates OMM proteins.

Taken from *Extended Data Fig. 8d, e*.

3. RNAi knockdown of SEL-1 or SEL-11 stabilized the mitoUFD substrate at the OMM (*Fig. 3f, g*).

Taken from Fig. 3f, g.

4. RNAi knockdown of SEL-1 or SEL-11 is sufficient to enhance mitochondrial respiratory activity (Fig. 5j, Extended Data Fig. 12i).

Taken from Fig. 5j and Extended Data Fig. 12i.

Overall, our results suggest that high concentrations of leucine decrease SEL-1-SEL-11-dependent degradation of OMM proteins and increase mitochondrial respiration in worms.

6. In Fig 6, the authors noted that leucine supplementation increased both basal and maximal mitochondrial respiration, highlighting the necessity of intact TOM-mediated protein import. However, the study lacks direct evidence demonstrating that mitochondrial import efficiency was indeed elevated to transport specific target proteins.

We thank the reviewer for this comment. As our proteomic analysis shows, although changes in individual mitochondrial proteins are modest, the overall increase in mitochondrial protein abundance is clear and statistically significant (Fig. 4c, h). This global, translation-independent increase supports our conclusion that mitochondrial import is enhanced by leucine supplementation. Importantly, our conclusion does not depend on the selective import of specific proteins. Rather, we propose that leucine promotes a general enhancement of mitochondrial import efficiency, likely by stabilizing TOM complex components at the outer mitochondrial membrane (OMM). A standard import assay¹⁴ tracks the import efficiency of a protein of choice in an *in vitro* system. Given the sensitivity and limited dynamic range of the assay and the fact that the effect of leucine represents a physiological adaptation rather than a

binary switch, we did not expect the assay to detect significant changes in individual proteins. That said, the cumulative evidence from our previous findings and newly added experiments consistently supports our finding regarding the increase in mitochondrial protein import:

1. In our proteomics analysis, we demonstrate that mitochondrial protein content increases even in the presence of cycloheximide both in *C. elegans* and human cells (Fig. 4c, h). This excludes the possibility that protein translation increases mitochondrial protein content, suggesting that protein import into mitochondria increases instead.

Taken from Fig. 4c, h.

2. We performed a chase assay and observed a significantly increased level and stability of the TOM complex subunit TOMM40 (Extended Data Fig. 11a, b), which supports the idea that there is enhanced protein import into mitochondria.

Taken from Extended Data Fig. 11a, b.

3. In a new proteomic analysis of worms overexpressing CDC-48, whose OMM proteins are more efficiently degraded, we found that leucine could not increase the content of OMM proteins or mitochondrial proteins (Fig. 5c-f) in contrast to WT animals. This suggests that the increase in import induced by leucine is prevented in worms that overexpress CDC-48, which does not stabilize OMM proteins.

Taken from Fig. 5c-f.

7. Consequently, despite the inhibition of outer mitochondrial membrane (OMM) protein turnover by leucine, as shown in the previous section, it remains unclear how leucine exerts its effect on OMM protein levels, subsequently increasing mitochondrial import efficiency and respiration. The existing gaps between these conclusions and the lack of experimental support underscore the study's reliance on correlations rather than robust causal relationships.

As discussed above, we have now established a clear mechanism by which leucine regulates OMM degradation and its physiological consequences. The revised model is shown in Fig. 5s: High levels of leucine inhibit GCN2 and reduce SEL1L-HRD1-dependent degradation of OMM proteins, thereby stabilizing key components of the protein import machinery, such as TOMM40. This leads to the expansion of the mitochondrial proteome for the respiratory chain, the TCA cycle, metabolism, and metabolite transport. Consequently, mitochondrial respiratory capacity and cell viability are enhanced.

Taken from Fig. 5s.

Minor Concerns:

1. Line 68 states that "mitoUFD was ubiquitously expressed in all *C. elegans* tissue," but no corresponding result is presented to support this claim.

We thank the reviewer for the comment. First, the *eft-3* promoter is known to be expressed in all *C. elegans* tissues. Second, we presented images of the entire worm showing the expression pattern of the mitoUFD substrate (**Extended Data Fig. 1a, b**).

2. From Fig. 2, the authors primarily rely on the percentage change of BioSorter fluorescence intensity to indicate alterations in mitoUFD content. However, this measurement may be influenced by various factors such as mitochondrial mass. It may be beneficial to conduct further validations using fluorescence microscopy and Western Blot analysis.

As shown in previous Fig. 2 and now in **Extended Data Fig. 2f, g**, we provided western blot and confocal images of the mitoUFD substrate.

3. The use of the term "crosstalk" in Line 94 might be overstated and potentially misleading, as it implies a relationship between two factors solely based on their mutual influence.

We removed this statement due to an overall change in focus of the revised manuscript.

4. In Fig. 1e, f, and S1b, the "hecd-1" and "rpn-8" groups can be interchanged to align with their first mention in the text.

We have removed the previous data regarding HECD-1 and RPN-8. The new set of results in the revised manuscript provides stronger support for the previous conclusion about the leucine-controlled degradation of OMM proteins.

5. It is unclear which animal group serves as the control for calculating the percentage change in Fig. 3a.

As previously explained, this graph has been removed. However, we have changed the title of the Y-axis to make the comparisons clearer.

6. The insets of Fig. 4b should include the Y-axis title to prevent confusion.

This is now **Fig. 2e**. The two graphs have the same Y-axis title.

7. If applicable, additional p-values of non-significant groups can be displayed in the figures.

We have now included the p-value of non-significant groups in the figures.

8. The Discussion section would benefit from a comprehensive evaluation of the study's limitations and suggestions for future research directions.

We revised the manuscript extensively and included a large amount of new data. We also changed the discussion section, accordingly, addressing the limitations of the current study and possible future research directions.

References:

1. Segref, A. *et al.* Pathogenesis of Human Mitochondrial Diseases Is Modulated by Reduced Activity of the Ubiquitin/Proteasome System. *Cell Metab* **19**, 642–652 (2014).

2. Efstathiou, S. *et al.* ER-associated RNA silencing promotes ER quality control. *Nat Cell Biol* **24**, 1714–1725 (2022).
3. Wei, X. *et al.* Proteomic screens of SEL1L-HRD1 ER-associated degradation substrates reveal its role in glycosylphosphatidylinositol-anchored protein biogenesis. *Nat. Commun.* **15**, 659 (2024).
4. Zhou, Z. *et al.* Endoplasmic reticulum-associated degradation regulates mitochondrial dynamics in brown adipocytes. *Science* **368**, 54–60 (2020).
5. Metzger, M. B., Scales, J. L., Dunkleberger, M. F., Loncarek, J. & Weissman, A. M. A protein quality control pathway at the mitochondrial outer membrane. *Elife* **9**, e51065 (2020).
6. Heo, J.-M. *et al.* A Stress-Responsive System for Mitochondrial Protein Degradation. *Mol Cell* **40**, 465–480 (2010).
7. Liao, P.-C. & Pon, L. A. Analysis of the mitochondria-associated degradation pathway (MAD) in yeast cells. *Methods Enzym.* **707**, 585–610 (2024).
8. Stolz, A., Hilt, W., Buchberger, A. & Wolf, D. H. Cdc48: a power machine in protein degradation. *Trends Biochem Sci* **36**, 515–523 (2011).
9. Wu, X., Li, L. & Jiang, H. Doa1 targets ubiquitinated substrates for mitochondria-associated degradation. *J Cell Biology* **213**, 49–63 (2016).
10. Mårtensson, C. U. *et al.* Mitochondrial protein translocation-associated degradation. *Nature* **569**, 679–683 (2019).
11. Karbowski, M. & Youle, R. J. Regulating mitochondrial outer membrane proteins by ubiquitination and proteasomal degradation. *Curr Opin Cell Biol* **23**, 476–482 (2011).
12. Ravanelli, S. *et al.* Reprogramming of proteasomal degradation by branched chain amino acid metabolism. *Aging Cell* e13725 (2022) doi:10.1111/acel.13725.
13. Ryan, D. G. *et al.* Disruption of the TCA cycle reveals an ATF4-dependent integration of redox and amino acid metabolism. *Elife* **10**, e72593 (2021).
14. Murschall, L., Peker, E., MacVicar, T., Langer, T. & Riemer, J. Protein Import Assay into Mitochondria Isolated from Human Cells. *BIO-Protoc.* **11**, e4057 (2021).

Point-by-point response to reviewer comments

Reviewer #1:

In the revised manuscript, the authors include several new experiments that address most of the issues from the original submission. Furthermore, they established new connections and extended endogenous protein analyses, as well as added validation in cultured mammalian model cells. The well written work is easy to follow and describes novel interesting findings. I believe that the current version of this manuscript could be accepted for publication without any additional experimental work.

We would like to sincerely thank the reviewer for the constructive feedback that helped us improve our study. We are also grateful for the acknowledgment of our efforts to address the original concerns and for the positive comments about the originality and clarity of our work.

Reviewer #2:

The authors have adequately addressed my queries.

We would like to thank the reviewer for the helpful feedback on how we can improve our study. We are also grateful for the positive comments about our work.

Reviewer #3:

While the revised version of the manuscript demonstrates improved data quality with the addition of new experiments, several critical issues remain unresolved, and some data from the previous submission have been removed, which I find concerning. It is somewhat disappointing that the authors removed the data related to mitochondrial ETC inhibitors, particularly given the original proposal that decreased mitochondrial activity could serve as a trigger for the increased accumulation of OMM proteins to compensate for impaired mitochondrial function. This crucial experiment had the potential to establish a clear link between mitochondrial dysfunction and the turnover of OMM proteins. Its removal weakens the significance of the overall discovery, and the interpretation of the results lacks the necessary context to justify the conclusions drawn.

We thank the reviewer for this comment and acknowledge the concern about removing the ETC inhibition data. As we discussed during the initial round of revisions, the extent and mode of ETC inhibition can affect the degradation of outer mitochondrial membrane (OMM) proteins differently. Additionally, RNA interference (RNAi) knockdown of ETC subunits can significantly impact cytosolic proteostasis. However, due to the complexity of ETC perturbation and its potential indirect effects, we focused exclusively on regulating OMM proteostasis with leucine and adapting mitochondrial respiratory activity. Our research revealed an unknown pathway in which leucine acts as a nutrient signal that drives mitochondrial respiration by remodeling OMM protein ubiquitylation and degradation. While we agree that the connection between ETC dysfunction and OMM turnover is interesting, further investigation is warranted. However, this topic is beyond the scope of this study. We

appreciate the reviewer's suggestion and are committed to pursuing this line of research in future studies.

The manuscript still struggles to explain why specifically leucine, and no other amino acids, is the key modulator of OMM protein turnover. Despite the increased experimental effort to dissect the mechanism through the GCN2-SEL1 axis, it remains puzzling why leucine specifically regulates the turnover of OMM proteins.

We thank the reviewer for this comment and agree that the specific property of leucine that gives it its unique role in OMM protein degradation is unclear. As outlined in the original manuscript, our initial genetic screening combined with amino acid supplementation revealed that leucine supplementation alone stabilizes OMM proteins and enhances mitochondrial respiration. Additionally, as discussed in the first round of revisions, further analysis of the branched-chain amino acids isoleucine and valine revealed no significant changes in mitochondrial UFD levels in worms. Through Seahorse analysis of other essential amino acids in cell culture, we found that leucine is the only amino acid that significantly increases mitochondrial respiration. Together, these findings underscore the distinctive function of leucine in metabolic adaptation. Further investigation into the biochemical or structural properties unique to leucine and its interplay with GCN2 is necessary to uncover the mechanistic basis of this specificity.

The authors have made considerable efforts to elucidate the mechanism through which leucine mediates OMM protein turnover via the GCN2-SEL1 axis. However, the rationale for specifically targeting mitochondrial-localized SEL1 remains unclear. The mechanism behind this specificity should be explored in more depth to understand why the mitochondria are the primary site of action for SEL1 in this process. Additionally, the immunogold analysis presented to support the mitochondrial localization of SEL1 is not very convincing. The quality of the images raises concerns, and the arrows indicating SEL1 localization appear to point to regions where the mitochondria and ER come into close contact. This weakens the argument that SEL1 is truly localized to the mitochondria.

We thank the reviewer for the thoughtful comment and apologize for any confusion caused. As stated in our initial revisions, our conclusion that SEL-1 acts at the OMM is not based solely on immunogold labeling. Rather, this conclusion is supported by multiple approaches, including a subcellular fractionation assay, a MitoUFD pulldown experiment, and a Seahorse oxygen consumption rate measurement. These independent lines of evidence strongly suggest that SEL-1 plays a direct and functionally significant role in regulating proteostasis and mitochondrial respiration at the OMM. We acknowledge the reviewer's concern regarding the immunogold data. While we agree that the resolution of the images does not fully rule out the presence of SEL-1 at endoplasmic reticulum (ER)-mitochondria contact sites, this does not contradict the role of SEL-1 at OMM. As discussed in the manuscript, the precise mechanism by which SEL-1 is targeted to the OMM requires further investigation. We do not exclude the possibility that the ER may serve as a reservoir or trafficking route for SEL-1 delivery to mitochondria.